# The medial septum controls hippocampal supra-theta oscillations

Bálint Király [1,2], Andor Domonkos [3], Márta Jelitai[3], Vítor Lopes-dos-Santos [4], Sergio Martínez-Bellver[1,5], Barnabás Kocsis[1,6], Dániel Schlingloff[1], Abhilasha Joshi [7], Minas Salib [7], Richárd Fiáth [6,8], Péter Barthó[8], István Ulbert [6,8], Tamás F. Freund [9], Tim J. Viney [7], David Dupret [4], Viktor Varga [3] & Balázs Hangya [1] ✉

Hippocampal theta oscillations orchestrate faster beta-to-gamma oscillations facilitating the segmentation of neural representations during navigation and episodic memory. Supra-theta rhythms of hippocampal CA1 are coordinated by local interactions as well as inputs from the entorhinal cortex (EC) and CA3 inputs. However, theta-nested gamma-band activity in the medial septum (MS) suggests that the MS may control supra-theta CA1 oscillations. To address this, we performed multi-electrode recordings of MS and CA1 activity in rodents and found that MS neuron firing showed strong phase-coupling to theta-nested supra-theta episodes and predicted changes in CA1 beta-to-gamma oscillations on a cycle-by-cycle basis. Unique coupling patterns of anatomically defined MS cell types suggested that indirect MS-to-CA1 pathways via the EC and CA3 mediate distinct CA1 gamma-band oscillations. Optogenetic activation of MS parvalbumin-expressing neurons elicited theta-nested beta-to-gamma oscillations in CA1. Thus, the MS orchestrates hippocampal network activity at multiple temporal scales to mediate memory encoding and retrieval.

Hippocampal theta oscillations (4–12 Hz) of the local field potential (LFP) reflect rhythmic inputs that orchestrate neuronal firing, occurring typically during exploratory or memory-guided behaviors or rapid eye movement (REM) sleep, and have been linked to learning and memory[1–9]. Previous studies revealed an important diversity of theta cycles along different dimensions, including length[10–13] and shape[14] of the cycles or the presence of phase-coupled oscillations in the beta (12-30 Hz) and gamma (30-140 Hz) frequency bands[15–22]. Importantly, in rodents performing memory tasks, cycle-by-cycle variations with differential contributions to memory processes were found based on theta-nested spectral components (tSCs), related to distinct spiking dynamics of CA1 pyramidal neurons[15,23].

The medial septum has a broad role in coordinating hippocampal rhythmic activity through cholinergic, GABAergic and glutamatergic projections[12,24–33]. Importantly, the MS has long been proposed as a key nucleus in the generation of hippocampal theta oscillations[30,34–38], while the roles of other subcortical structures including the supra-mammillary nucleus[39–41] and the nucleus incertus[42–46] are increasingly recognized. Parvalbumin-expressing (PV) GABAergic neurons of the MS, targeting the inhibitory neurons of the CA1, CA3 and dentate gyrus areas[24,47,48], have been suggested to play a pivotal role in controlling

[1]Lendület Laboratory of Systems Neuroscience, Institute of Experimental Medicine, Budapest, Hungary. [2]Department of Biological Physics, Institute of Physics, Eötvös Loránd University, Budapest, Hungary. [3]Subcortical Modulation Research Group, Institute of Experimental Medicine, Budapest, Hungary. [4]Medical Research Council Brain Network Dynamics Unit, Nuffield Department of Clinical Neurosciences, University of Oxford, Oxford, UK. [5]Department of Anatomy and Human Embryology, Faculty of Medicine and Odontology, University of Valencia, Valencia, Spain. [6]Faculty of Information Technology and Bionics, Pázmány Péter Catholic University, Budapest, Hungary. [7]Department of Pharmacology, University of Oxford, Oxford, UK. [8]Institute of Cognitive Neuroscience and Psychology, Research Centre for Natural Sciences, Budapest, Hungary. [9]Laboratory of Cerebral Cortex Research, Institute of Experimental Medicine, Budapest, Hungary. ✉e-mail: hangya.balazs@koki.hu

hippocampal theta rhythm[49–52]. Theta waves entrain faster, supra-theta oscillations of the beta and gamma bands, thought to be routed from different areas representing distinct memory processes[15,18,22,53,54]. According to the current models, these oscillations are generated by different extra- and intrahippocampal sources, mainly through inputs from the CA3 (slow gamma, 30–50 Hz), the entorhinal cortex (mid-gamma, 50–100 Hz) or local CA1 networks (fast gamma, 100–140 Hz)[19,53–60]. However, the burst structure of MS neurons suggests the septal presence of beta/gamma-band spectral components[61–63]. Indeed, entorhinally projecting Orchid neurons of the MS can couple to CA1 mid-gamma oscillations[64], and low-rhythmic MS neurons, projecting to CA3 and dentate gyrus, can couple to slow- and mid-gamma bands in dorsal CA1[47]. This raises the question whether hippocampal theta-nested beta and gamma oscillations couple to MS activities.

To address whether supra-theta oscillations in the CA1 are mediated by the MS, we recorded CA1 LFP and MS single neuron activity simultaneously in freely moving mice as well as urethane-anesthetized mice and rats. We uncovered a wide-spread presence of beta/gamma band spectral components in the MS, which showed transient coherence with CA1 spectra at the time of MS neuron bursts. In the CA1, we identified distinct tSCs representing beta (tSC1), slow gamma (tSC2), mid-gamma (tSC3 and tSC4) and fast gamma (tSC5) oscillations[15,53,54,57–59]. We found that firing rates and theta-coupling of MS neurons changed with the occurrence of distinct CA1 tSCs, suggesting that tSCs are also represented in MS network states. Furthermore, the majority of MS neurons, including most constitutively theta-rhythmic cells, showed strong phase-coupling to one or more CA1 tSCs. To better understand the causal relations in the reciprocally connected septo-hippocampal circuit, we determined that most MS neurons best locked to phase values of the tSC signals occurring at a small temporal delay, i.e., activity changes of most MS neurons anticipated correlated changes in hippocampal tSCs. Next, we investigated anatomically defined MS Orchid, Teevra and Low-rhythmic Neurons[47,48,64] with known projection targets, which revealed the coupling of entorhinal-projecting, parvalbumin-expressing MS Orchid cells to mid-gamma components and CA3-projecting MS cells to slow gamma components, suggesting that indirect pathways between the MS and CA1 via EC vs. CA3 mediate different gamma components. These results encouraged us to test whether MS GABAergic neurons are capable of evoking theta-nested beta and gamma oscillations, by optogenetically stimulating PV-expressing MS neurons with theta-modulated bursts of laser light, mimicking physiological tSCs. We found that MS PV stimulation elicited artificial tSCs in the CA1 that resembled their physiological counterparts in spectral content and laminar distribution, demonstrating that the MS GABAergic network can mediate supra-theta hippocampal oscillations. In contrast with previous models, these results raise the possibility that the MS has a hitherto overlooked role in CA1 oscillation genesis beyond the theta frequency range.

## Results

### Supra-theta spectral components are present in the MS and the CA1 of freely moving mice

To test whether MS neuronal activity correlates with hippocampal theta-nested beta and gamma oscillations, we analyzed dual silicon probe recordings from mice ($n = 6$) moving along a linear track, monitoring CA1 local field potentials (LFPs) concurrently with MS neuronal spiking ($n = 365$ neurons; Fig. 1a). Many MS neurons fired bursts of action potentials, with inter-spike intervals corresponding to beta/gamma band activity (Fig. 1a, b). In accordance, spectral analysis revealed strong beta/gamma band activities besides theta-rhythmicity of MS single neurons (Fig. 1c). We tested whether these beta/gamma band spectral components were coherent with CA1 LFP and found an increase in MS-CA1 coherence in the supra-theta bands after the burst

onset of individual MS units lasting ~50 ms, likely corresponding to bursts of action potentials (Fig. 1d).

Next, to decompose the CA1 LFP recorded from the radiatum layer into theta and supra-theta components, we applied single-cycle spectral profiling using empirical mode decomposition (EEMD), an adaptive unsupervised spectral decomposition technique well-suited for non-stationary signals[65]. By performing independent component analysis (ICA) on the supra-theta spectral content of the theta cycles, we confirmed the presence of five different theta-nested spectral components[15] (tSCs; Fig. 1e) that consistently occurred across individual theta cycles, but remained hidden when only the average spectral signature of all theta cycles was considered (Fig. 1f). Each tSC attributed large weights to different characteristic frequency components in the frequency range spanning beta to fast gamma bands (Fig. 1g and Supplementary Fig. 1a, peak frequencies of the average power spectra across all recordings: tSC1: 20 Hz, tSC2: 32 Hz, tSC3: 50 Hz, tSC4: 74 Hz, tSC5: 179 Hz). All five tSCs were robustly detected across mice ($n = 6$) and recordings ($n = 31$) and an average of 30% of all theta cycles contained at least one strong tSC (see Methods).

We asked whether the presence of strong tSCs was correlated with other parameters in which theta cycles showed diversity. First, we found that theta cycles strongly expressing a tSC had a higher theta frequency than those containing no strong tSCs (up to >1 Hz difference on average) and cycles with gamma frequency tSCs (tSC2-5) had higher theta frequency than tSC1-expressing cycles (Fig. 1h and Supplementary Fig. 1b, repeated measures ANOVA with Tukey's test for post hoc analysis, $p < 0.001$). We also found a positive correlation between the strength of each tSC and theta frequency (Supplementary Fig. 1d, Spearman correlation, $p < 0.001$, tSC4 strength was the most correlated). Second, we investigated whether cycles during long (>30 s) continuous theta-dominant periods differed from those during shorter periods or near theta segment boundaries. During long theta epochs, tSCs were more frequent, and each tSC was detected in similar proportions of theta cycles (similarly to ref. 15). In contrast, both during short theta epochs and non-theta-to-theta transitions, tSC1-expressing cycles were overrepresented and gamma frequency tSCs (tSC2-5) were underrepresented (Fig. 1i).

Since hippocampal theta and gamma oscillations are correlated with movement and velocity, we tested the expression of tSCs as a function of animal speed. In line with previous studies[66], we found that mice were significantly faster during theta cycles with strong gamma tSCs (tSC2-5) compared to tSC1-expressing cycles or cycles with no tSC (repeated measures ANOVA with Tukey's test for post hoc analysis, $p < 0.01$, Fig. 1j and Supplementary Fig. 1c). The strength of gamma tSC expression was positively correlated with animal speed (Supplementary Fig. 1e, Spearman correlation, $p < 0.001$, strongest correlation for tSC4). In contrast, tSC1 expression showed a weak negative correlation with speed. Since theta frequency also correlated with animal speed, we calculated partial correlations among tSC strength, theta frequency and velocity. All correlations remained strongly significant ($p < 0.001$) and decreased only slightly by removing the effect of the third variable, demonstrating that these correlations occurred independently (Supplementary Fig. 1f). Overall, these data confirmed the presence of distinct theta-nested spectral components in mouse CA1 LFP recordings.

### MS single neuron firing is correlated with hippocampal tSCs

To test whether MS neurons' activity patterns are linked to hippocampal tSCs, we first examined changes in the firing rate of MS single units across theta cycles with different tSCs ($n = 198$ MS neurons with firing rate > 3 Hz and stable spike amplitude). We found that MS neurons were more active during theta cycles expressing tSCs than during those without strong tSC presence (Figs. 2a and 2b left, repeated measures ANOVA with Tukey's test for post hoc analysis, $p < 0.01$). Furthermore, the occurrence of gamma frequency tSCs, particularly

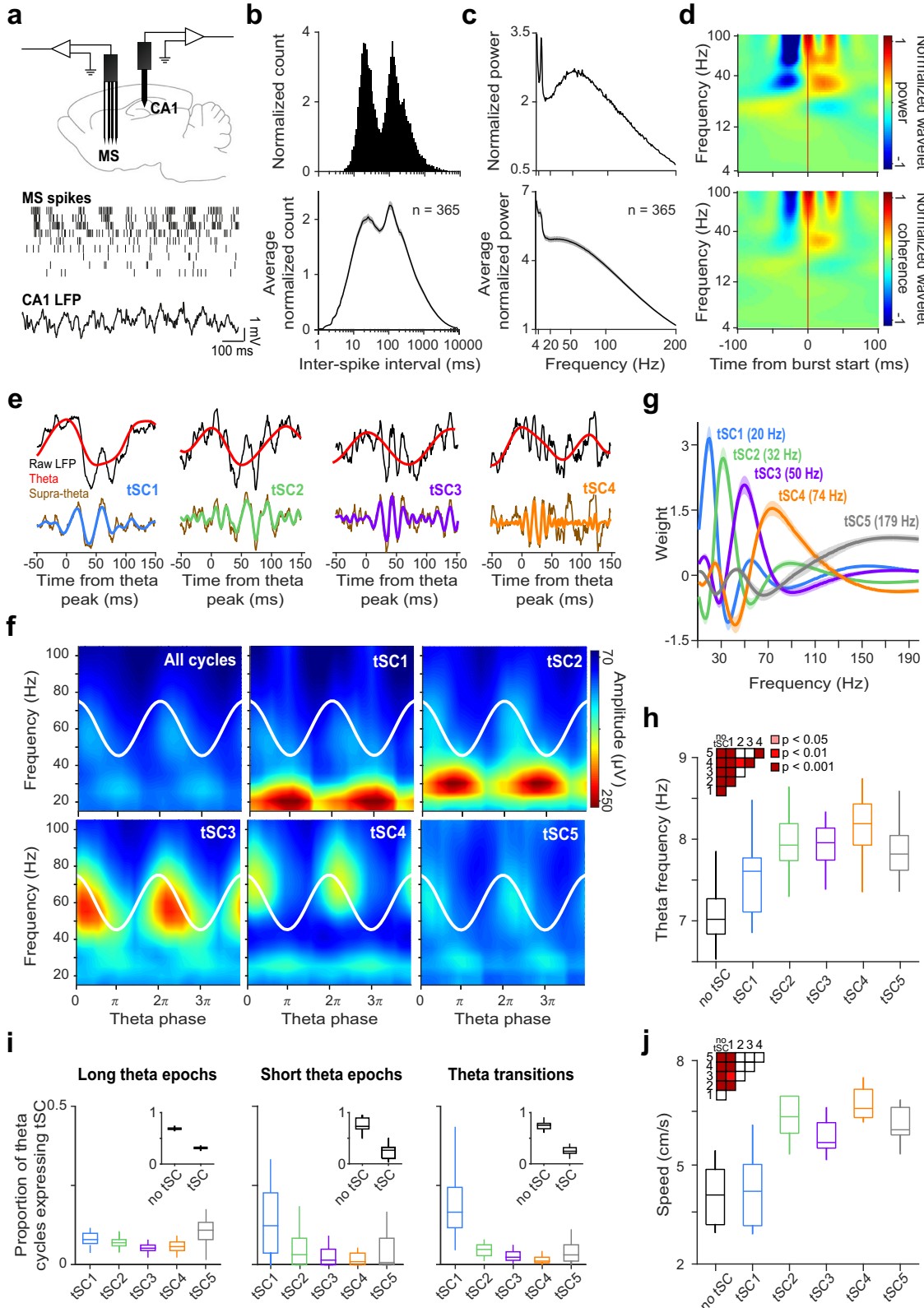

tSC4 in the mid-gamma range, were associated with the highest MS firing rates (82%, 163/198 neurons were most active during theta cycles with gamma-band tSCs, Figs. 2a and 2b, repeated measures ANOVA with Tukey's test for post hoc analysis, $p < 0.05$, example neurons #1 and #2). However, there was a subset of MS neurons (14%, 27/198) that were the most active during cycles expressing the beta range tSC1 component (Figs. 2a and 2b right, example neuron #3).

Many MS neurons show phase-coupling to hippocampal theta oscillation[63,67]. We hypothesized that a possible association between hippocampal tSCs and MS activity might also be reflected in differential theta-phase coupling properties of MS neurons. Therefore, we tested whether MS neurons phase-coupled to hippocampal theta (91%, 181/198; Rayleigh's test for circular uniformity, $p < 0.01$) changed their preferred theta phase (mean angle) and phase-coupling strength

**Fig. 1 | Supra-theta spectral components in the MS and the hippocampal CA1 radiatum layer of freely moving mice. a** Top, schematic of dual silicon probe recordings from the hippocampal CA1 region and the MS. Bottom, example of synchronously recorded CA1 LFPs and MS extracellular spike rasters. **b** Inter-spike interval histogram of an example MS neuron (top) and average of all recorded MS neurons (bottom, $n = 365$ neurons from 6 awake mice). **c** Power spectrum of an example MS neuron (top) and average of all recorded MS neurons (bottom, $n = 365$). **d** Top, normalized average wavelet spectrogram of an example MS spike train (convolved by a Gaussian window, see Methods) triggered on first spikes of bursts. Bottom, average coherence between MS spiking and CA1 LFP triggered on first spikes of bursts of an example MS neuron. Spectrograms were normalized by the mean power for each frequency, thus visualizing the spectral changes around the triggering event. **e** Example theta cycles dominated by different tSCs. Black, raw LFP signal; red, theta band signal; brown, supra-theta signal extracted by EEMD. Blue, green, purple and orange lines show the dominant tSCs in each example cycle.

**f** Mean amplitude of supra-theta spectral components as a function of theta phase was computed from the raw LFP of an example session, for all cycles (top left) and for cycles strongly expressing a given tSC. Two theta cycles are shown, indicated by white cosine curves. **g** Average power spectrum of tSCs across all sessions. Peak frequencies are shown in the brackets. Error shades show the standard error of the mean. **h** Frequency of theta cycles expressing different tSCs. **i** Average proportion of theta cycles expressing each tSC during long theta periods (left), short theta epochs (middle) and around theta segment boundaries (right). Insets show the proportion of theta cycle expressing any of the tSCs. **j** Speed of mice during theta cycles expressing different tSCs. Box-whisker plots show median, interquartile range and non-outlier range in panels **h**–**j**. Differences were tested with two-sided repeated measures ANOVA and Tukey's post hoc test and significant differences are indicated by the color-coded matrices in (**h** and **j**). Source data are provided as a Source Data file.

---

(mean resultant length) according to the presence of hippocampal tSCs. Indeed, we found many examples of MS neurons showing different phase-coupling strength (Fig. 2a), and sometimes even different preferred theta phase (example neuron #2) depending on which tSC was present. On average, theta-coupling was strongest in theta cycles expressing gamma frequency tSCs, particularly mid-gamma tSC4 (repeated measures ANOVA with Tukey's test for post hoc analysis, $p < 0.001$), and weakest in cycles expressing the beta range tSC1 ($p < 0.01$, Fig. 2c and d). The majority of theta-coupled MS neurons fired at an earlier phase in tSC1-expressing cycles compared to other theta cycles, most prominently in comparison with tSC4-expressing cycles (60%, 108/181, Fig. 2e; largest difference in preferred phase was found for the neurons most active during tSC1 cycles; Supplementary Fig. 2).

MS neurons often show phase-locked rhythmic bursting activity during hippocampal theta oscillations[48,50,68,69]. Furthermore, constitutively bursting (CB) MS neurons, which express theta-rhythmic burst firing both in the presence and absence of CA1 theta rhythms, have been proposed to be important part of the theta-generating subcortical network (referred to as putative pacemakers by some studies)[48–50,61,63]. To test whether this rhythmicity property is correlated with the tSC-related firing behavior of MS neurons, we classified MS neurons based on their rhythmicity during hippocampal theta and non-theta periods[52] (Supplementary Fig. 3, see also Methods). Along with the putative pacemaker constitutively bursting neurons (10% of all recorded MS units, Supplementary Fig. 3a–d), we identified two additional special subpopulations based on their rhythmic properties. Theta-associated bursting neurons (12%) were theta-rhythmic but only when theta oscillation was detected in the hippocampus (Supplementary Fig. 3e–h). Tonically active neurons (3%) showed regular rhythmic activity with little to no phase coupling to hippocampal LFP (Supplementary Fig. 3i–l). Constitutively bursting and theta-associated bursting populations showed increased firing rate and theta coupling in theta cycles with gamma-band tSCs, especially tSC4 (Supplementary Figs. 2g, h and 4b, c), similarly to the entire population of theta-coupled MS neurons (Fig. 2b–d). Interestingly, tonically active MS neurons were also firing more in theta cycles with gamma-band tSCs compared to those expressing tSC1, despite their lack of phase coupling to theta (Supplementary Fig. 4d, repeated measures ANOVA with Tukey's test for post hoc analysis, $p < 0.05$).

Finally, we tested whether burst properties such as intra-burst frequency, intra-burst spike number or theta-cycle skipping[47,70,71] showed tSC-associated differences. We found that while intra-burst frequencies of theta-coupled MS neurons were higher in cycles with strong tSCs (except for tSC5 cycles, Fig. 2f, repeated measures ANOVA with Tukey's test for post hoc analysis, $p < 0.05$), burst parameters were otherwise surprisingly stable across different types of theta cycles (Supplementary Fig. 4a-c).

## Most MS neurons are phase-coupled to beta-gamma band hippocampal tSCs

In the previous section we demonstrated that the firing of MS single units was robustly modulated on a theta cycle basis according to the presence of different tSCs. To directly test whether MS neurons' spiking correlated with hippocampal tSCs on shorter time scales, we first calculated hippocampal LFP averages triggered on the spikes of each medial septal neuron (spike triggered LFP average, STA; Fig. 3a, top). This analysis allows visualizing the average hippocampal local field activity around MS action potentials. Beyond the well-documented phase locking of septal firing to hippocampal theta cycles, these STA calculations often revealed additional faster oscillatory components, indicating that MS neurons may phase lock to supra-theta hippocampal frequency bands as well (Fig. 3a top, black arrowheads). To confirm this, we next calculated average LFP spectrograms triggered on MS action potentials (spike triggered LFP spectrograms, STS; Fig. 3a). Analogous to STA, this analysis shows the average LFP spectral amplitude and phase around MS spikes. We normalized these spectrograms for each individual frequency, revealing those spectral components that change near MS firing. This normalization removed the stationary theta-frequency component from the STS power and clearly revealed spectral increases associated with MS firing in the beta and gamma bands (Fig. 3a, middle and bottom).

This prompted us to test whether individual MS neurons showed phase-coupling to hippocampal tSCs. Indeed, we found many examples where MS spikes were coupled to tSC phases (Fig. 3b and Supplementary Fig. 5). Neurons phase-coupled to locally or distantly recorded oscillatory activity often show a temporal offset between the two signals, which can be estimated by calculating the phase-locking strength as a function of possible time lags and localizing the maximum of this function (Z-shift method)[49,72]. To estimate and account for these offsets, we calculated Rayleigh's Z-statistic for each MS neuron, quantifying phase-locking strength, with temporal shifts added to the spikes in the −100 ms – 100 ms range. We found that 51% of all recorded MS neurons were phase-coupled to at least one tSC (Fig. 3c; $p < 0.05$ at any of the temporal shifts, Rayleigh's test for circular uniformity, type I errors were controlled with Storey's false discovery rate method[73]; no robust phase coupling to tSC5 were found). This was not explained by theta-phase correlations of tSCs and MS spikes, since shuffling spikes across theta cycles reduced coupling strength in 93% of cases (Supplementary Fig. 6a) and removed most of the significant phase-coupling of MS neurons to tSCs (72, 95, 100 and 95% for tSC1, 2, 3 and 4, respectively; Supplementary Fig. 6b). Moreover, a closer look at intra-burst spike timing revealed a subset of MS neurons showing correlations of MS intra-burst frequencies with the concurrent tSCs (Supplementary Fig. 7). The preferred

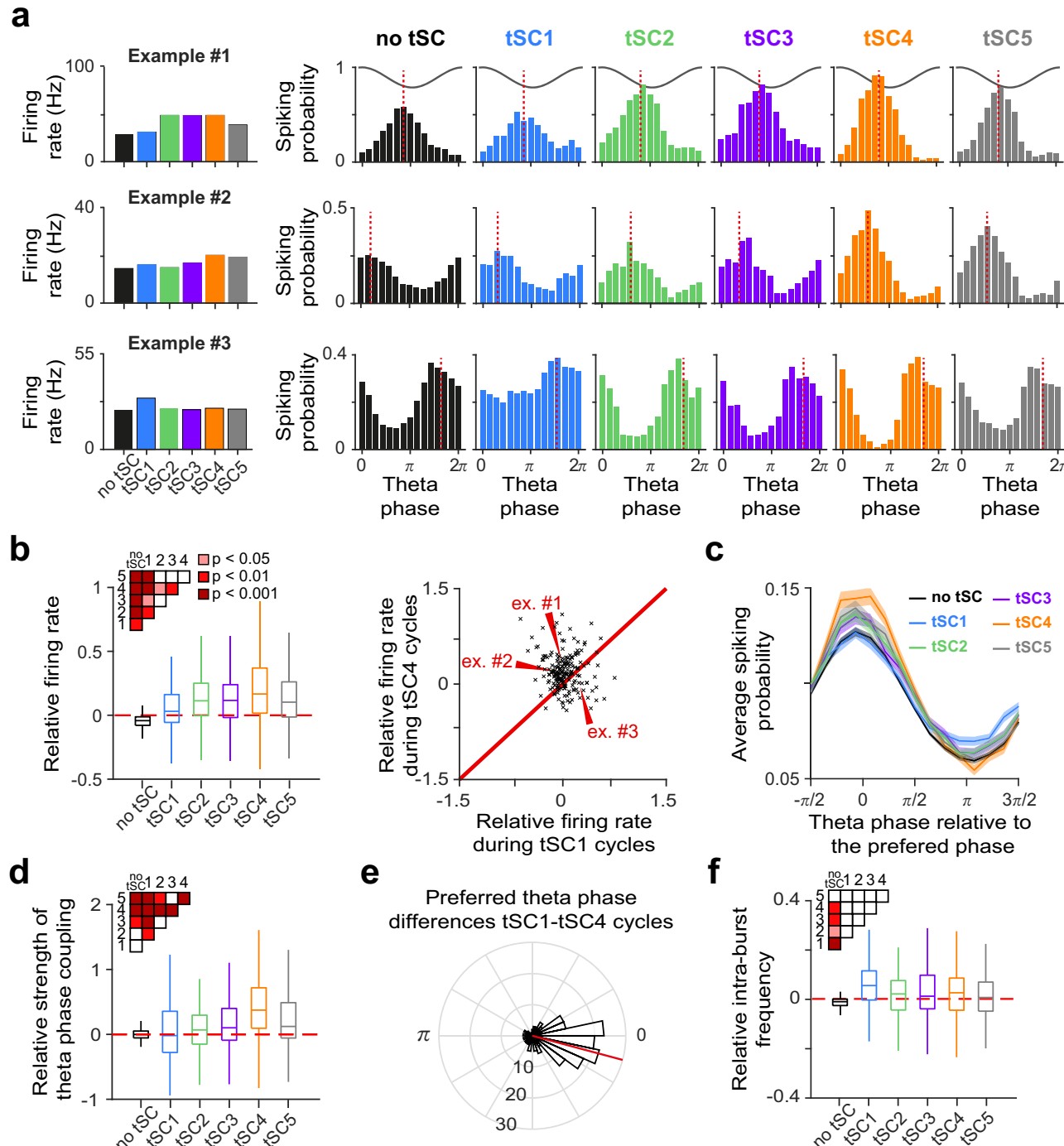

**Fig. 2 | MS single neuron firing is correlated with hippocampal tSCs. a** Three example neurons' average firing rate (left) and theta phase histogram (right) during theta cycles expressing different tSCs. Cosine curves indicate the theta phase; red dashed lines show the preferred phase. **b** Left, firing rate distribution of MS neurons ($n = 198$) as a function of tSC presence, relative to the average firing rate over all cycles (red dashed line). Boxes and whiskers show median, interquartile range and non-outlier range. Differences were statistically tested with two-sided repeated measures ANOVA, followed by Tukey's test for post hoc comparison. Significant differences are indicated by the color-coded matrix in the inset. Right, relative firing rate of MS neurons during tSC1-expressing theta cycles vs. tSC4-expressing cycles. Example neurons from panel **a** are marked by red arrowheads. **c** Average theta phase histogram (relative to the preferred phase) of theta-phase-coupled MS neurons' ($n = 181$) firing during theta cycles with different tSCs. Error shades show the standard error of the mean. **d** Phase coupling strength distribution of theta-coupled MS neurons ($n = 181$) measured by the mean resultant length over cycles

expressing a particular tSC, relative to the average coupling strength over all cycles (red dashed line). Boxes and whiskers show median, interquartile range and non-outlier range. Differences were statistically tested with two-sided repeated measures ANOVA, followed by Tukey's test for post hoc comparison. Significant differences are indicated by the color-coded matrix in the inset. **e** Phase histogram of differences between the preferred phase of theta-phase-coupled MS neurons in tSC1 and tSC4 cycles. The red line indicates the mean difference. **f** Intra-burst frequency distribution of theta-coupled MS neurons ($n = 181$) as a function of tSC expression, relative to the average intra-burst frequency over all cycles (red dashed line). Boxes and whiskers show median, interquartile range and non-outlier range. Differences were statistically tested with two-sided repeated measures ANOVA, followed by Tukey's test for post hoc comparison. Significant differences are indicated by the color-coded matrix in the inset. Source data are provided as a Source Data file.

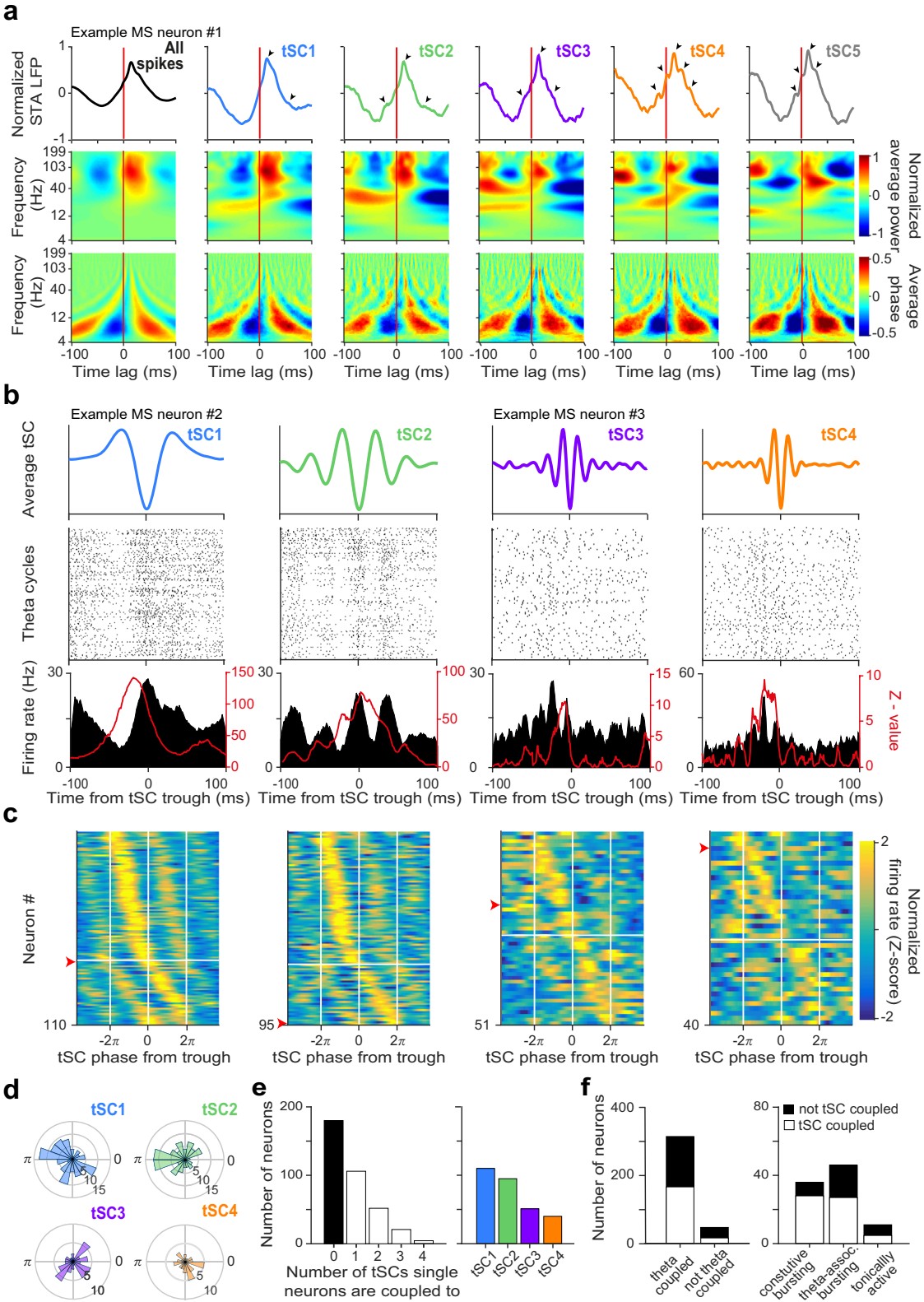

phase of MS neurons covered the entire tSC cycles, albeit not uniformly in case of tSC1 and tSC2 (Fig. 3d, Rayleigh-test; tSC1, $p = 0.0446$; tSC2, $p = 0.0024$). Several neurons were coupled to more than one tSC and coupling to slower tSCs was more frequent than to faster tSCs (Fig. 3e and Supplementary Fig. 5). Phase-locking to tSCs was more common among cells phase-coupled to hippocampal theta as well (53% vs. 35% in the non-theta-coupled subpopulation), and tSC-coupled MS neurons showed higher firing rate, stronger theta-coupling, longer bursts and more gamma-band inter-spike intervals than MS neurons not significantly coupled to tSCs (Supplementary Fig. 8). Interestingly, the majority of constitutively bursting (78%) and theta-associated bursting (59%) neurons were strongly coupled to tSCs, while 5 of 11 tonically active neurons showed tSC coupling (Fig. 3f).

**Fig. 3 | MS neurons show phase coupling to hippocampal tSCs. a** Top, spike triggered averages (STA) of the raw CA1 LFPs of an example MS neuron. Theta cycles were sorted by tSC expression (from left to right, all cycles included, theta cycles expressing tSC1 to tSC5). Arrowheads indicate tSC peaks. Middle, spike triggered spectral power images (STS power) of the CA1 LFP for the same example neuron. Bottom, corresponding spike triggered spectral phase images (STS phase) of the same MS neuron. **b** Firing pattern of two tSC-coupled example MS neurons (example neuron #2 to tSC1 and tSC2, example neuron #3 to tSC3 and tSC4). Top, average tSC signals. Middle, raster plots of spike times aligned to the most negative trough of the tSC signal within each theta cycle. Bottom, peri-event time histograms (PETHs) corresponding to each spike raster (black; y-axis on the left) and Rayleigh's *Z*-value as a function of temporal offset between hippocampal tSCs and MS spike trains (red; y-axis on the right). **c** *Z*-scored phase histograms of all tSC-coupled MS neurons, sorted into four groups based on the tSC they are coupled to (blue, low firing rate; yellow, high firing rate). Zero phase corresponds to tSC troughs (white vertical lines). Cells within each group are sorted by their preferred phase in two blocks: top, cells with maximum firing rate before the most negative tSC trough; bottom, cells with maximal firing after the most negative tSC trough. Red arrowheads mark the example neurons from panel (**b**). **d** Histograms of preferred tSC phases for all tSC-coupled MS neurons, sorted into four groups based on the tSC they are coupled to. **e** Left, histogram of the number of tSCs single MS neurons are coupled to. Right, number of neurons phase-coupled to each tSC. **f** Stacked bar chart showing the proportion of tSC-coupled neurons (empty bars) as a function of their theta-coupling (left) and rhythmic firing properties (right). Source data are provided as a Source Data file.

## MS neurons temporally lead hippocampal beta-gamma band activity

To determine the direction of possible causal relationships between MS and hippocampal activity, we investigated whether MS activity changes consistently anticipated or lagged correlated changes in hippocampal tSCs. For that, we quantified the coupling strength (Rayleigh's Z-value) between MS firing and hippocampal tSCs as a function of a temporal lag, similar to a crosscorrelation analysis. Peak coupling strength in the negative domain indicated that the MS neuron best coupled to the future of the hippocampal tSC, thus temporally led hippocampal population activity. We examined the distribution of the optimal lags that provided the strongest coupling, and the average of the normalized Z-value functions for tSC-coupled MS neurons (Fig. 4). We found that the majority of the tSC-coupled MS neurons preceded the corresponding hippocampal tSCs, indicated by (i) negative optimal lags in 58% of the neurons (tSC1, 65%; tSC2, 52%; tSC3, 55%; tSC4, 56%; median optimal lags, tSC1, −8 ms; tSC2, −1 ms; tSC3, −6 ms; tSC4, −9 ms), (ii) negative peak locations of the average Z-value functions (tSC1, −8 ms; tSC2, −4 ms; tSC3, −8 ms; tSC4, −9 ms), and (iii) larger area under the average Z-value function in the negative vs. the positive half-plane (the lags splitting the area under the curve to two equal halves: tSC1, −6 ms; tSC2, −5 ms; tSC3, −6 ms; tSC4, −7 ms). Furthermore, the vast majority (81%) of tSC-coupled constitutively bursting neurons were preceding tSCs (tSC1, 96%; tSC2, 71%; tSC3, 73%; tSC4, 71%; Supplementary Fig. 9), with a significantly larger lag compared to the rest of the coupled population ($p = 0.0024$, one-tailed Mann-Whitney U-test; the lags splitting the areas to equal halves: tSC1, −13 ms; tSC2, −11 ms; tSC3, −9 ms; tSC4, −5 ms). Theta-associated bursting MS neurons showed similar temporal properties for tSC1 and tSC2 (Supplementary Fig. 10, $p = 0.033$, one-tailed Mann-Whitney U-test; preceding neurons: tSC1, 83%; tSC2, 65%; lags splitting the areas to equal halves: tSC1, −11 ms; tSC2, −8 ms).

Revealing the optimal lag between spike timing of each tSC-coupled MS neuron and the concurrent tSCs allowed us to align MS spikes with the corresponding tSC cycles, which enabled further refinement of the correlation analysis between MS intra-burst spike timing and tSCs. We found systematic tSC cycle length variations, with cycles around the largest trough being slower than flanking cycles. This slowing of tSCs was reflected in longer time intervals between corresponding troughs and peaks of MS firing rate changes (Supplementary Fig. 11). These correlations between MS intra-burst spike timing and concurrent tSCs suggest that the MS might be closely integrated into the hippocampal gamma generation mechanisms.

## Differential coupling of anatomically identified MS cells to CA1 tSCs

Several kinds of neurons in the MS have been defined based on their combined firing patterns, neurochemical profile, and cortical projection targets[47,48,64] (Fig. 5a). Based on the differential engagement of CA1 inputs in theta-nested slow, mid and fast gamma oscillations[21,22,54,59], we reasoned that these defined MS cells may participate differently in CA1 tSCs of distinct frequencies, in correlation with their projections to CA1 input regions. To test this, we analyzed MS neurons recorded and labeled juxtacellularly in awake mice, with concurrent recordings of CA1 LFP, from three studies[47,48,64] (Fig. 5b).

GABAergic MS Teevra cells fire short-duration theta-rhythmic bursts, project mainly to the CA3 (some of them also to CA1 to a lesser degree), target PV+ axo-axonic and CCK+ interneurons, and are themselves PV-expressing[48]. We analyzed $n = 9$ Teevra cells from 9 mice and found that they mostly couple to beta/slow-gamma tSC1 ($n = 5/9$) and tSC2 ($n = 4/9$), while no coupling to faster tSCs was detected (Fig. 5c, d).

MS Orchid cells fire longer duration theta-bursts, send projections to the entorhinal cortex, the dorsal presubiculum and the retrosplenial cortex, target PV+ and nNOS+ interneurons, and are themselves PV-expressing[64]. We analyzed $n = 7$ Orchid cells from different mice and found that these neurons could couple to both low- and mid-gamma tSCs (Fig. 5c and Supplementary Fig. 5c, tSC1, $n = 7/7$; tSC2, $n = 6/7$; tSC3, $n = 4/6$; tSC4, $n = 3/6$; one neuron could not be tested for tSC3 and 4 due to insufficient data, see Methods).

Low rhythmic neurons (LRNs) form a group of MS cells that fire with a lower degree of rhythmicity, are suppressed during CA1 sharp-wave ripples and mostly lack detectable PV immunoreactivity, with common projections mainly to CA3 and the dentate gyrus[47]. By analyzing $n = 10$ LRNs from separate awake mice (Fig. 5c), we found that they showed coupling to the beta-band tSC1 ($n = 6/10$), and only occasional coupling to other tSCs (tSC2, $n = 3/10$; tSC3, $n = 0/10$; tSC4, $n = 1/8$; two neurons could not be tested for tSC4 due to insufficient data, see Methods). LRN neurons coupled to tSC1 or tSC2 but not to tSC4 were projecting to CA3 ($n = 3/3$, no projection data for two neurons). Interestingly, the only tSC4-coupled LRN neuron (MS104e) was a remarkable outlier sharing some properties of Orchid neurons: it was coupled to both tSC1 and tSC2, expressed PV, projected to the CA1, retrosplenial cortex, dorsal subiculum and dorsal presubiculum via the fornix and did not project to CA3 or dentate gyrus. The group of LRNs coupled only to tSC1 (cells MS16d, TV77q, TV78l) comprised the PV-negative Calbindin-negative subpopulation of LRNs[47].

Next, we visualized the distribution of preferred phases of these anatomically identified MS neurons relative to the different tSCs (Fig. 5d). We observed a non-uniform distribution of phase preference in case of tSC1 and tSC2, similar to our previous, unlabeled cell data (Fig. 3d), with different identified MS neuron types exhibiting tendencies of firing at distinct tSC phases (Fig. 5d bottom). We also tested whether Teevra, Orchid and LRN neurons preceded or followed CA1 tSC activity by repeating the time-shifted phase coupling analysis (Fig. 4) and found a clear temporal antecedence in case of the Orchid neurons (Fig. 5e and Supplementary Fig. 5c). Finally, we found that intra-burst frequencies of MS PV+ neurons showed a correlation with concurrent tSCs (Supplementary Fig. 12) based on average inter-spike interval histograms, similar to what we found for a subset of unidentified MS neurons (Supplementary Fig. 7).

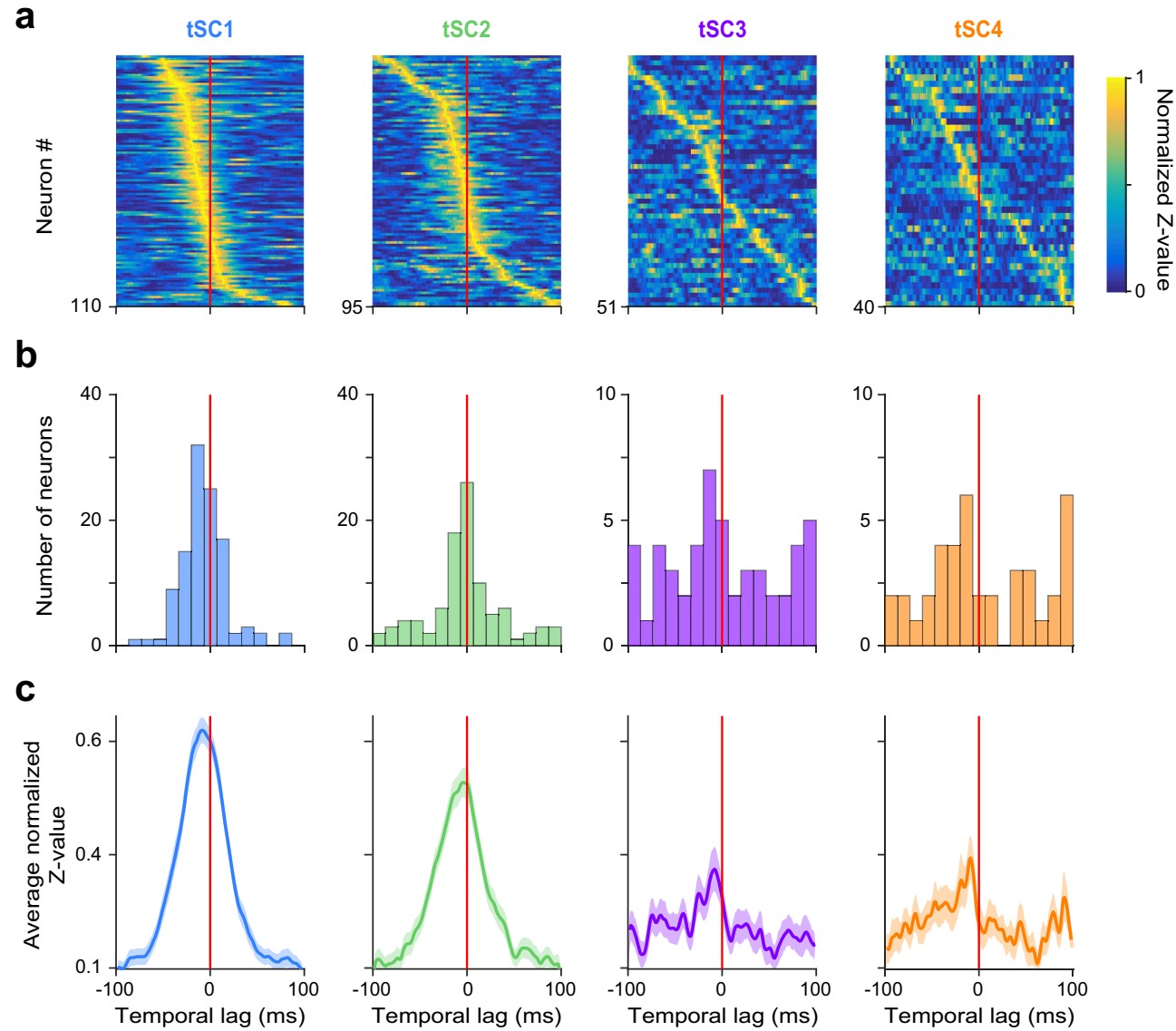

**Fig. 4 | Firing of most MS neurons predicts tSCs signals. a** Normalized Rayleigh's *Z*-value as a function of temporal lag between hippocampal tSCs and MS spike trains. All MS neurons coupled to the given tSC are shown. *Z*-values at negative lags quantify how well MS signals predict future tSC values. **b** Histograms showing the distribution of time lags across MS neurons that realize the maximal phase locking as quantified by the *Z*-values, separately for each tSC. **c** Average normalized *Z*-value of tSC-coupled MS neurons as a function of time lag. Error shade represents SEM. Source data are provided as a Source Data file.

Thus, two anatomically identified, theta-rhythmic PV-expressing MS GABAergic cell types exhibited strong correlations with CA1 tSCs, the CA3-projecting Teevra cells being locked to beta/low-gamma, and the entorhinal-projecting Orchid cells coupled to beta to mid-gamma tSCs, in line with the assumed role of their projection targets in the control of distinct gamma oscillations of the CA1.

### Optogenetic stimulation of PV-expressing MS neurons evokes tSC-like CA1 activities

The finding that MS firing anticipated hippocampal tSCs raised the possibility that the MS has a role in tSC expression. Putative theta pacemaker MS neurons, previously suggested to express PV[48,52,63,74], were most prominently predicting tSCs in the CA1 (Fig. 5e and Supplementary Fig. 9). In accordance, we also demonstrated that *n* = 10/14 identified PV-expressing MS neurons showed coupling to more than one tSC (*n* = 1/8 PV-negative neurons, Fig. 5c) and the PV-expressing Orchid cell group consistently anticipated hippocampal tSCs.

Therefore, we focused on PV-expressing MS neurons and tested whether they are capable of evoking theta-nested beta and gamma

oscillations. First, we performed in vitro whole-cell patch clamp recordings from acute hippocampal slices of PV-Cre mice (*n* = 6) injected with an AAV vector containing Cre-dependent ChR2-eYFP in the MS and confirmed a direct connection from MS PV+ neurons to PV+ and somatostatin-expressing (SOM+) interneurons in the dorsal CA1 and CA3 (Supplementary Fig. 13a-d). We performed photo-stimulation of MS PV+ fibers with theta-modulated bursts of laser light pulses (theta-modulated stimulation, tmS1-4), mimicking physiological tSC1-4 rhythms, and found that most hippocampal interneurons were capable of following the tmS1-3 burst stimulation patters with IPSCs, IPSPs and rebound spikes (Supplementary Fig. 13e–g).

Next, we performed acute optogenetic stimulation experiments in awake PV-IRES-Cre mice injected with AAV2/5.DIO.h-ChR2.eYFP (*n* = 7, see Supplementary Fig. 14 for histological track reconstruction and viral expression in PV cells) or with the AAV2/5.DIO.eYFP control virus (*n* = 3). We stimulated channelrhodopsin2-expressing MS PV neurons with tmS1-4 stimulation patterns, and recorded LFP and single unit activities from CA1 concurrently (Fig. 6a). We observed high amplitude stimulation-evoked tSC-like

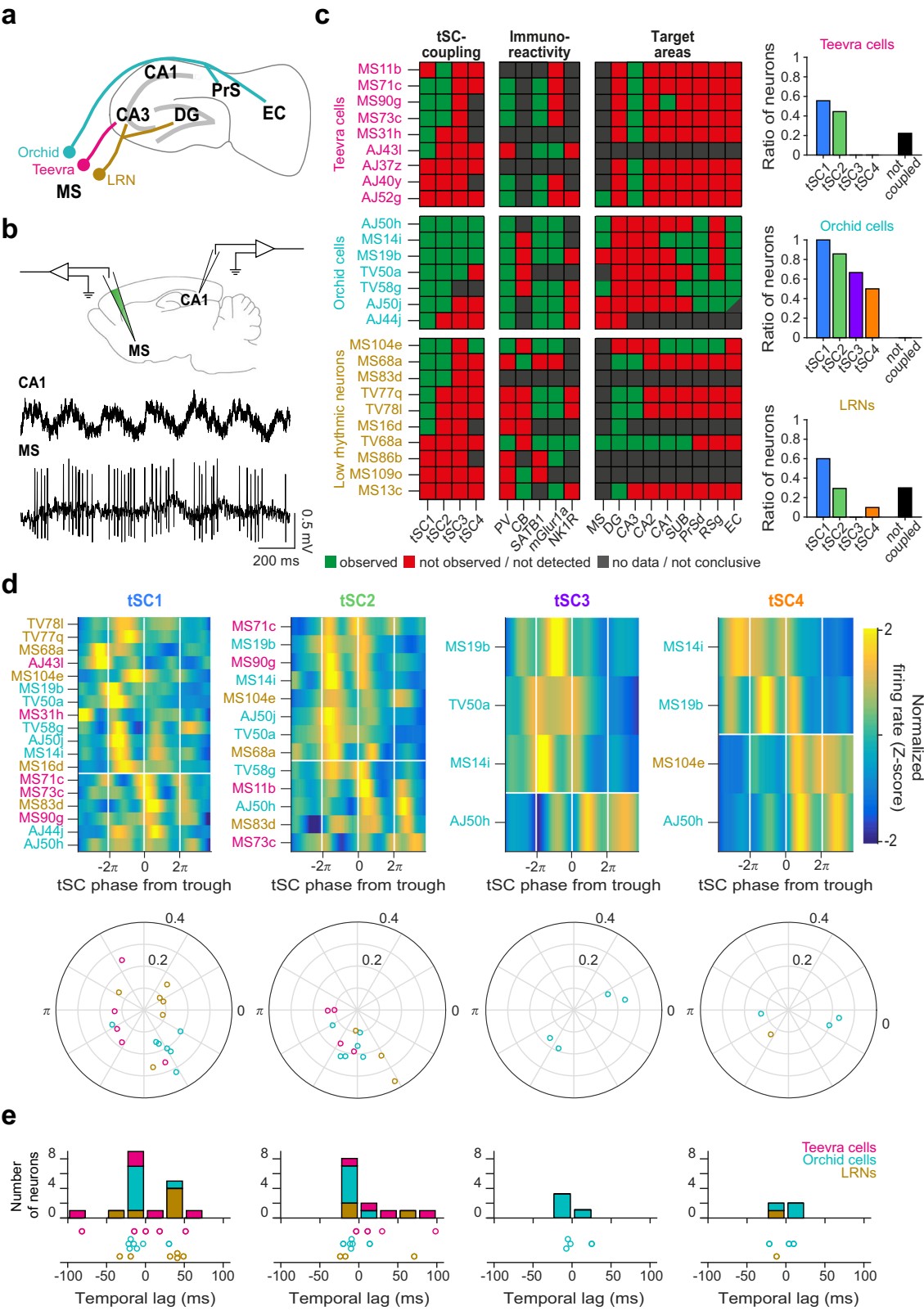

oscillations from CA1 in all channelrhodopsin2-expressing animals but not in control mice. These oscillations largely followed the stimulation frequency and resembled physiological tSCs in spectral content (Fig. 6a, b and Supplementary Fig. 15). We also found stimulus-evoked tSC-associated firing of single CA1 neurons, with some putative CA1 interneurons robustly following the stimulation frequency from tmS1 to tmS4 (Fig. 6c and Supplementary Fig. 16). In

sum, these experiments demonstrated that MS PV neurons are capable of evoking tSC-like activity in the CA1 network.

We next examined the laminar profiles of both spontaneous and stimulation-induced tSCs to examine their layer-specific sources. Therefore, we performed current-source density (CSD) analysis with electrode arrays spanning the full somato-dendritic axis of CA1 pyramidal neurons ($n = 4$ mice for spontaneous and $n = 4$ mice for

**Fig. 5 | Anatomically identified MS cells show differential coupling to CA1 tSCs. a** Main projection targets of Teevra, Orchid and low rhythmic (LRN) MS neurons (MS, medial septum; DG, dentate gyrus; PrS, presubiculum; EC, entorhinal cortex)[47, 48, 64]. **b** Top, schematic of the juxtacellular labeling and recording experiment with concurrent recordings from CA1 pyramidal layer in awake mice. Bottom, example of simultaneously recorded CA1 LFPs and MS spikes. **c** Left, tSC-coupling, immunoreactivity and projection targets of identified MS neurons. Cells were sorted based on the fastest tSC they were coupled to. Right, number of Teevra, Orchid and LRN neurons phase-coupled to each tSC. PV, parvalbumin; CB, calbindin; mGluR1a, metabotropic glutamate receptor 1a; NK1R, neurokinin 1 receptor; SUB, dorsal subiculum; PrSd, dorsal presubiculum; RSg, granular retrosplenial cortex. AJ50j is a putative EC-projecting neuron, as its main axon faded just rostral to caudo-dorsal EC. **d** Top, Z-scored phase histograms of all identified tSC-coupled MS neurons, sorted into four groups based on the tSC they are coupled to (blue, low firing rate; yellow, high firing rate). Zero phase corresponds to tSC troughs (white vertical lines). Cells within each group were sorted by their preferred phase in two blocks: top, cells with maximum firing rate before the most negative tSC trough; bottom, cells with maximal firing after the most negative tSC trough. Bottom, polar plot showing the phase preference (angle) and coupling strength (radius) of different MS neuron types for each tSC. **e** Distribution of time lags across different MS neuron types that realize the maximal phase locking as quantified by Rayleigh's Z-values, separately for each tSC. Source data are provided as a Source Data file.

stimulation-induced tSCs). We examined the frequency – theta phase relationship of the CSD signal, allowing us to specifically compare currents related to the different tSCs. In line with previous studies[15,22,55,58,59], spontaneous fast gamma signals (tSC5) were dominant in the pyramidal layer, while slower oscillations emerged from deeper layers (Fig. 6d and Supplementary Fig. 17). Spontaneous mid-gamma oscillations (tSC3 and tSC4) were characterized by prominent currents in stratum lacunosum moleculare, while tSC1 and tSC2 signals were most distinct from mid-gamma signals in stratum radiatum. Examining laminar profiles of the stimulation-induced theta cycles revealed similar spatial current source density distributions. The pyramidal layer CSD was dominated by fast gamma components regardless of the intra-burst frequency of the stimulation. Meanwhile, stimulation-evoked tSCs following the stimulation frequencies in the beta to mid-gamma range had strong currents in the deeper layers, most prominently in stratum lacunosum moleculare (Fig. 6e).

## GABAergic septo-hippocamal feedback suppresses tSC generation

The MS not only sends GABAergic projections to the hippocampus, but also receives inputs from hippocampal somatostatin-expressing (SOM+) GABAergic neurons forming a reciprocally connected inhibitory circuit[75–79]. To fully explore the role of this circuit in the generation of the supra-theta oscillations, we performed optogenetic suppression experiments using SwiChR in 4 of the freely moving SOM-IRES-Cre mice used in this study (Fig. 6f). Mice underwent bilateral injections of AAV5-Ef1a-DIO-SwiChRca-TS-EYFP in the dorsal CA1. To enable the inhibition of the hippocampo-septal projections, each shank of the septal silicon probes in these animals were equipped with an optical fiber. Four additional mice were injected with AAV5-Ef1a-DIO-EYFP control virus and implanted with 2 optical fibers bilaterally.

These experiments revealed that the proportion of theta cycles strongly expressing tSCs significantly increased during the inhibition of these feedback projections compared to control periods without stimulation (paired, two-sided Wilcoxon signed rank test, $p = 0.0015$, Fig. 6g; no effect observed in control animals, $p = 0.7422$, Supplementary Fig. 18a). These results suggest that this inhibitory feedback loop may play a role in suppressing the septo-hippocampal generator of supra-theta oscillations.

Next, we examined the impact of optogenetic suppression of SOM+ hippocampo-septal projections in the MS on tSC-coupled and non-coupled MS neurons. More tSC-coupled than non-coupled MS neurons showed a >10% firing rate increase during stimulation (tSC-coupled, 20%; non-coupled, 13%). These neurons were likely directly targeted by the hippocampo-septal pathway and thus disinhibited upon inhibition of the hippocampo-septal GABAergic fibers. We also found MS neurons that showed firing rate suppression, possibly via the disinhibited neurons. We found that less tSC-coupled MS neurons were suppressed than non-coupled neurons (tSC-coupled, 32%; non-coupled, 39%; Supplementary Fig. 18b).

We also tested the effect of 20 Hz optogenetic stimulation of SOM + fibers in the Fimbria of channelrhodopsin-expressing SOM-Cre mice ($n = 2$) on tSC-coupled and non-coupled MS neurons. Many MS neurons showed firing rate suppression, which might be due to direct targeting of these cells by the hippocampo-septal inhibitory pathway. Some MS neurons showed firing rate increase, possibly disinhibited through the suppressed group. This population was more numerous among the not tSC-coupled MS neurons compared to the tSC-coupled ones (tSC-coupled, 6%; non-coupled, 18%; Supplementary Fig. 18c).

We found a significantly higher proportion of tSC-coupled neurons among the MS neurons activated upon SwiChR-mediated inhibition of hippocampo-septal GABAergic fibers then among the suppressed ones (65% vs. 48%, chi-squared test, $p = 0.0322$). This difference was due to a larger percentage of 'follower' neurons (ones with positive lags in the Z-shift analysis, demonstrating activity changes following/lagging CA1 activity changes) in the activated group (Supplementary Fig. 18d). In accordance, during optogenetic stimulation of the hippocampo-septal pathway, we found the lowest proportion of tSC-coupled neurons among the activated group, which was significantly less compared to the non-reactive neurons (31% vs, 63%, chi-squared test, $p = 0.0428$, Supplementary Fig. 18d). These results may indicate that there might be more tSC-coupled neurons among MS neurons directly targeted by the hippocampo-septal feedback.

## MS neurons couple to tSCs in anesthetized mice and rats

To test whether these findings can be generalized across theta rhythms in awake and anesthetized rodents and across mice and rats[10,11], we attempted to discriminate different types of theta cycles from the LFP recorded from the CA1 radiatum layer of urethane-anesthetized rats ($n = 6$) and mice ($n = 5$) using a similar approach. The EEMD algorithm successfully detected tSCs in urethane-anesthetized rodents (Fig. 7a, b and Supplementary Fig. 19a, b). However, these were substantially less common than in awake mice (24% of all theta cycles both in rats and mice, Fig. 7c and Supplementary Fig. 19c). They were also described by slower characteristic frequencies, consistent with a general slowing of many oscillatory patterns during anesthesia including theta[80–83]. Additionally, they showed a weaker separation in theta phase across tSCs, arguing for potential mechanistic differences in oscillation genesis under anesthesia.

To test the potential contribution of the MS, we examined whether concurrently recorded MS neurons showed phase-coupled firing to tSCs in anesthetized rodents, too. Indeed, we found significant phase-locking in 33% and 35% of MS neurons recorded from urethane-anesthetized rats and mice, respectively (Rayleigh's test, $p < 0.05$, type I errors were controlled with Storey's false discovery rate method[73], Fig. 7c and Supplementary Fig. 19d). These data indicate that the MS network may contribute to theta-gamma coordination in the CA1 even in the absence of sensory inputs.

## Theta-nested spectral components are present in the ventral CA1

Finally, running EEMD analysis on a public dataset[84] revealed beta-gamma band tSCs in ventral hippocampus recordings (Supplementary Fig. 20). Distinct tSCs followed the same characteristic frequencies in

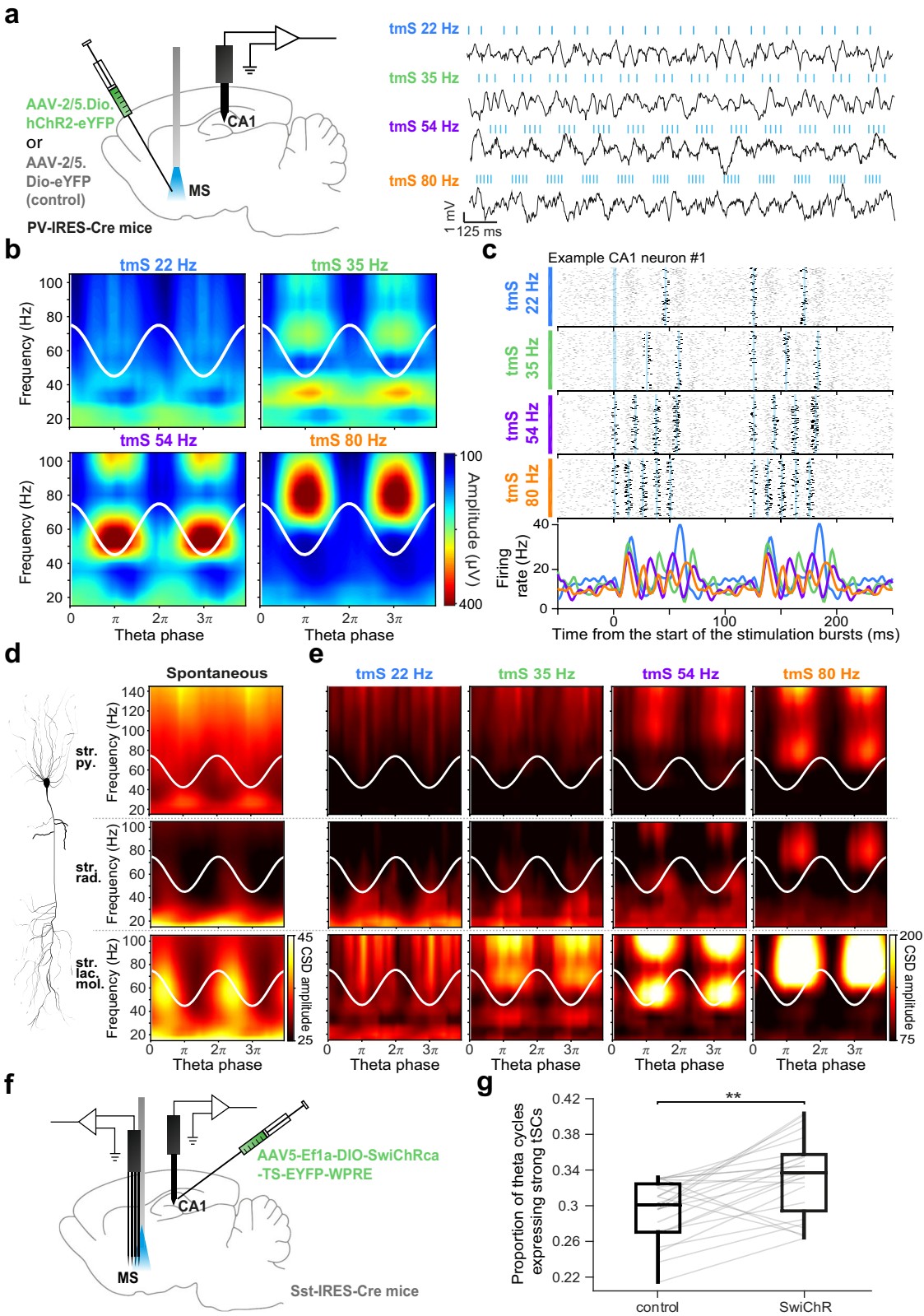

the beta to mid-gamma range as in the dorsal CA1, with largely similar theta-gamma phase-amplitude coupling. In contrast, tSC5 measured in the ventral CA1 was typically slower (with a peak around 80–90 Hz) and preferred the same phase as the mid-gamma components. Thus, while tSC5 in dorsal CA1 is thought to reflect local processing[19,55], ventral hippocampal tSC5 might rather reflect conceptually different network mechanisms, possibly controlled by external inputs similarly

to mid-gamma components. Observing dorso-ventral similarities in tSCs1-4 may reflect similar oscillation mechanisms across the CA1.

## Discussion

Hippocampal network activities of different time scales are not independent but interact in specific ways. The momentary phase of the 4–12 Hz theta oscillation modulates the fluctuating spectral power of

**Fig. 6 | Optogenetic stimulation of PV-expressing MS neurons evokes tSC-like activity patterns in the CA1. a** Left, schematic of the acute experiment with optogenetic stimulation of PV-expressing MS neurons in awake mice. Right, raw LFPs recorded from the CA1 radiatum layer (black) during photostimulation of MS PV-expressing neurons. Blue ticks mark 2 ms laser pulses. Stimulation was performed in theta-modulated bursts of laser pulses (tmS) mimicking the different frequencies of physiological tSCs (22 Hz, 35 Hz, 54 Hz, 80 Hz). **b** Mean amplitude of supra-theta spectral components as a function of theta phase during the different tmS protocols. Two theta cycles are shown, indicated by the white cosine curves. **c** Spike rasters and peri-stimulus time histograms of a putative CA1 interneuron aligned to the onset of each tmS sorted by stimulation frequency (blue lines, photostimulation). Note the brief initial suppression of spiking upon photostimulation before the firing rate increase (see also Supplementary Fig. 16). **d** Laminar profile of currents related to spontaneous tSCs. Left, schematic of a pyramidal cell showing CA1 layers (str. py., stratum pyramidale; str. rad., stratum radiatum; str. lac. mol., stratum lacunosum moleculare). Stoyo, Karamihalev.

(2020). CA1 pyramidal neuron. Zenodo. https://doi.org/10.5281/zenodo.4312494. Right, mean amplitude of supra-theta CSD signals as a function of theta phase from an example session of a freely moving mouse (see also Supplementary Fig. 17 for the same analysis on cycles strongly expressing a particular tSC). Note that the upper limit of the y-axis is extended in the stratum pyramidale panel to visualize fast gamma-bands. White cosine curves, two theta cycles are shown. **e** Mean amplitude of supra-theta CSD signals as a function of theta phase during different tmS protocols, showing the laminar profile of currents related to photostimulation-evoked tSCs. White cosine curves indicate the theta phase defined from the radiatum layer LFP. **f** Schematic of the experiment using SwiChR injected into the CA1 and an optic fiber implanted into the MS. **g** Proportion of theta cycles expressing tSCs during inhibition and control periods. Data points belonging to the same recording session ($n = 23$) are connected with gray lines. Box-whisker plots show median, interquartile range and non-outlier range. $^{**}p = 0.0015$, two-sided Wilcoxon signed-rank test. Source data are provided as a Source Data file.

faster, beta/gamma band oscillations[85–88]. This so-called cross-frequency coupling has been shown to be important for cognitive functions such as working memory[89,90]. Medial septal GABAergic neurons have a central role in orchestrating theta oscillations; however, whether they also participate in regulating faster, supra-theta oscillations of the hippocampus is unclear. While phase coupling to entorhinal or hippocampal gamma oscillations has been demonstrated for some MS cell types[47,64], hippocampal gamma oscillations are typically interpreted without including the MS in the gamma generating network[17,53]. We challenge this view by demonstrating that MS firing correlates with, predicts changes of and is capable of generating theta-nested beta/gamma band oscillations in the CA1.

Early accounts of CA1 oscillatory activity revealed cross-frequency coupling between theta and gamma oscillations[91–93], giving rise to the concept of theta-nested gamma bouts[94–96]. A landmark discovery of functionally distinct CA1 oscillatory patterns within the gamma band suggested the parallel presence of multiple fast-scale network mechanisms organized by theta phase[21,22,54]. Recent studies revealed that individual theta cycles can be reliably categorized by their beta/gamma content, and these distinct theta-nested spectral components also segregate at the network and functional levels[15,97]. This strengthens the prevailing view that theta cycles can be considered as individual processing units that organize spatial processing[7,98–100], episodic memory encoding and retrieval[70,101], by temporal arrangement of information routing from distinct sources[9,22,54].

Medial septal GABAergic neurons, particularly their parvalbumin-expressing subpopulation, have been implicated in the genesis of CA1 theta oscillations. MS PV neurons project to the hippocampal formation and innervate GABAergic interneurons[24,74,102–105]. They were proposed to pace the theta rhythm through providing rhythmic disinhibition of pyramidal neurons via their interneuron targets[49,51,63,67,106,107], probably as part of a wider theta synchronization network that includes other MS populations[47,48,50,52,64,108,109], the supramammillary nucleus[39–41], the nucleus incertus[42–46,110] and the raphe nuclei[111,112]. Additionally, previous studies suggested the presence of higher frequency spectral elements in the MS[47,64], which we also confirmed (Fig. 1b, c). However, while the involvement of cholinergic mechanisms in theta-gamma coupling has been demonstrated[33], the role of the MS GABAergic projection in controlling hippocampal gamma rhythms is less explored[17]. Since basal forebrain PV neurons that project to the neocortex can control gamma rhythms in the frontal cortices[113,114], we hypothesized that MS PV neurons projecting to the hippocampus may exert a similar influence on CA1 supra-theta oscillations.

First, we confirmed the presence of tSC components in the CA1 of mice moving freely on a linear track, which appeared in coherence with MS bursts with beta-gamma band intra-burst spike frequencies[61–63]. In line with former results[15], we reliably detected five distinct theta-nested

spectral components well-separated in frequency and preferred theta phase. One or more of these tSCs were strongly expressed in nearly one third of all theta cycles. The identified tSCs exhibited different characteristic frequencies spanning the beta to fast gamma range, resembling supra-theta oscillations reported in previous studies[15,21,53,54,58,59]. For instance, tSC2 matched the frequency of slow gamma oscillations, thought to promote memory retrieval driven by CA3[19,54,57], whereas tSC3 and tSC4 were characterized by mid-gamma frequencies suggested to originate from the medial entorhinal cortex[57]. We confirmed that the strength of most tSCs (tSC2-tSC5) positively correlated with animal speed[15,66], while interestingly, tSC1 showed a significant negative correlation. Faster locomotion is accompanied by higher theta frequencies; this might be related to place coding, which is probably a strong organizing factor while collecting episodic memories during movement[115]. Theta frequency was gradually increasing with the frequency of the spectral component nested in the theta cycle for tSC1-tSC4, likely reflecting that mid-gamma components are related to memory encoding processes[17,54]. Interestingly, theta frequency during theta cycles expressing tSC5 components, which are thought to reflect internal processing in CA1, did not follow this trend. We found an over-representation of tSC1, corresponding to beta band oscillations, in short theta epochs and during non-theta-to-theta transitions. In line with this, a recent study found that theta-nested hippocampal beta oscillations rapidly increase at the onset of exploratory behavior, after which they gradually adapt and fade, not explained by locomotion alone. The presence of these beta bouts predicted subsequent performance in an object location test[116]. Other studies found that hippocampal beta oscillations are related to odor sampling[117] and originate from the olfactory bulb[118]; thus, these results together suggest that tSC1-dominated theta cycles may be hallmarks of olfactory exploration and reflect a transient increase in the synchrony between the olfactory system and the hippocampus. We further propose that tSC1-dominated cycles interspersed in longer theta episodes may also reflect increased odor processing subserving exploration, planning and memory, providing a window on the information routing process of the hippocampal network.

Most MS neurons showed higher firing rate and stronger theta phase coupling during theta cycles in which tSC4 was present, corresponding to mid-gamma oscillations that have been associated with entorhinal inputs[22]. More place cells phase-lock to this gamma band than to other supra-theta oscillations[54], and this activity pattern might be particularly important for memory encoding and long-term potentiation. In accordance, Lopes-dos-Santos et al.[15] found that the tSC4 component strongly increased and predicted subsequent performance during learning in a crossword maze[15]. Our results indicate that the MS is strongly involved in the regulation of these learning-related theta cycles, reflected in higher activity levels and stronger

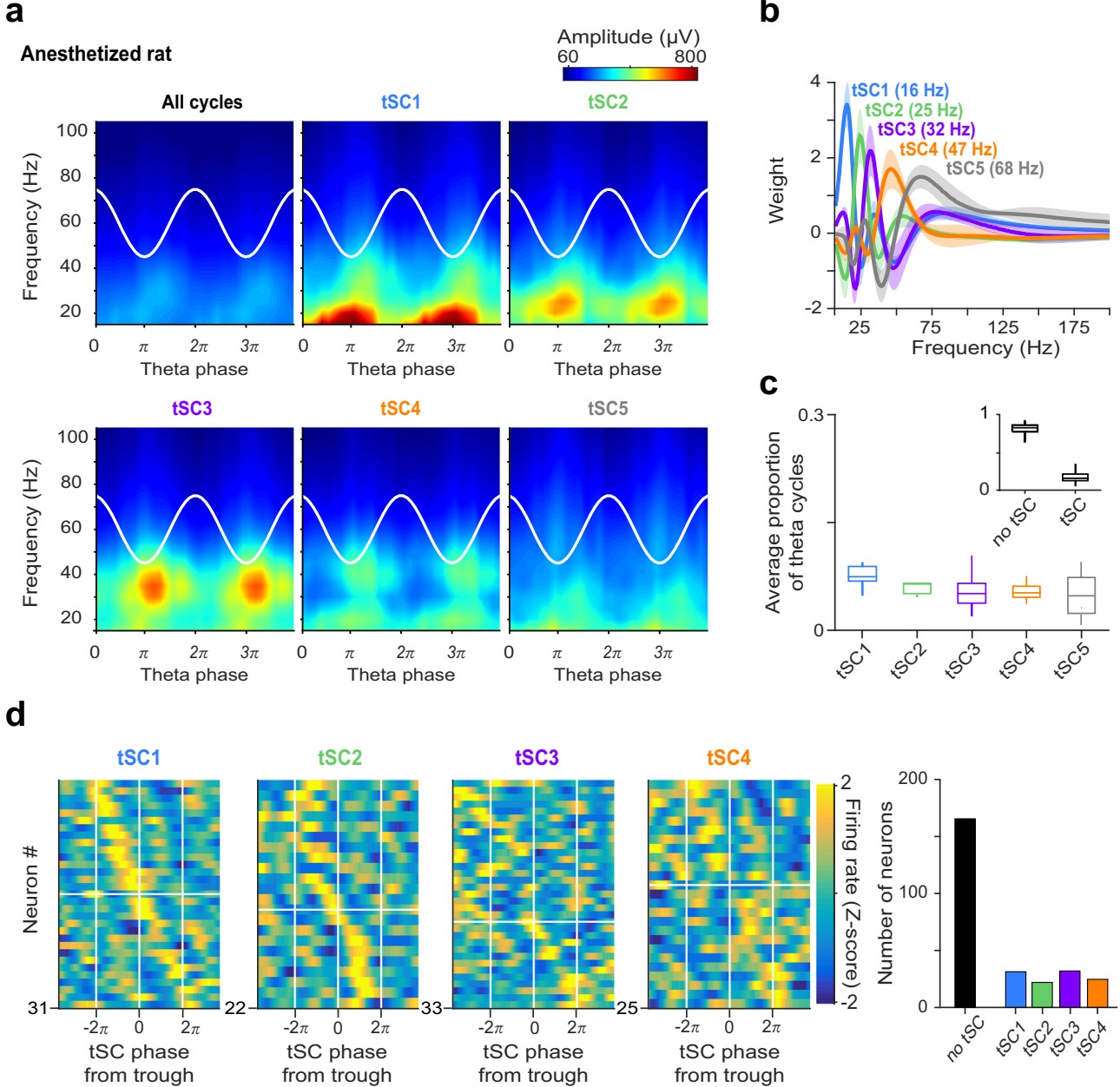

**Fig. 7 | MS neurons show phase coupling to hippocampal tSCs in anesthetized rats. a** Mean amplitude of supra-theta spectral components as a function of theta phase was computed from the raw LFP of an example session of an anesthetized rat, for all cycles (top left) and for cycles strongly expressing a given tSC. Two theta cycles are shown, indicated by the white cosine curves. **b** Average tSCs spectra in anesthetized rats (n = 6). Error shades show the standard error of the mean. Median peak frequencies for all recordings are shown in the brackets. **c** Proportion of theta cycles expressing each tSC (n = 13 sessions). Boxes and whiskers show median, interquartile range and non-outlier range. Insets show the proportion of theta cycle expressing any of the tSCs. **d** Left, Z-scored phase histograms of all tSC-coupled MS neurons, sorted into four groups based on the tSC they are coupled to (blue, low firing rate; yellow, high firing rate). Zero phase corresponds to tSC troughs (white vertical lines). Cells within each group are sorted by their preferred phase in two blocks: top, cells with maximum firing rate before the most negative tSC trough; bottom, cells with maximal firing after the most negative tSC trough. Right, number of neurons phase-coupled to a given tSC. Source data are provided as a Source Data file.

phase-locking. This is in accordance with a long history of lesion, pharmacology, stimulation and recording studies that identified the MS as a crucial subcortical structure in declarative memory formation[37,45,115,119–131]. Human patients with MS lesions, sometimes caused by jet bleeding after arteria communicans anterior aneurysm rupture, present the classic Korsakoff triad of anterograde and retrograde amnesia and confabulation[132,133].

tSC5 attributed large weights to frequencies in the fast gamma band, which are thought to be generated by local CA1 networks of pyramidal cells and interneurons[19,55]. Also in line with previous

findings[15,22,58,59], current source density analysis confirmed that tSC5 signals were most prominent in the pyramidal layer, while slower oscillations emerged from the deeper layers: tSC3 and tSC4 showed strong currents in the lacunosum moleculare layer, while sources of tSC1 and tSC2 signals were more localized to the stratum radiatum than mid-gamma tSCs.

Although the strong phase-coupling of MS neurons to hippocampal theta is well established[49,50,63,67,106], correlations between MS firing and hippocampal gamma oscillations are much less explored[47,64]. We found that a large proportion of MS neurons were phase-coupled

to beta/gamma band tSCs. More neurons coupled to slower tSCs compared to fast ones, although this might partially be the consequence of our analysis being more exposed to noise in case of shorter tSC cycles. MS neurons coupled to tSC1 and tSC2 showed a bias towards the ascending slope of the tSCs. A more in-depth analysis of MS intra-burst spike timing revealed a theta-phase-independent, cycle-by-cycle correlation between MS spike timing and concurrent tSCs, arguing for a close integration of the MS into CA1 gamma generation mechanisms (Supplementary Figs. 6, 7, 11, 12).

In the case of two correlated time series, small changes in either of the variables may predict time-locked changes in the other, reflecting cause-effect relations. For instance, sharp low-latency peaks in spike crosscorrelations have been taken as indirect evidence for functional monosynaptic connections[2,134–137]. Analogously, changes in a point process like a series of spike times may lead to corresponding changes in correlated network oscillations, detectable by a time-varying measure of coupling strength, expressed as a function of time lag between the two variables[49,72]. This has been used to demonstrate directional coupling between the hippocampus and prefrontal cortex[72]. By using the same approach, we found that MS spiking predicts and thus temporally leads CA1 tSCs. The MS population showed negative median time lags, and thus temporal antecedence, by either testing average phase coupling strength (−6 ms) or optimal correlations in individual neurons (−6 ms), despite large variability across tSC-coupled neurons in their optimal time lags (interquartile range: 34 ms). This unexpected result points to a role of the MS in controlling hippocampal network activity that goes beyond its known function in pacing theta and suggests that it is also an important node in the coordination of hippocampal supra-theta oscillations.

A subpopulation of MS neurons exhibit theta-rhythmic burst firing regardless of whether theta oscillation is present in the hippocampus. These constitutive theta-bursting neurons are often considered putative pacemakers of hippocampal theta oscillations, although they may be part of a distributed theta-rhythmic oscillatory network[39,40,49–51,63,110]. Another set of MS neurons only exhibits theta-rhythmic firing when the hippocampus is also in theta state, referred to as theta-associated bursting MS neurons[52]. Although we found that constitutive and theta-associated bursting MS neurons showed similar tendencies to the entire MS population in their relation to supra-theta CA1 oscillations, we also uncovered a number of differences. A larger proportion of the constitutive bursting neurons were coupled to tSCs (78% vs 48% of other MS neurons), and often to more than one component (68%). Constitutive bursting neurons provided a significant part of tSC-anticipating MS neurons, as the vast majority (81%) of their tSC-coupled population were characterized with negative optimal time lags (median: −14 ms), suggesting a key importance of these neurons in the control of tSCs. The theta-associated bursting neurons were also overrepresented among tSC-coupled neurons but to a lesser extent (59%). They also anticipated tSCs (median optimal lag: −14 ms) significantly more than the rest of the population when only the slower tSC1 and tSC2 components were considered.

We tested whether the MS GABAergic network is capable of inducing hippocampal tSCs by optogenetic stimulation; thus, whether the MS is sufficient for tSC generation. We targeted PV-expressing GABAergic MS neurons based on their known projections and roles in CA1 oscillatory control[47,48,63,64,74] and previous reports on theta evoked by PV stimulation[25,30]. We found that stimulating PV-expressing MS neurons by theta-modulated beta/gamma bursts of laser light that matched spontaneous tSCs in frequency induced tSC-like oscillations in the CA1 that resembled spontaneous tSCs in their average spectral content and laminar distribution. However, stimulus-evoked gamma components (mostly in the mid-gamma range) differed from spontaneous tSCs in their amplitude modulation by the hippocampal theta phase, possibly due the stimulation-induced artificial synchrony of the MS PV

network, which could not reproduce their well-known physiological diversity (Supplementary Fig. 9, see also refs. 48,50,52,63).

In line with these results, a previous study found that septal infusion of the GABA$_A$ receptor agonist muscimol decoupled theta and gamma rhythms and disrupted memory retrieval, while electrical stimulation of the fimbria-fornix with theta-modulated high frequency (500 Hz) stimulation increased theta-gamma co-modulation and partially rescued memory performance[138]. Inactivation or lesion of MS GABAergic neurons reduced hippocampal gamma oscillations induced by NMDA antagonists[139]. Functionally intact inhibition onto hippocampal PV interneurons has been shown to be necessary for physiological theta-gamma coupling[140] and 40-Hz stimulation of MS PV neurons could rescue impaired theta-gamma coupling in a mouse model of Alzheimer's Disease[141]. Nevertheless, whether MS PV neurons are capable of producing the physiological spectrum of theta-nested spectral components of the CA1 has not been addressed before.

Urethane-anesthetized rodents have been used extensively as models of septo-hippocampal theta-frequency synchronization. However, much less is known about hippocampal supra-theta oscillations during anesthesia. Similar to a previous study[22] that described theta-coupled gamma oscillations in the CA1 that used urethane-anesthetized rats supplemented with ketamine-xylazine, we observed tSCs during urethane anesthesia in both mice and rats; however, tSC expression was diminished compared to awake mice. These tSCs were characterized by slower frequencies and uniform theta phase preference. Despite these differences, a smaller proportion of MS neurons exhibited similar phase coupling to these tSCs as in freely moving mice, strengthening the argument that the MS has a general role in hippocampal oscillation genesis.

By what mechanism does the MS contribute to CA1 gamma formation? It seems reasonable to assume that part of the influence is direct: gamma-locked MS GABAergic neurons innervate gamma-locked CA1 interneurons, likely contributing to CA1 gamma oscillations (Fig. 8, pathway (i)). Since septo-hippocampal projections likely innervate multiple types of CA1 interneurons including PV-expressing basket and axo-axonic cells[24,48,103,142,143], gamma-rhythmic septal drive should necessarily contribute to their known gamma-locked discharge[19,56]. Indeed, within the hippocampus, bistratified cells were found to express the strongest phase coupling to gamma[19,144], and were shown to receive selective GABAergic MS input[102].

However, part or even most of the septal influence on tSCs may be indirect via the input regions to CA1. For instance, long-range inhibitory MS neurons innervate the EC[64,103,145–151], thus having the potential to entrain the EC sources of CA1 gamma. More specifically, it has been shown recently that MS Orchid cells project to the EC (Fig. 8, pathway (ii)) and phase-lock to EC mid-gamma oscillations, lending strong support to this hypothesis[64]. To directly test this, we analyzed anatomically identified Orchid, Teevra and low rhythmic neurons projecting to distinct parts of the hippocampal formation[47,48,64]. Teevra cells, which are rhythmically bursting PV-expressing GABAergic MS neurons that mostly project to the CA3[48], were coupled to beta/low-gamma tSC1 and tSC2, in accordance with the assumed role of CA3 in controlling low-gamma oscillations of CA1[22,54] (Fig. 8, pathway (iii)). Orchid cells, also PV-expressing and firing long theta-rhythmic bursts[64], were the only tested neurons which locked to mid-gamma tSC3 and tSC4, confirming the above-mentioned hypothesis of their potential role in controlling the mid-gamma band in CA1 via the entorhinal cortex (Fig. 8, pathway (ii)). Since Orchid cells were found to branch into the dorsal presubiculum[64], which also projects to their termination zone in the entorhinal cortex[152], the potential role of a MS-presubiculum-entorhinal cortical route should also be considered. Importantly, Orchid neurons not only coupled to but also predicted CA1 tSCs. In contrast, the non-PV LRNs only showed consistent coupling to the beta band tSC1. Since these cells innervate interneurons in the dentate gyrus, they may contribute to beta/slow-gamma oscillations observed

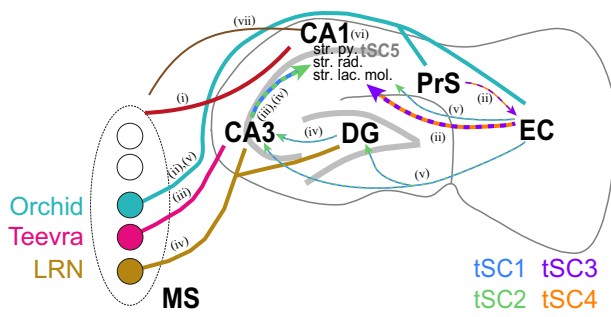

**Fig. 8 | Hypothetical schematic of direct and indirect septo-hippocampal pathways mediating tSCs in the CA1.** In this schematic, we depict the direct and indirect pathways between the MS and the CA1 implicated in controlling distinct CA1 supra-theta oscillations. Color codes for MS projections, including anatomically identified MS neurons, are shown on the left. Cortico-hippocampal and intrahippocampal pathways are color-coded by their dominant tSCs: tSC1, blue; tSC2, green; tSC3, purple; tSC3, green; tSC4, orange; tSC5, gray. **i** We suggest that direct GABAergic MS-CA1 projections (dark red curve) contribute to tSCs1-4. **ii** Orchid cells (teal) are likely important determinants of tSC3-4 in CA1 via their projections to PrS and EC. **iii** Teevra neurons (magenta) contribute to tSC1-2 through the CA3. **iv** LRNs (mustard) also form part of the tSC1-2 network via their projections to CA3 and DG. **v** Further indirect MS-EC-CA3/DG-CA1 pathways probably also contribute to the control of tSC1-2 components in the CA1. **vi** Local processing in the CA1 strongly contributes to fast gamma (tSC5) oscillations. **vii** Hippocampo-septal feedback by somatostatin-expressing GABAergic neurons (brown) could control tSC generation by negative feedback. MS, medial septum; EC, entorhinal cortex; PrS, dorsal presubiculum; DG, dentate gyrus; LRN, Low-rhythmic neurons; str. py., stratum pyramidale; str. rad., stratum radiatum; str. lac. mol., stratum lacunosum moleculare.

in the DG[59] that might in turn influence CA3 slow-gamma rhythms[153] (Fig. 8, pathway (iv)). However, CA1 has been shown to retain slow gamma activities in the absence of CA3 inputs, arguing for alternative sources of these rhythms in the CA1[154], which may include direct MS inputs to CA1 or the MS-EC pathway[25] (Fig. 8, pathway (v)). Also, as for hippocampal theta, intra-hippocampal gamma oscillators may play a role, especially for tSC5 (Fig. 8, pathway (vi)).

We found that the activity of most MS neurons predicted corresponding CA1 tSCs; however, a subpopulation of MS cells, including non-PV-expressing LRNs rather followed CA1 activity. This raised a potential role of the hippocampo-septal feedback in supra-theta hippocampal oscillations[75–79] (Fig. 8, pathway (vii)). By optogenetic suppression of somatostatin-expressing hippocampo-septal GABAergic fibers, we found that this inhibitory feedback projection suppresses tSC generation in the MS, probably through a biased targeting of tSC-coupled MS neurons. While PV-expressing MS neurons, including Orchid, Teevra, and part of Low Rhythmic Neurons are likely all GABAergic[63,74], Calbindin-expressing PV-negative LRNs may be, at least in part, excitatory glutamatergic neurons. Two of the 3 Calbindin-positive LRNs were not coupled to any of the tSCs, which may further emphasize the role of the GABAergic MS projection in contrast to glutamatergic MS projection neurons[26]. In summary, we propose that the septo-hippocampal GABAergic projection participates in CA1 beta/gamma genesis and the well-established cross-frequency coordination of theta and supra-theta oscillations, known to be crucial for spatial and working memory[20,92,138,155].

Our results also suggest updating the potential role of MS in hippocampal oscillopathies. Specifically, the MS was proposed to play important roles in Alzheimer's disease, schizophrenia, anxiety and pain, and that appropriate MS stimulation strategies may alleviate the negative consequences of these conditions[156,157]. Treatment approaches of memory impairments related to the MS, such as MS cholinergic stimulation and vagus nerve stimulation were also discussed[158]. Griguoli and Pimpinella stressed the importance of the MS in social

learning, suggesting that psychopathologies accompanied by the disruption of social life may be another condition in which MS-mediated oscillogenesis is impaired[159]. We expect that exploring mouse models of these pathological conditions and test possible alterations of the MS orchestrating CA1 neurons, focusing on a broad range of frequencies from theta to mid-gamma, will be exciting new directions in the future.

## Methods
### Experimental model and subject details
We used adult SOM-IRES-Cre mice ($n = 10$; all males; weight: 28–30 g) for the chronic freely moving experiments, adult wild type mice ($n = 11$; 7 males; 6 mice excluded due to <360 detected theta cycles; 22–30 g) for acute mouse recordings, adult wild type Wistar rats ($n = 7$; all males; one rat excluded due to <360 theta cycles; 200–400 g) for acute rat recordings and adult PV-IRES-Cre mice ($n = 19$; 6 males; 22–30 g) for slice electrophysiology, acute awake optogenetic stimulation experiments and verification of the ChR2 expression in PV-expressing neurons. All experiments were approved by the Animal Care and Use Committee of the Institute of Experimental Medicine or the Animal Care Committee of the Research Centre for Natural Sciences and the Committee for Scientific Ethics of Animal Research of the National Food Chain Safety Office and were performed according to the 2010/63/EU Directive of the EC Council.

Animals were kept in temperature- and humidity-controlled (21 ± 1 °C and 50–60%, respectively) housing facilities with a standard 12-h light–dark cycle, with food and water available ad libitum. Mice were housed in 36 × 20 × 15-cm cages. Implanted animals were housed individually. Additionally, we analyzed the firing of previously published juxtacellularly labeled neurons[47,48,64] recorded from head-fixed C57Bl7/J mice ($n = 30$; all males; 24–37.5 g) spontaneously running and resting on a circular treadmill or disc and ventral hippocampal LFP recordings from mice in an elevated plus maze downloaded from a public repository (https://doi.org/10.7272/Q6ZP44B9)[84].

Hippocampal oscillations were examined in both sexes under awake conditions and we did not observe sex-related differences in the theta-gamma phase-amplitude coupling. Chronic electrophysiology recordings were carried out in male mice, because males tolerate chronic implants better due to weight considerations. Nevertheless, optogenetic stimulation of parvalbumin-expressing MS neurons evoked hippocampal theta-nested oscillations in female mice in line with our findings from chronic recordings performed in male mice, thus suggesting that the MS has a similar role in hippocampal oscillation genesis in both sexes. Due to these study design complexities and sample size considerations, we refrained from post-hoc sex-based analyses.

### Surgical procedures
Surgeries were performed under general anesthesia. For virus injection surgeries, mice were anesthetized with an intraperitoneal injection of ketamine-xylazine mixture (dose by body weight, 167 and 7 mg/kg, respectively). Mouse chronic electrode implantation surgeries were performed under isoflurane anesthesia induced by ketamine-xylazine mixture (4:1) diluted 6x in Ringer's lactate solution (intraperitoneal injection; dose by body weight, 0.01 mg/g). In acute anesthetized experiments, rats and mice were anesthetized with an intraperitoneal injection of urethane (dose by body weight, 1.48 mg/g and 1.3 mg/g for rats and mice, respectively). The body temperature of the animals was maintained during the experiments with a heating pad. Animals were placed in a stereotaxic frame (David Kopf Instruments, Tujunga, US), the scalp was shaved, disinfected (povidone-iodine) and the skin and connective tissue of the scalp was infused by local anesthetic (Lidocaine s.c.). The eyes of the animals were protected with eye ointment (Laboratories Thea, Clermont-Ferrand, France). The skin was opened, the skull was cleaned, and the head of the animal was levelled using Bregma, Lambda and a pair of lateral points equidistant from the

**Table 1 | Implant types and target coordinates in the five experiments**

| Experiment | CA1 implant | CA1 coordinates | MS implant | MS coordinates | Ground location |
|---|---|---|---|---|---|
| Urethane anesthetized rat | linear silicon probe (A1x32-6mm-50-177; NeuroNexus Technologies, Ann Arbor, US) | AP: −4.5 mm ML: 3 mm angle: 0° | Buzsáki-type silicon probe (Buzsaki32; NeuroNexus Technologies, Ann Arbor, US) | AP: 0.4 mm ML: 1.6 mm angle: 15° | nuchal muscles |
| Urethane anesthetized mice | linear silicon probe (A1x32-6mm-50-177) | AP: −2.2 mm ML: 1.5 mm angle: 0° | single shank custom silicon probe with 32 × 4 square-shaped recording sites[184] | AP: 0.6 mm ML: 0.5 mm angle: 8° | neck muscle |
| Freely moving mice | linear silicon probe (A1x32-6mm-50-177) or Buzsáki-type silicon probe (Buzsaki32) | AP: −2.5 mm ML: 2 mm angle: 0° | Buzsáki-type silicon probe (Buzsaki32) | AP: 0.9 mm ML: 0.9 mm angle: 12° | cerebellum |
| Awake head restrained mice for optogenetic stimulation | 128 channels UCLA silicon microprobes (128 J or 128 A, Masmanidis lab) in both hippocampi | AP: −2.5 mm ML: 2.5 mm angle: 0° | optic fiber (200 μm core diameter, Thorlabs GmbH) | AP: 0.9 mm ML: 0 mm angle: 0° | cerebellum |
| Awake head restrained mice for juxtacellular recording | glass electrode filled with 0–3.0% neurobiotin (wt/vol) in 0.5 M NaCl | AP: −2.5 mm ML: 1.7 angle: 10° | glass electrode filled with 2.5–3.0% neurobiotin (wt/vol) or BDA in 0.5 M NaCl | AP: 0.85 mm ML: 0 mm angle: 0° | cerebellum |

Target coordinates were defined based on[160, 168]. Coordinates in the table refer to the coordinates of the craniotomy. AP, antero-posterior; ML, medio-lateral.

sagittal suture. Before implantation, red DiI (Thermo Fisher Scientific, Waltham, US) was applied on the probes to aid later histological reconstruction. Stereotaxic coordinates[160] were used to determine the proper antero-posterior (AP) and medio-lateral (ML) positions, craniotomies were opened above the medial septum and the hippocampus, and silicon probe or optic fiber implants were lowered above the dorsal boundary of the MS and into the dorsal CA1. The MS was implanted at an angle with respect to the dorso-ventral direction in the coronal plane to avoid the sinus sagittalis superior. Finally, the ground/reference electrodes were inserted (see Table 1 for implant types, coordinates and the location of the ground/reference electrodes for the different experiments).

In case of the chronic implants, adhesive agents (OptiBond, Kerr, Brea, US) were applied on the cleaned surface of the cranium. The craniotomies were sealed with artificial dura (Cambridge NeuroTech Ltd, Cambridge, UK). Septal and hippocampal probes (Table 1) were mounted on custom-made, adjustable microdrives, and the probe-microdrive assemblies were shielded by copper mesh, preventing the recordings from contamination by environmental electric noise. The mesh was covered by dental acrylic. Before finishing the surgery, Buprenorphine (dose, 0.045 μg/g body weight) was injected subcutaneously. Recordings were started after a one-week-long post-surgery recovery and habituation to connectorization.

Mice involved in the freely moving experiments underwent bilateral injection of AAV2/5-Ef1a-DIO-ChR2-YFP-WPRE or AAV5-Ef1a-DIO-SwiChRca-TS-EYFP or AAV5-Ef1a-DIO-EYFP into the dorsal hippocampi. In case of SwiChR-transduced animals, each four shank of the septal probe was equipped with an optical fiber (50 μm core diameter, 0.22 NA) with tip positioned 75-100 micrometers above the uppermost recording site and glued by optical adhesive. In case of ChR2-transduced animals, one optical fiber (105 μm core diameter, 0.22 NA) was implanted to illuminate the fimbria, independently from the septal silicon probe. Control periods (without stimulation) were analyzed in Figs. 1–4. Stimulation periods were analyzed in Fig. 6f-g and Supplementary Fig. 18. In case of EYFP control transduced animals, two optical fibers (105 μm core diameter, 0.22 NA) were implanted bilaterally into the septum (AP, +0.90 mm; ML, 0.90 mm, at a 10° angle). The SOM-IRES-Cre mouse line and the above AAV constructs were used in a large number of studies without reporting adverse phenotypic changes, and results of these studies were interpreted as general for mice and not specific for the particular strain[64,113,135,161,162]. Therefore, we do not consider using this particular mouse line a significant limitation of our study.

Virus injection surgeries were performed for the optogenetic stimulation experiment in PV-IRES-Cre mice. MS was targeted via a

trephine hole (AP, +0.90 mm; ML, 0.90 mm) at a 10° angle. An adeno-associated virus vector allowing Cre-dependent expression of channelrhodopsin2 [AAV 2/5. EF1a.Dio.hChR2(H134R)-eYFP.WPRE.hGH] or eYFP (AAV2/5.DIO.eYFP) was injected into the MS at 3.95, 4.45 and 5.25 mm depth from skull surface (200 nl at each depth). After the viral injections, the skin was sutured and local antibiotics (Neomycin) and analgesics (Buprenorphine 0.1 μg/g, s.c.) were applied.

To perform in vivo, awake, head restrained recordings during optogenetic stimulation, a small lightweight head plate was attached to the skull under isoflurane anesthesia using OptiBond (Kerr, Brea, US) adhesive and Paladur dental acrylic (Kulzer, Hanau, Germany). Two cranial windows (1.5 × 1.5 mm) were drilled above the left and right hippocampi (AP, −2.5 mm; ML, 2.5 mm) under stereotaxic guidance. An optic fiber (200 μm core diameter, Thorlabs GmbH, Newton, US) was implanted above the MS (AP, +0.9 mm; ML, 0 mm; 3.6 mm depth from the brain surface) and fixed with dental acrylic. A hole was drilled above the cerebellum for the insertion of the ground electrode. The craniotomies and drill hole were sealed with fast sealant (Body Double, Smooth-On, Macungie, US). After surgery, the mice were continuously monitored until recovered, then they were returned to their home cages for at least 48 h before starting habituation to head restraint.

### Chronic recording procedures

After recovery from the surgery, the Buzsaki-type septal probes were advanced in steps of 75-150 μm per day to reach the zone of cells with theta rhythm-modulated firing pattern. Within this zone, the probe was advanced in steps of 45 μm per day, as long as the theta-modulated firing pattern was still present, through a total of 8-35 days of recording. In four animals, the hippocampus was implanted with a linear silicon probe covering the CA1 layers, which was kept in a fixed position. In two animals, a Buzsaki-type silicon probe was implanted instead of the linear probe, which was advanced stepwise to reach the pyramidal layer of the dorsal hippocampus after the post-surgery recovery period.

The electrophysiological activity was registered by a multiplexing data acquisition system (KJE-1001, Amplipex Ltd, Szeged, Hungary) at 20 kHz sampling rate while the animals were moving freely either in a linear track or in their home cages. The position of the animal was tracked by a marker-based, high speed (120 frame/s) motion capture system and reconstructed in 3D (Motive, OptiTrack, NaturalPoint Inc, Corvallis, US).

### Optogenetic manipulations in the chronically implanted mice

Distinct groups of mice were used for activation (transduction with ChR2) and suppression (transduction with SwiChR) of the

hippocampo-septal fibers, as detailed above in the surgical procedures section. In the ChR2-transduced mice, the fibers passing the fimbria were illuminated by 5-ms-long pulses of blue light (447 nm, 5–8 mW, Roithner Lasertechnik, Vienna, Austria) at 20 Hz frequency for 10 s. The illumination epochs were repeated 14-22 times with manual timing during the home cage and linear track recordings. In the SwiChR-transduced mice, a consecutive pair of 10-s-long blue (447 nm, 5–8 mW, Roithner Lasertechnik, Vienna, Austria) and 1-s-long yellow (593 nm, 8-10 mW, IkeCool Corporation, Los Angeles, USA) light pulses (separated by a 2-s-long gap) was delivered by optical fibers glued 75-100 μm above the recording sites of the silicon probe. In mice injected with the EYFP control virus the same illumination protocol was used as in the SwiChR-transduced mice. The illumination was controlled by manual timing, repeated 8–26 times per home cage and linear track recording sessions. After each recording session, the probe implanted into the medial septum was lowered by 75 micrometers and after 12–24 h waiting, a new recording session was started. In both the ChR2- and SwiChR-experiments, the response upon the optogenetic manipulation was evaluated by the average firing rate change calculated in the combination of the home cage and linear track recordings. Neurons were classified as activated or suppressed based on a 10% or greater increase or decrease, respectively, in average firing rate during illumination compared to the 10-s pre-illumination period.

### Acute anesthetized recordings

Neural signals were acquired and amplified by multifunction input/output cards (PCI-6259, National Instruments, Austin, US) in rats or an electrophysiological recording system (RHD2000, Intan Technologies, Los Angeles, USA) in mice and digitized at 20 kHz. To increase the single unit yield, in case of the rat recordings, we waited 30 min for the tissue to stabilize after implantation, while implants in mice were inserted slowly (2 μm/s for MS, -10 μm/s for CA1) with a motorized micromanipulator (Robot Stereotaxic, Neurostar, Tübingen, Germany) to minimize tissue damage[163]. We carried out 30-min-long recordings in rats with 100 μm dorso-ventral separation, and 15-min-long recordings in mice at three distinct dorso-ventral positions within in the MS. Theta oscillations were induced by tail pinch (3 times 1 min in rats; once for 5 mins in mice).

### Acute optogenetic stimulation experiments

Prior to the day of the recording, mice were head-fixed and let to spontaneously run or sit on an air supported free-floating Styrofoam ball to adapt to the behavioral conditions. On the day of the recording, 128-channels silicon microprobes (UCLA 128 J, masmanidislab.neur-obio.ucla.edu) were lowered into the left and right dorsal hippocampi under isoflurane anesthesia. In four animals, the right hippocampus was implanted with silicon probes equipped with long vertical recording area (1.050 mm; UCLA 128 A, masmanidislab.neur-obio.ucla.edu) to enable CSD analysis through CA1 layers. Ground electrode was placed above cerebellum. Mice were allowed to recover from anesthesia for 1 h before recording.

The probes were advanced by using a micromanipulator (Kopf Instruments, Tujunga, US) until the pyramidal layer was detected by increased unit activity and the occurrence of ripple events. Electrophysiological recordings were performed by a signal multiplexing head-stage (RHD 128, Intan Technologies, LA, USA). Signals were acquired at 20k sample/s (Open Ephys 0.4.4.1[164]). Mouse locomotor activity was monitored with an optical computer mouse positioned close to the spherical ball at the equator. Following a 15–30-min-long control period, theta-modulated bursts of laser light (2 ms pulse length, 447 nm, 6-10 mW, Roithner Lasertechnik GmbH, Austria) were transmitted to the MS to stimulate PV neurons (tmS, theta-modulated stimulation). Four different intra-burst frequencies (22 Hz, 35 Hz, 54 Hz, 80 Hz, mimicking the frequency of physiological tSCs1-4) were applied at 8 Hz burst occurrence (Spike2 acquisition software and the CED 1401 laboratory interface family, Cambridge Electronic Design Limited, Cambridge, UK) in separate epochs. Laser stimulations were 2–3 min long during quiet wakefulness, controlled manually. Stimulations were repeated 2–3 times.

### Histological verification of the implant positions

After the recording sessions, animals were transcardially perfused with saline for 2 min and 4% para-formaldehyde (PFA) for 20 min and the brain was carefully removed and post-fixed in PFA for 24 h at 4 °C. Standard histological track reconstruction techniques[165] were used to verify the position of the silicon probe and optic fiber implants. Briefly, 60 μm thick coronal sections were cut (Vibratome VT1200S, Leica, Wetzlar, Germany), mounted on microscopic slides and covered with mounting medium (Vectashield, Vector Labs, Burlingame, CA, USA). Silicon probe implant tracks were localized by imaging red fluorescent DiI that had been applied on the electrodes before implantation, using a fluorescence microscope (Nikon Eclipse Ni microscope, Nikon Instruments, Melville, NY, USA). Fluorescent imaging of PV-expressing MS neurons confirmed the septal location of the fiber optic implant in the stimulation experiments (Supplementary Fig. 14). Finally, images were aligned with the corresponding sections of the stereotaxic mouse brain atlas[160] to enable precise reconstruction of the implant trajectory.

### Acute slice electrophysiology

Mice were decapitated under deep isoflurane anesthesia. The brain was removed and placed into an ice-cold cutting solution, which had been bubbled with 95% $O_2$–5% $CO_2$ (carbogen gas) for at least 30 min before use. The cutting solution contained the following (in mM): 205 sucrose, 2.5 KCl, 26 $NaHCO_3$, 0.5 $CaCl_2$, 5 $MgCl_2$, 1.25 $NaH_2PO_4$, 10 glucose. Coronal slices of 300-μm thickness were cut using a Vibratome (Leica VT1200S). After acute slice preparation, slices were placed into an interface-type holding chamber for recovery. This chamber contained standard ACSF solution at 35 °C which gradually cooled down to room temperature. The ACSF solution contained the following (in mM): 126 NaCl, 2.5 KCl, 26 $NaHCO_3$, 2 $CaCl_2$, 2 $MgCl_2$, 1.25 $NaH_2PO_4$, 10 glucose, saturated with 95% $O_2$–5% $CO_2$. Recordings were performed under visual guidance using Nikon Eclipse FN1 microscope with infrared differential interference contrast (DIC) optics. Patch pipettes were pulled from borosilicate capillaries (with inner filament, thin-walled, outer diameter (OD) 1.5) with a PC-10 puller (Narishige). The composition of the intracellular pipette solution was as follows (in mM): 110 potassium gluconate, 4 NaCl, 20 4-(2-hydroxyethyl)−1-piperazine-ethanesulfonic acid, 0.1 (ethylenebis(oxonitrilo))tetra-acetate, 10 phosphocreatine, 2 ATP, 0.3 GTP, 3 mg ml⁻¹ of biocytin adjusted to pH 7.3–7.35 using KOH (285–295 mosmol l⁻¹). Recordings were performed with a Multiclamp 700B amplifier (Molecular Devices), digitized at 20 kHz with Digidata analog-digital interface (Molecular Devices), and recorded with pClamp11 Software suite (Molecular Devices). For in vitro light illumination, we used a blue LED light source (Prizmatix Ltd.) integrated into the optical light path of the microscope. Two ms long light pulses were delivered at different theta-modulated gamma (tmS1-4, 2 pulses @22 and 3 pulses @ 35/50/80 Hz) frequencies. To record optogenetically evoked IPSCs, cells were clamped to −10 mV. Current clamp recordings of spiking modulation by PV+ MS input stimulation were performed with constant somatic current injection via the recording pipette to evoke spiking.

### Immunohistochemistry

To verify the specificity of the ChR2 expression in PV expressing neurons, a separate cohort of PV-Cre mice were sacrificed following 2–3 weeks of the virus injection into the MS (*n* = 3). After re-sectioning of the brain into 50 μm thick slices, sections containing the MS were washed in 0.1 M phosphate buffer and tris-buffered saline, and incubated in blocking medium (1% human serum albumin + 0.1% Triton-X

detergent) for 1 h. Then, sections were incubated in primary antibody against PV (PV 27, Swant, Switzerland, 1:2000) mixed with primary antibody against GFP (Thermofisher Scientific, USA, cat#A10262, 1:1000) in TBS at 4 °C for two days. After an extensive wash in TBS, the tissue was incubated in a secondary antibody solution containing Alexa 488 conjugated goat anti-chicken (Thermofisher Scientific, USA, Cat#A11039, 1:1000), Alexa 594 conjugated donkey anti-rabbit (Jackson ImmunoResearch Europe Ltd., UK, cat#711585152, 1:500) or Alexa 647 conjugated donkey anti-rabbit (Jackson ImmunoResearch Europe Ltd., UK, cat#711605152, 1:500) antibodies at 4 °C overnight. Finally, the sections were mounted on slides in aqua-Poly/Mount mounting medium (cat#18606, Polysciences, Inc., USA) and images were taken with a Nikon A1R confocal microscope.

After acute slice electrophysiology experiments, brain sections were fixed overnight in 4% PFA. After extensive wash in TBS, a similar staining protocol was used to detect parvalbumin or somatostatin immunoreactivity of the recorded cells. In these experiments, we applied SOM primary antibody (Origene, #AP3364SU-N, 1:200) and included 0.1% Triton-X detergent in every incubation step due to the thickness of the brain sections.

## Hippocampal theta state detection

The hippocampal pyramidal layer was detected in linear silicon probe recordings by analyzing the phase reversal of the theta oscillation, and was verified by histological reconstruction[6,166,167] similar to ref. 52. We used the signal recorded from the stratum radiatum for detecting theta oscillation, since theta rhythm had the highest amplitude and thus enable the most reliable detection in this layer (~400 μm and 250 μm below pyramidal layer in rats and mice, respectively, based on[160,168]). In case of the 2 mice with Buzsaki-type probes implanted in the hippocampus, we used the deepest channels for theta detection.

Hippocampal LFP was resampled at 1 kHz and filtered in the theta and delta bands (Supplementary Fig. 3a, e and i). Frequency band boundaries were defined based on the Fourier spectra of the raw LFP (delta, 0.5–4 Hz, theta, 4–12 Hz), similarly to ref. 52. The filtered theta and delta signals were Hilbert transformed, and their instantaneous amplitude was calculated as the magnitude of the complex Hilbert-transform. Hippocampal theta and non-theta states were determined based on the amplitude ratio of theta and delta band filtered signals. Theta oscillation was detected whenever this theta-delta amplitude ratio exceeded 2 for at least 3 s. Short interruptions in longer segments were not considered as state transitions (<3 s). Long theta epochs were defined as theta episodes >20 s. Theta transition periods were defined at the margins of long theta epochs, as the time windows in which the amplitude ratio was between 1.5 and 2.

## tSC extraction from hippocampal LFP

tSCs were extracted as described by Lopes dos Santos et al.[15] using their open source tSC extraction package (https://data.mrc.ox.ac.uk/data-set/tsc). First, LFPs were downsampled at 1 kHz and Ensemble Empirical Mode Decomposition (EEMD[65]) was applied to decompose the signal into its elementary spectral components called Intrinsic Mode Functions (IMFs). Low frequency, theta and supra-theta signals were defined as the sum of the IMFs with mean instantaneous frequencies below, within and above the theta frequency band, respectively. Second, we searched for local maxima and minima in the theta signal with absolute values above the low-frequency signal to detect potential theta peaks and troughs. We defined individual theta cycles by consecutive troughs occurring at theta band frequency (4–12 Hz in awake mice, 3–7 Hz in anesthetized rats, 2–6 Hz in anesthetized mice), separated by a peak. Third, spectrograms of the supra-theta signal were computed for each detected theta cycle with a set of complex Morlet wavelets with main frequencies ranging from the upper limit of the theta band to 200 Hz, in 1 Hz steps (Figs. 1f, 7a and Supplementary Fig. 14a). Fourth, spectral signatures were defined for each detected

theta cycle by taking the mean amplitude of these frequency components. Fifth, dimensionality reduction was performed by principal component analysis on the spectral signatures and tSCs were extracted from the spectral signatures based on the first five principal components using independent component analysis (ICA). Finally, tSC strength across individual theta cycles was defined as the projection of the given tSC onto the single cycle spectral signature. To examine septal firing during strong tSC presence, we defined strong tSCs by the following threshold applied on the tSC strengths, as introduced by[15]:

$$\text{Threshold} = \frac{\sigma \times \text{median}(|p - \text{median}(p)|)}{0.6745} + \text{median}(p) \quad (1)$$

where $p$ is strength distribution for the given tSC and $\text{median}(p)/0.6745$ is an estimate of the standard deviation of $p$ if outlying values are not considered. $\sigma$ (i.e. the number of standard deviations from the median) was set to 2 in line with[15]. For the short duration juxtacellular recordings we applied the same threshold with $\sigma = 1$ and tSCs expressed in <35 theta cycles recorded simultaneously with the MS neuron were not further analyzed. This lead to the exclusion of 6 juxtacellularly labeled neurons (MS33, MS77b, AJ48j, AJ42m, AJ43n, AJ45h). In an additional 7 juxtacellularly labeled neurons, coupling to tSC3 and/or tSC4 could not be analyzed (marked gray in Fig. 5c).

## Spike sorting of septal units

Mouse MS and CA1 recordings were fed to the template-based spike sorting software Kilosort[169]. Kilosort was run on an Nvidia Geforce GTX 1080 graphic card using the Matlab Parallel Computing Toolbox to reduce execution time. Automatically assigned clusters were manually curated by examining violations of the refractory periods, spatial distributions of action potential (AP) energies among neighboring channels, AP shapes and amplitudes, the principal components (PC) of the AP shapes, and crosscorrelograms between pairs of clusters using the Phy template-gui graphical user interface module (https://github.com/cortex-lab/phy). Quality of the retained clusters were assessed by objective measures derived from the interspike interval histograms and cluster isolation distances.

Spike sorting of the anesthetized rat MS recordings were performed using KlustaKwik[170] (http://github.com/klusta-team). Results were curated manually to exclude multi-unit activities and noise clusters by the examination of autocorrelograms (ACG, to exclude refractory period violations) and crosscorrelograms[134]. Similar to above, cluster quality was assessed by the Mahalanobis distance-based cluster isolation distances.

## Firing pattern analysis of MS neurons

Inter-spike interval histograms normalized by the number of spikes were calculated for each neuron separately and the average of these histograms was taken for all neurons to demonstrate bursting activity (Fig. 1b). To examine the spectral content of MS spiking activity, single unit spike trains were convolved with an 8-ms Gaussian kernel to generate pseudo-continuous signals[171] and a fast Fourier transform algorithm was applied to compute the frequency spectra of the signals. Spectra were smoothed with a 1 Hz wide moving average window, normalized with the area under the curve in the 4-200 Hz range and averaged across neurons (Fig. 1c).

## MS neurons' spiking activity as a function of tSC presence

We defined a number of spiking properties, to compare MS neurons' firing during theta cycles with different tSC presence. First, average firing rates were computed across theta cycles (Fig. 2a) for each neuron. Then, to determine spike phases, hippocampal theta phase was computed by linearly interpolating values between peaks (0 or 2π), zero-crosses (descending, π/2; ascending, 3π/2) and troughs (π), and

the phase values corresponding to the time of the spikes were taken. Significance of theta coupling was judged by Rayleigh's test for circular uniformity at $p < 0.01$ significance level. Theta phase−spiking probability histograms were calculated with π/8 bin width (Fig. 2a). Phase coupling strength and preferred phase were defined by the magnitude and the angle of the first trigonometric moment of the spike phases for each theta-coupled MS neuron.

Next, bursts were detected as series of two or more action potentials with inter-spike intervals <40 ms (typical threshold employed in medial septum studies[52,172]), which allowed us to calculate the burst-skip ratio of each neuron as the ratio of theta cycles without a burst. We calculated the mean intra-burst spike number as the number of spikes in the burst per theta cycle and the mean intra-burst frequency as the ratio of the length of the bursts and the intra-burst spike number.

Differences in these properties (i.e. firing rate, phase coupling strength, intra-burst frequency, intra-burst spike number, burst-skip ratio) as a function of tSC presence were tested with repeated measures ANOVA followed by Tukey's test for post hoc comparisons. In this analysis neurons with stable, sufficient activity were used, to avoid potential bias due to the non-uniform time distribution of tSCs ($n = 198$, MS neurons with >3 Hz firing rate and stable spike amplitude). The entire set of well isolated MS units ($n = 365$) were used for all other analyses, since those were not sensitive to this sampling issue. The properties ($P$) were normalized with their grand averages across all theta cycles ($P_{all}$) to better visualize relative differences across theta cycles with different tSCs (Fig. 2b, c, e and Supplementary Fig. 4) with the following formula:

$$P_{tSC\ cycles,\ norm} = \frac{P_{tSC\ cycles} - P_{all\ cycles}}{P_{all\ cycles}} \tag{2}$$

Average phase histograms were calculated from phase values relative to the preferred phase of each neuron (Fig. 2c and Supplementary Fig. 2f–h) or from the preferred phase of the tSCs (Supplementary Fig. 2b).

We calculated the inter-spike interval (ISI) histograms of the action potentials of single neurons during different theta cycles with 5 ms bins (Supplementary Fig. 7). To visualize ISI surplus related the concurrent tSCs, we first calculated the ISI histogram during theta cycles strongly expressing a given tSC. Then we took the same number of theta cycles expressing the given tSC the least (control histogram) and compared these two histograms, focusing on the bins corresponding to the frequency of the concurrent tSC. We repeated this analysis for identified PV-expressing MS neurons, normalized the difference histogram with the baseline firing rate of the neurons and averaged them across neurons. ISI histograms were plotted with both linear and logarithmic scaling to properly visualize ISI differences at lower and higher tSC frequencies (Supplementary Fig. 12).

## MS Rhythmicity groups

Rhythmic firing of MS neurons was quantified using autocorrelograms (ACG), similar to ref. 52. ACGs were calculated in a $\pm 3$ s window with 1 ms resolution and smoothed with a 20 ms moving average. MS neurons with >2 spikes per ACG bins on average were further analyzed. ACG peaks were detected between time lags corresponding to the delta (0.5–4 Hz) and theta (4–12 Hz) frequency bands. ACG values were averaged in $\pm 20$ ms windows around the peaks. Similarly, ACG values were averaged in $\pm 20$ ms windows around assumed troughs at the half and one and a half peak locations. The difference of these mean ACG values around peaks and troughs was normalized to the larger of the two, yielding a Delta and a Theta Rhythmicity Index between −1 and 1. Both indices were calculated during theta and non-theta segments for each MS neuron. To determine the significance of rhythmic modulation, we simulated spike trains using a Poisson-process, with

frequencies matched to the firing rate of the recorded neurons. Rhythmicity indices were calculated for the simulated spike trains as above, resulting in distributions corresponding to the null hypothesis of no rhythmic modulation. Critical values corresponding to the $p = 0.05$ significance levels were determined based on these distributions and applied as thresholds for judging significance of rhythmic modulation of MS neurons, for non-theta and theta segments separately. Rhythmicity was categorized by the larger of the two rhythmicity indices for those neurons that were both theta- and delta-rhythmic.

MS neurons that were significantly theta-rhythmic during both non-theta and theta segments were further categorized based on the presence of rhythmic burst firing. A Theta Burst Index (TBI) was calculated by comparing the average ACG in a burst lag window (10-30 ms) to the overall mean (see also ref. 52). MS neurons with TBI > 0 displayed constitutive rhythmic bursting activity, thought to be important for generating hippocampal theta oscillations[11,50,63,106]; they are referred to as constitutive bursting neurons. Theta-rhythmic MS neurons with TBI < 0 showed a regular rhythmic firing pattern, resembling the firing patterns of tonically active neurons of the striatum[173–175]; therefore, we refer to them as tonically active MS neurons. MS neurons that were theta-rhythmic when theta oscillations were present in the CA1 but non-rhythmic otherwise were referred to as theta-associated bursting MS neurons.

## Spike/stimulus triggered averages and spectrograms

The CA1 LFP signal or the pseudo-continuous MS signal (see above) was triggered on spikes of MS neurons and the average signal was calculated in $\pm 100$ ms time windows around MS spikes (spike triggered average, STA). Individual LFP averages were Z-scored. Similarly, stimulus triggered LFP averages were calculated aligned to optogenetic stimulation in $\pm 200$ ms time windows. Photovoltaic artefacts were observed in both the channelrhodopsin2-expressing and the control mice within 7 ms following stimuli, as reported before[176]. These were removed, and corresponding data points were spline interpolated. Stimulation-evoked theta oscillations and tSCs were clearly distinct from photovoltaic artefacts, characterized by longer latencies and phase changes as a function of recording depth.

Spectrograms were calculated by wavelet transformation using Morlet wavelets[177]. Spike and stimulus triggered spectrograms (STS) were calculated similarly to the LFP averages, separately for wavelet amplitudes and wavelet phases. Individual frequencies were normalized with their averages to give equal weight to each spectral component (Figs. 1d, 3a, and Supplementary Figs. 15).

To measure the coherence between CA1 LFP and (pseudo-continuous) MS spiking, we calculated the magnitude squared wavelet coherence. Similar to STS, individual frequencies were normalized with their averages (Fig. 1d).

## Phase coupling of MS neurons to tSCs

The IMF with the closest median frequency to the peak frequency of the tSC was considered as a reference signal for theta cycles strongly expressing the given tSC. MS spikes in each theta cycle were aligned to the largest trough of this reference signal and visualized as raster plots and peri-event time histograms (PETHs), smoothed by a Gaussian kernel (standard deviations: tSC1, 10 ms; tSC2, 8 ms; tSC3, 6 ms; tSC4, 4 ms; Fig. 3b and Supplementary Fig. 6).

To quantify phase coupling to tSCs, instantaneous tSC phase was computed as the phase of the complex Hilbert-transform of the reference signal. We considered two full cycles around the largest trough, reflecting the typical number of tSC cycles with sufficient signal-to-noise ratio within one theta cycle. Phase values were assigned to the spikes within this window by taking the instantaneous phase values at the spike times. This resulted in systematically longer time windows for slower tSCs, which could potentially lead to a systematic

bias in statistical power. To correct for this, we determined the tSC with the lowest number of action potentials, and spikes were randomly subsampled to yield an equal number of spikes for the other tSCs for unbiased statistical comparison (except for juxtacellular recordings, because of their short duration). Since we did not observe clear tSC-coupled firing patterns on the raster plots, we did not include the fastest tSC5 components in this analysis. We tested significant phase locking to the different tSCs after applying different temporal shifts on the MS spike train in the −100 ms – 100 ms range, recognizing that different temporal offsets were possible between the two time series[49,72]. Phase-coupling was tested by Rayleigh's test at each shift separately at $p < 0.05$. To control type I errors caused by testing phase coupling multiple times, we used Storey's false discovery rate method to adjust $p$-values[73]. Neurons with insufficient number of action potentials (<10 in theta cycles during any of the tSCs) were automatically considered as non-coupled to avoid false positives. Preferred tSC phase of each tSC-coupled MS neuron was defined as the angle of the first trigonometric moment of the spike phases. To control for correlations driven by associations with theta phase, MS spikes were randomly shuffled between theta cycles while conserving their time lag from the theta peaks and phase coupling to tSC was tested as for the non-shuffled spikes.

To visualize tSC-coupling on the population level, peri-event phase histograms (PEPHs) were calculated (Figs. 3c, 5d, 7d and Supplementary Fig. 19d). We calculated the firing rate in bins of π/20 radians around the largest tSC trough. PEPHs were smoothed with a Gaussian kernel (0.3π standard deviation) and Z-score normalized.

We examined the temporal structure of correlation between MS spiking and CA1 tSCs. Similar to crosscorrelation approaches, the temporal lag that maximizes the correlation, here expressed as phase coupling strength, reveals the temporal offset between corresponding changes in the underlying variables. Negative temporal lags indicate shifting the CA1 tSC signal back or, equivalently, shifting the MS spike train forward in time. We calculated Rayleigh's Z-value, the test statistic of Rayleigh's test for circular uniformity, as a function of a temporal shift added to the MS spikes in the range of −100 ms to +100 ms, with a resolution of 1 ms (similar to refs. 49,72). These Z-value functions were normalized by their maxima and averaged across all coupled MS neurons for each tSC. Average Z-value functions were smoothed with a Gaussian kernel (1.5 ms standard deviation) for visualization purposes (Fig. 4c, Supplementary Figs. 9c and 10c).

To examine the frequency accommodation of tSCs, we measured the time difference between the largest trough and the two troughs preceding and following the largest trough in each theta cycle strongly expressing the given tSC (Supplementary Fig. 11a). The average of these time differences was computed for each session and each tSC. To compare these dynamics with their septal counterparts, we first calculated PETHs triggered by the largest tSC trough for each single unit coupled to the given tSC, then bandpass-filtered them around the frequency of the concurrent tSC (passband frequency ranges: tSC1, 18-35 Hz; tSC2, 20-50 Hz; tSC3, 30-70 Hz; tSC4, 60-100 Hz). We searched for local maxima and minima in the filtered signal to detect peaks and troughs. To identify troughs and peaks most likely related to the largest amplitude tSC cycle, we located the trough and peak closest to the optimal lag that maximized the phase coupling and calculated the time difference between preceding and following troughs and peaks (Supplementary Fig. 11b–d).

### Analysis of stimulus-evoked CA1 tSCs and spiking activity
Stimulus evoked tSCs were analyzed using the same algorithms as the spontaneous tSCs (see the "tSC extraction from hippocampal LFP" section). Theta cycles and theta phase were automatically detected using EEMD as described in the "tSC extraction from hippocampal LFP"

section. Spikes of CA1 single units were aligned to the onset of the first pulse of each tmS burst and visualized as raster plots and peri-stimulus time histograms (PSTHs) smoothed by a Gaussian kernel of 2 ms standard deviation (Fig. 6c). Putative interneurons were identified based on their high firing rate, narrow spike shape and characteristic autocorrelogram (Supplementary Fig. 16b[2,178,179]).

### Current source density analysis
Current source density (CSD) analysis was performed on the laminar LFP recorded along the somato-dendritic axis of CA1 pyramidal neurons by linear equidistant ($\Delta z = 50\mu m$) silicon probe contact sites to localize currents sources of CA1 tSCs. It has been shown that if the extracellular conductivity tensor is assumed to be isotropic and homogenous, then the CSD signal can be expressed analogously with the Poisson equation of electrostatics (for the derivation see e.g. ref. 180). Consequently, in laminar structures, where the intra-layer changes in the extracellular potential ($\phi$) are negligible, the CSD can be approximated by the second spatial derivative of the LFP along the $z$-direction perpendicular to the layer[181,182]:

$$\mathrm{CSD} = -\sigma \frac{\partial^2 \phi}{\partial^2 z} \qquad (3)$$

where $\sigma$ is the isotropic and homogeneous extracellular conductivity. Therefore, we estimated the space and time dependent component of the CSD signal ($\widetilde{CSD}$) on the $n$-th contact site at time $t$ according to the following formula[15,22]:

$$\widetilde{CSD}(t)_n \sim \mathrm{LFP}(t)_{n-1} + \mathrm{LFP}(t)_{n+1} - 2 \times \mathrm{LFP}(t)_n \qquad (4)$$

where $LFP(t)_n$ is the potential measured on the $n$-th contact site at time $t$. The theta phase – frequency plots of the supra-theta CSD signal amplitude (Fig. 6d, e and Supplementary Fig. 17) were computed by the same wavelet approach as that applied on the LFPs (e.g. Fig. 1c, see the "tSC extraction from hippocampal LFP" section), similarly to refs. 15,22. Theta phase was defined based on the radiatum layer LFP for all plots, and tSCs were extracted from the radiatum layer LFP for Supplementary Fig. 17.

### Reporting summary
Further information on research design is available in the Nature Portfolio Reporting Summary linked to this article.

## Data availability
The awake and anesthetized rodent electrophysiology recording data used in this study have been deposited at https://doi.org/10.6084/m9.figshare.23798184. The optogenetic stimulation and electrophysiology recording data generated in this study have been deposited at https://doi.org/10.5281/zenodo.8191988. The juxtacellular recording data from anatomically identified neurons used in this study have been deposited at https://doi.org/10.5281/zenodo.8187903. Ventral hippocampal LFP recordings from mice in an elevated plus maze used in this study are available at https://doi.org/10.7272/Q6ZP44B9[84]. All data points underlying means, line graphs, box plots and scatter plots presented in the figures are provided in a Source Data Excel file with further labeled .mat files for panels presenting multidimensional data. Further data is available from the lead contact upon request. Source data are provided with this paper.

## Code availability
MATLAB code developed to analyze the data is available at https://github.com/kiralyb/MS_mod_tSC[183] (https://doi.org/10.5281/zenodo.8197408). Test data of a demo session of a chronically implanted mouse is available at https://doi.org/10.6084/m9.figshare.22060964.

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

## Acknowledgements
We thank Drs. György Buzsáki, Péter Somogyi, and László Záborszky for thoughtful comments and discussions on the manuscript. We thank István Katona for kindly providing access to an in vitro electrophysiology setup and Albert M. Barth for additional electrophysiology equipment. We thank Luigi Petrucco and Stoyo Karamihalev for open access science arts at SciDraw (https://doi.org/10.5281/zenodo.3925903 and https://doi.org/10.5281/zenodo.4312494). This work was supported by the "Lendület" Program (LP2015-2/2015) and NAP3.0 (NAP2022-I-1/2022) of the Hungarian Academy of Sciences, NKFIH KH125294, NKFIH K135561, the European Research Council Starting Grant no. 715043, the European Union project RRF-2.3.1-21-2022-00004 within the framework of the Artificial Intelligence National Laboratory and SPIRITS 2020 of Kyoto University to B.H.; NKFIH FK 129019 to M.J. and V.V.; National Brain Research Program 1.2.1-NKP-2017-00002 to R.F. and I.U.; NKFIH PD124175 and PD134196 to R.F.; NKFIH TUDFO/51757-1/2019-ITM to I.U.; ÚNKP-19-3, ÚNKP-20-3 and ÚNKP 21-3 New National Excellence Program of the Ministry for Innovation and Technology from the source of the National Research, Development and Innovation Fund to B.Ki.; the Requalification of the Spanish university system 2021–2023 Maria Zambrano modality (ZA21-009) program to S.M.B.; the Medical Research Council UK (Award MC_UU_00003/4) to D.D. and V.L.d.S. and (Award MRI/R011567/1) to A.J., M.S. and T.V. We thank the FENS-Kavli Network of Excellence for fruitful discussions.

## Author contributions
A.D., M.J., S.M.B., D.S., A.J., M.S., T.V., R.F., P.B. and B.H. performed experiments; V.V., T.F.F., T.J.V., I.U. and B.H. supervised experiments; B.Ki. and B.Ko. performed analysis and figures; B.H., V.L.S. and D.D. supervised analysis; B.Ki. and B.H. wrote the manuscript with input from all authors.

## Competing interests
The authors declare no competing interests.
