## [Peer Review File · Nature Communications]

The medial septum controls hippocampal supra-theta oscillationsREVIEWER COMMENTS

Reviewer #1 (Remarks to the Author):

Medial septum is classically associated with theta rhythm generation as its inhibition completely abolishes theta oscillations in the hippocampus. However, the role of MS in supra-theta rhythmogenesis has been largely unexplored. In this study, the authors performed simultaneous in vivo multi-electrode recordings in MS and CA1 hippocampal region in both mice and rats. They showed that MS spiking activity strongly correlates with theta-nested spectral components. Moreover, they report that MS firing predicted changes in supra-theta CA1 oscillations. They even go further and show that anatomically defined MS cell types with known projection targets couple to specific supra-theta oscillations. Finally, using optogenetic activation of PV+ neurons, the authors demonstrate the causal importance of these cells in driving supra-theta rhythms. This is a very elegantly designed and thoroughly conducted study which insights will significantly advance the field by shedding novel light on the medial septum role in hippocampal rhythmogenesis beyond theta oscillations. I only have several questions and suggestions that could make the claims in the study even stronger.

I am confused about the Figure 1D – why can't we see power and coherence in theta band? The authors are stating "...likely corresponding to theta-coupled bursts of action potentials". Please clarify.

Extended data Figure 1b – This is an interesting result showing that theta cycle expressing a tSC had higher frequency than those containing no strong tSC. While there a gradual increase in frequency for tSC1-tSC4, it seems that there is a decrease for tSC5. Could the authors speculate on physiological reasons underlying this phenomenon?

Figure 5 – The authors show convincingly that a significant proportion of Orchid, Teevra, and LRN cells couple to tSC. I am curious whether there is anything particular, in terms of electrophysiological and morphological properties, regarding the neurons that did not couple to tSC. That would be interesting to show.

While the authors discuss very nicely the potential role of the tSC driven by MS PV neurons stimulation in learning and memory, they don't show experimental evidence for it. In my opinion, performing a learning task while including MS PV stimulation protocol would amplify the impact of these already important findings.

Minor:

Methods section: There is a mentioning of SST-IRES-Cre in the Methods section. However, I haven't seen any evidence of this line being used in the study. Did I miss anything? Lines 609 and 610 also state virus injection in the dorsal hippocampi. However, in the results shown, the virus was only injected in the medial septum. Please clarify.

Reviewer #2 (Remarks to the Author):

Using a combination of high-density unitary and juxtacellular recordings as well as optogenetics, the authors explore the role of burst firing of the medial septal (MS) projections to the hippocampus for supra-theta frequency oscillations. They find systematic relationships between firing rates of MS cells and beta as well as gamma oscillations, show that MS cells temporally lead the occurrence of supra-theta LFP components, and link the activity of the anatomically identified types of MS cells with oscillations at specific supra-theta frequencies. Optogenetic stimulation revealed the sufficiently of burst-like input patterns of different frequencies for the generation of the supra-theta LFP oscillations.

Neuronal interactions generating crucial for cognition and behaviour fast network oscillations, and their coupling with slower rhythms, is a fundamental question in neuroscience. The authors contribute a conceptually novel set of results addressing this long-standing question for the important septo-hippocampal circuit using state-of-the-art experimental and analytical approaches. To back up their findings, they conducted numerous essential controls for potential confounds.

I have a number of comments aimed at improving the clarity of the presentation and of the interpretations.

In Fig. 2e, how were the intraburst frequencies related to the LFP frequencies defining individual tCSs? This seems of relevance for the interpretation of optogenetic stimulation results (Fig. 6) using different frequencies that evoke hippocampal LFP responses at corresponding frequencies. In other words, does the optogenetic pacing of beta and gamma via MS using different frequencies has counterparts in the physiological correlations of MS burst frequency with the frequency of the concurrent tCSs?

It would be important to show or at least discuss whether CA3 basket interneurons fire rebound spikes after individual spikes within bursts of MS neurons, like during the synchronous optogenetic stimulation of all MS projections, or whether they respond rather with the inhibition during bursts and discharge more profoundly only after a burst ends. In the former case, MS would be very closely integrated into the hippocampal gamma generation mechanisms, with the participation of known recurrent

connections between the regions, and a possible differential involvement in different supra-theta oscillations. In the latter, MS would be probably important for a theta-phase dependent global resetting of hippocampal and EC gamma generators. What other features of bursts can be relevant for cross-frequency coupling apart from their intrinsic frequency (e.g. the duration of bursts)?

327-329 "Interestingly, when stimulating with theta rhythmic single pulses at 8 Hz, the evoked theta cycles also carried beta-gamma-band spectral content, but with smaller amplitude and without the characteristic phase-amplitude coupling of

both spontaneous tSCs and tSCs evoked by theta-burst stimulation (Extended Data Fig. 12)." However, the amplitude of theta-modulated tSCs, according to the colour axis, in Ext. Data Fig. 12c (continuous laser pulses) seems higher than in spontaneous theta (Fig. 1c) but indeed lower than in Fig 6b (bursts of laser pulses), as far as one can judge by qualitative comparisons. LFP during pacing of theta by continuous pulses does feature the pattern of the gamma amplitude modulation similar to spontaneous theta oscillations (Bender et al., Nat. Commun., 2015, Fig 1c,e, compared in Fig 1d; Robinson et al., J Neurosci., 2016, Fig 4b; Quirk et al., Nat. Neurosci., 2021, Fig 1d). These recordings were obtained in freely behaving mice and, in part, by the stimulation of MS projections instead of somata, i.e. differently from the present study. The authors are encouraged to discuss possible similarities and differences between the preparations and provide precise quantitative comparisons of the cross-frequency coupling in a set of recordings, between spontaneous activity and various stimulation protocols.

Fast oscillations feature intrinsic accommodation of their frequency, likely related to the progressively diminishing excitatory drive during the envelopes. This property is likely present in the data here, since differences in gamma cycle amplitudes are mentioned. Do these LFP features during specific tSC correlate with the changes in (accommodating) intra-burst frequency of MS cells entrained to these tSCs?

To investigate the dynamics of MS-hippocampal interactions during theta-coupled fast oscillations, the authors may consider optogenetic inhibition of hippocampal projections to the MS.

The results in Fig 3 are nicely introduced by a clear description of the analysis shown in Fig 3a, yet the Results text related to a more extensive analysis in Fig. 3b and Extended Data Fig. 5 appears to be less clear. Line 217-218 state that "individual MS neurons showed phase-coupling to hippocampal tSCs" but Fig 3b has no phases, only time lags from tSC trough. Rayleigh's Z-value curves are shown for the first time only for further example cells in Extended Data Fig. 5. Further, line 220 "see the examples #3 and #4 in Fig. 3b", yet the #4 example cell does not seem to be in that figure. Lines 221-224 "Therefore, to test phase-locking of MS neurons to tSCs statistically, we performed Rayleigh's test for each neuron with temporal shifts added to the spikes in the -100 ms – 100 ms range. We found that 51% of all recorded MS neurons were phase-coupled to at least one tSC at any of the temporal shifts (Fig. 3c; $p < 0.05..$)" :

how are the rasters of firing probability according to the gamma phase shown on Fig. 3c related to the Z-shift analysis mentioned in the previous sentence?

It might be helpful to include references 49 and 130 in line 222, where the Z-shift method is briefly mentioned, and maybe to add more information to Methods as well.

Ext Fig 1d and e, considering that speed and theta frequency are correlated, one of these variables can be potentially spuriously correlated with ICS strength. What are the partial correlations for individual tSCs?

Lines 159-161 We hypothesized that a possible association between hippocampal tSCs and MS activity might also be reflected in differential theta-phase coupling properties of MS neurons." Would not testing this possibility actually involve the analysis of tSC theta phases in relation to the discharge phases of MS cells? The included in the manuscript analysis of theta phases of MS cells preferentially active during theta cycles with individual tSCs is informative but would be ultimately conclusive regarding the above hypothesis if theta LFP phases of tSC are also considered.

Lines 170-173 "...The majority of theta-coupled MS neurons fired at an earlier phase in tSC1-expressing cycles than in tSC4-expressing cycles (60%, 108/181, Fig. 2d; largest difference in preferred phase was found for the neurons most active during tSC1 cycles, Extended Data Fig.2a)." A more detailed presentation using mean phases and/or preferred phase histograms during each tSC would help here. Fig 2d seems to show that the (largest) difference between tSC1 and tSC4 phase is actually zero degrees, assuming that the red line is the mean of the distribution.

Color codes are confusing in Ext Data Fig 3: The colours of frequency bands (theta and delta) in filtered signals, present in both theta and non-theta states in the panel 3a,e, i, indicate not the frequency bands but the different states in 3b, d, f, h, j, l.

Fig. 3a, the y axis of the spectrogram is clipped to frequencies lower than the characteristic one of the tSC5, which makes the latter not as informative as plots for other tSCs.

The organization of Fig 2b-e needs optimization. The lower panel in Fig 2c is not readily associated with the upper one, where individual curves are difficult to distinguish, giving the impression that the point of 2c is not the difference in modulation strengths but something else, e.g. the similarity of phase patterns. The same happens with 2b, but there, the upper panel stands on its own and the lower one goes at first unnoticed as it is expected to come after 2d.

Line 1446, Top, average theta phase histogram (relative to the preferred phase): it would help to state that here, the phases of the discharge are considered, not to confuse it e.g. with the preferred phase of tSC.

Ext Fig 2 "tSC1 activated" sounds like they were experimentally activated rather than spontaneously more active during tSC1.

Line 72, "Indeed, entorhinally projecting orchid neurons of the MS can couple to entorhinal gamma oscillations Ref 64". In that work, CA1 gamma oscillations originated from the entorhinal cortex.

Line 85 "...neurons best phase locked to the future of tSC signals" needs clarification.

Line 106 "intra-burst inter-spike intervals corresponding to beta/gamma band activity": sounds like intraburst intervals can be outside fast frequency bands, by definition they usually correspond to beta and gamma. Alternatively, e.g. inter-spike intervals matching those of beta and gamma bands.

Line 1413 wavelet spectrogram of which time series are referred to here? (of MS spike trains?)

Line 123-124 "...an average of 30% of all theta cycles contained at least one strong tSC". It is a little surprising that only 30 % of cycles show tSC while certainly more cycles show the pattern of phase-modulated fast oscillations of LFP in the mouse.

Lines 174-175 "MS neurons often show phase-locked rhythmic bursting activity during hippocampal theta oscillations 48,50,68": also King, Recce, O'Keefe, EJN, 98.

Fig. 1f, how were the LFP amplitudes for different frequencies according to the theta phase computed and normalized?

How were the data in Ext Data Fig 1 b and c, 2b exactly normalised with their grand average?

The weight of tSC5 is 179 Hz but the maximum of the corresponding curve in Fig 1g seems to be above 200 Hz.

Please clarify AAV injections for different experiments in methods. Lines 609-611: Why were AAVs injected into the dorsal hippocampus? There seems to be no experiment using this preparation.

Reviewer #3 (Remarks to the Author):

The study by Kiraly B et al provides evidence that the MS contributes to CA1 activity dynamics, suggesting that the MS orchestrates hippocampal network activity through multiple temporal scales. Generally, this study was well designed, the data were clearly interpreted, and the manuscript was well organized and written. However, I have some concerns or comments as below:

1. The hippocampus is divided into two portions: dorsal hippocampus and ventral hippocampus, which have different functions. The dorsal portion is primarily for learning and memory while the ventral portion is for emotions. Is there any difference in the MS orchestrating neuron activities between the dorsal and ventral portions?

2. It remains unclear about the physiological relevance of MS orchestrating CA1 neurons, and what physiological or pathological conditions can alter or rewire the ability for MS to modulate dorsal and ventral CA1 neurons. Moreover, if chemogenetic or optogenetic assisted loss of function or gain of function of the septal-hippocampal projections can correct or reverse the altered CA1 activities in certain conditions, it would gain significant influence of this study in the research fields.

3. ChR2-assisted circuit mapping and whole cell patch clamp recordings are needed to determine whether PV-expressing GABAergic neurons in the MS project to the neurons localized within the dorsal and ventral CA1, CA2, CA3, and/or DG respectively, which will identify the involved direct and indirect circuit pathways to increase the significance of this study.

4. Does CA1 modulate MS neurons, particularly the PV neurons in the MS?

Response to Reviewers: The medial septum controls hippocampal supra-theta oscillations

Structure:

Reviewer comments: black, italic

Our replies: blue

Additions to the text: gold

We thank the Reviewers for their constructive comments that allowed for a clear improvement of our study. Notably, in the revised manuscript we address their comments by providing new data and analyses, as suggested:

1. We include multiple new analyses that showing that the MS is closely integrated into the hippocampal gamma generation mechanisms.
2. We add new analysis on the non-tSC-coupled MS neurons.
3. We include experiments on the hippocampo-septal inhibitory feedback projection.
4. We reveal tSCs in the ventral hippocampus by analyzing a publicly available data set.
5. We report a channelrhodopsin-assisted circuit mapping experiment that demonstrates direct targeting of PV+ and SOM+ interneurons by MS PV+ fibers in the dorsal CA1 and CA3.

All additions to the text are red-lined in the revised manuscript. Our point-by-point responses are laid out in details below.

Reviewer #1

Medial septum is classically associated with theta rhythm generation as its inhibition completely abolishes theta oscillations in the hippocampus. However, the role of MS in supra-theta rhythmogenesis has been largely unexplored. In this study, the authors performed simultaneous in vivo multi-electrode recordings in MS and CA1 hippocampal region in both mice and rats. They showed that MS spiking activity strongly correlates with theta-nested spectral components. Moreover, they report that MS firing predicted changes in supra-theta CA1 oscillations. They even go further and show that anatomically defined MS cell types with known projection targets couple to specific supra-theta oscillations. Finally, using optogenetic activation of PV+ neurons, the authors demonstrate the causal importance of these cells in driving supra-theta rhythms. This is a very elegantly designed and thoroughly conducted study which insights will significantly advance the field by shedding novel light on the medial septum role in hippocampal rhythmogenesis beyond theta oscillations. I only have several questions and suggestions that could make the claims in the study even stronger.

We thank the Reviewer for the positive evaluation of our study and for the constructive questions and suggestions.

I am confused about the Figure 1D – why can't we see power and coherence in theta band? The authors are stating "...likely corresponding to theta-coupled bursts of action potentials". Please clarify.

We thank the Reviewer for highlighting the lack of clarity on this issue. Wavelet power and wavelet coherence were normalized for each frequency separately to visualize changes (power/coherence increase/decrease) during the bursts compared to the time period before the bursts. Theta band power and coherence were approximately equally strong before and after the start of the bursts; therefore, this normalization removed the stationary theta-frequency component and revealed spectral increases in the beta and gamma bands. To clarify this, we added the following sentence in the figure legend in the revised manuscript: 'Spectrograms were normalized by the mean power for each frequency, thus visualizing the spectral changes around the triggering events.'

Extended data Figure 1b – This is an interesting result showing that theta cycle expressing a tSC had higher frequency than those containing no strong tSC. While there a gradual increase in frequency for tSC1-tSC4, it seems that there is a decrease for tSC5. Could the authors speculate on physiological reasons underlying this phenomenon?

This is an interesting question. According to the present models, mid-gamma components (tSC3 and 4) are related to memory encoding processes, while the slower tSC2 is linked to memory retrieval. Additionally, it has recently been shown that tSC1 is also amplified during memory retrieval¹. Memory encoding likely dominates explorative behaviors, which are often correlated with faster movement on average. Higher speed is accompanied by faster theta, which phenomenon might be related to place coding, which is probably a strong organizing factor while collecting episodic memories during movement. Regarding tSC5, it is thought to reflect internal processing in CA1 and we also found it to be unrelated to MS activity; this might be a reason why it does not follow the gradual frequency increase for tSC1-4, as accurately pointed out by the Reviewer. While this is yet rather speculative, we included a brief discussion of this exciting question in the Discussion section of the revised manuscript:

'Faster locomotion is accompanied by higher theta frequencies; this might be related to place coding, which is probably a strong organizing factor while collecting episodic memories during movement². Theta frequency was gradually increasing with the frequency of the spectral component nested in the theta cycle for tSC1-tSC4, likely reflecting that mid-gamma components are related to memory encoding processes^{3,4}. Interestingly, theta frequency during theta cycles expressing tSC5 components, which are thought to reflect internal processing in CA1, did not follow this trend.'

Figure 5 – The authors show convincingly that a significant proportion of Orchid, Teevra, and LRN cells couple to tSC. I am curious whether there is anything particular, in terms of electrophysiological and morphological properties, regarding the neurons that did not couple to tSC. That would be interesting to show.

We are grateful to the Reviewer for this comment, which prompted us to perform additional analyses that revealed interesting aspects of non-coupled MS neurons. We compared 5 key electrophysiological properties of the extracellularly recorded neurons coupled to at least one tSC to those not coupled to any. We included the following results as Extended Data Figure 8 and added corresponding text in the 'Most MS neurons are phase-coupled to beta-gamma band hippocampal tSCs' section of Results (one but last sentence).

First, tSC coupled neurons had significantly higher firing rates (Figure R1a below, two-sided Mann-Whitney U-test, $p < 0.001$).

Second, since most MS neurons' firing rate was correlated with the speed of the animal, we tested if there was a difference in speed correlation between tSC-coupled and non-coupled MS neurons. We found that the proportion of significantly speed correlated neurons (Spearman correlation, $p < 0.01$) was similar in the two neuron populations (Figure R1b, chi-squared test, $p = 0.555$), and the average strength of the speed correlation was not significantly different either (Figure R1c, $p = 0.699$).

Third, while the proportion of theta-coupled neurons was not significantly different between the tSC-coupled and non-coupled neurons (Figure R1d, chi-squared test, $p = 0.130$), the average strength of theta-coupling was higher in the tSC-coupled group (Figure R1e, two-sided Mann-Whitney U-test, $p < 0.001$).

Fourth, the distribution of preferred theta phases showed two opposing peaks in both groups. Interestingly, one peak was similar across groups, whereas the other peak was separated by 0.8 radians (Figure R1f, based on the fit of the mixture of two von Mises distributions).

Finally, we examined the burst properties of the neurons. Interestingly, when we examined the average inter-spike interval distribution of the two neuron groups, the non-coupled neurons also expressed a smaller, but prominent peak in the gamma range (Figure R1g). This could theoretically suggest the presence of gamma components in the MS not directly related to CA1 tSCs; however, a strong statistical caveat is that the amounts of data may preclude statistically conservative detection of tSC-coupling in MS neurons that show weaker tSC-coupling. Thus, some of the MS neurons classified non-coupled here may still show significant tSC-coupling in the infinite data limit. We did not find a significant difference in the average intra-burst frequency of the two neuron groups (Figure R1h, Mann-Whitney U-test, $p = 0.581$). In contrast, tSC-coupled MS neurons showed higher intra-burst spike counts (Figure R1i, two-sided Mann-Whitney U-test, $p < 0.01$) and skipped fewer theta cycles (Figure R1j, two-sided Mann-Whitney U-test, $p < 0.001$) than non-coupled MS neurons.

Figure R1 (Extended Data Figure 8). Firing properties of tSC-coupled and non-coupled MS neurons. **a** Mean firing rate distribution of tSC-coupled and non-coupled neurons ($p = 1 \times 10^{-5}$, Mann-Whitney U-test). **b** Proportion of MS neurons with significant speed correlation (Spearman correlation, $p < 0.01$) in the two MS neuron groups (chi-squared test, $p = 0.555$). **c** Spearman correlation coefficient between the firing rate and the speed of the animal ($p = 0.699$, Mann-Whitney U-test). **d** Proportion of theta-coupled (Rayleigh test, $p < 0.01$) neurons (chi-squared test, $p = 0.130$). **e** Theta-coupling strength distribution measured by the mean resultant length ($p = 3 \times 10^{-4}$, Mann-Whitney U-test). **f** Phase histogram of the preferred theta phase of the two neuron groups. **g** Average inter-spike interval histogram of the two neuron populations. The error shades show the standard error of the mean. **h-j** Distribution of burst parameters in the two MS neuron groups. **h** Intra-burst frequency ($p = 0.581$, Mann-Whitney U-test). **i** Intra-burst spike number ($p = 0.0099$, Mann-Whitney U-test). **j** Proportion of skipped theta cycles, termed 'burst-skip ratio' ($p = 5 \times 10^{-6}$, Mann-Whitney U-test). Boxes and whiskers show median, interquartile range and non-outlier range. * $p < 0.05$, ** $p < 0.01$, *** $p < 0.001$

Furthermore, results from the juxtacellularly recorded neurons suggest that “cell type” (defined by the combination of axon terminal field, neurochemical profile, and firing patterns) likely determines whether a neuron is not coupled to tSCs. We only found Teevra cells (CA3-projecting) and LRNs (CA3/DG-projecting) that were not coupled to tSCs, while all Orchid cells (presubiculum/EC-projecting) showed significant coupling to at least one tSC. Furthermore, calbindin-expressing, PV-negative LRNs are expected to be, at least in part, excitatory glutamatergic neurons, and 2 of 3 identified calbindin-expressing neurons were not tSC-coupled. In addition, the only NK1R-expressing neuron was not coupled to tSCs either. When we examined the anatomical location and the theta phase preference of the non-tSC-coupled neurons, we found that PV-negative neurons (MS86b, MS109o, MS13c) were located in the anterior regions of the MS (+1 mm from Bregma) and coupled to the descending phase of the pyramidal layer theta, while PV-expressing neurons were all located in the posterior-most region (+0.4 mm from Bregma) and coupled to the ascending phase of theta (see also Fig.2. in Salib et al., 2019). Nevertheless, given the small number of recorded and labeled neurons in the above subgroups, we did not include these latter observations in the manuscript.

While the authors discuss very nicely the potential role of the tSC driven by MS PV neurons stimulation in learning and memory, they don't show experimental evidence for it. In my opinion, performing a learning task while including MS PV stimulation protocol would amplify the impact of these already important findings.

We thank the Reviewer for this suggestion and agree that this would be a very exciting experiment to perform. However, it would also be a risky one, since given the importance of timing within the specific subsystems of the septo-hippocampal (or septo-hippocampo-entorhinal) circuit also highlighted here by the specific differences with respect to tSCs relatively close in frequency, it wouldn't be surprising to find it difficult to improve learning with still relatively crude optogenetic stimulation that imposes an artificial synchrony on the multifaceted PV-expressing MS populations (but see Figure R2). We already show some limitations of the stimulation approach by pointing out that there was a difference in the exact theta-phase preference of naturally occurring and artificially evoked tSCs. With these in mind, we feel that these experiments are presently out of the scope of this study, and they would most certainly take more time than the three-month time frame suggested by the Editor for submitting our revised manuscript.

Methods section: There is a mentioning of SST-IRES-Cre in the Methods section. However, I haven't seen any evidence of this line being used in the study. Did I miss anything? Lines 609 and 610 also state virus injection in the dorsal hippocampi. However, in the results shown, the virus was only injected in the medial septum. Please clarify.

We were conducting a parallel experiment focusing on hippocampo-septal feedback. As mentioned in the Methods, these sessions were not analyzed in the previous version of the manuscript. The SST-IRES-Cre mouse line (now referred to in the text as SOM-IRES-Cre for consistency) and the above AAV constructs were used in a large number of studies without reporting adverse phenotypic changes, and results of these studies were interpreted as general for mice and not specific for the particular strain⁵⁻⁹. Therefore, we did not consider using these mice a significant limitation of our study, especially as the results fit well with the juxtacellular recordings; nevertheless, we found it still important to include the injections in the Methods for a full disclosure.

However, in response to Reviewer questions regarding the possible role of this feedback loop in gamma generation, we now also discuss these experiments in the revised manuscript (see Figures R6 and R12; see also revised Methods lines 791-810).

Reviewer #2

Using a combination of high-density unitary and juxtacellular recordings as well as optogenetics, the authors explore the role of burst firing of the medial septal (MS) projections to the hippocampus for supra-theta frequency oscillations. They find systematic relationships between firing rates of MS cells and beta as well as gamma oscillations, show that MS cells temporally lead the occurrence of supra-theta LFP components, and link the activity of the anatomically identified types of MS cells with oscillations at specific supra-theta frequencies. Optogenetic stimulation revealed the sufficiency of burst-like input patterns of different frequencies for the generation of the supra-theta LFP oscillations.

Neuronal interactions generating crucial for cognition and behaviour fast network oscillations, and their coupling with slower rhythms, is a fundamental question in neuroscience. The authors contribute a conceptually novel set of results addressing this long-standing question for the important septo-hippocampal circuit using state-of-the-art experimental and analytical approaches. To back up their findings, they conducted numerous essential controls for potential confounds.

I have a number of comments aimed at improving the clarity of the presentation and of the interpretations.

We thank the Reviewer for finding our results novel and for the important comments, and we are happy that we share a view about the importance of the questions we addressed.

In Fig. 2e, how were the intraburst frequencies related to the LFP frequencies defining individual tSCs? This seems of relevance for the interpretation of optogenetic stimulation results (Fig. 6) using different frequencies that evoke hippocampal LFP responses at corresponding frequencies. In other words, does the optogenetic pacing of beta and gamma via MS using different frequencies have counterparts in the physiological correlations of MS burst frequency with the frequency of the concurrent tSCs?

We thank the Reviewer for raising this important point, which prompted us to perform new analyses. To address this question, we first compared the mean intra-burst frequencies of the neurons coupled to multiple tSCs during theta cycles expressing different tSCs, but this approach did not reveal significant differences between the means (repeated measures ANOVA, $p > 0.05$). Therefore, to better understand the underlying mechanisms, we compared the inter-spike interval (ISI) histograms of these neurons between theta cycles strongly expressing or not expressing a given tSC (Figure R2a, the same example neurons as in Figure 3). Confirming the Reviewers' assumption, we found MS neurons that showed clear ISI surplus around the corresponding frequency of the concurrent tSC if the neuron was coupled to that tSC (example MS neuron #2 and #3; see also example MS neuron #2 during tSC4 cycles, where no significant coupling was not detected and accordingly we found no ISI surplus around the frequency of tSC4). On the other hand, we also found MS neurons where the ISI histograms were not clearly correlated with tSC presence (example MS neuron #1), suggesting that in these cases, phase coupling manifested in MS spike timing modulation that did not necessarily affect all action potentials (or affected them unevenly) during the tSCs. This latter finding could underlie the lack of significant modulation of mean intra-burst frequencies mentioned above.

Interestingly, we also often observed inter-spike interval surplus in the adjacent bins around mean tSC frequency, and also distinctly in the fast gamma range (example neuron #3: tSC1, tSC2). This likely reflects our observation of fast-gamma bursts following each other in beta or slower gamma rhythmic patterns. Additionally, tSC frequencies show variability both within and across theta cycles (see also Figure R5a and c), which might also underlie the ISI surplus in the bins adjacent to the ones matching the mean tSC frequency. In some cases, we also observed ISI surplus around the half of the tSC frequency, likely related to cycle skipping (example neuron #1: tSC2, tSC3, tSC4; example neuron #2: tSC2, tSC3, tSC4).

Next, we focused on the identified PV+ neurons, to aid the interpretation of the experiments with the stimulation of the PV+ MS neurons. We took the difference between the ISI histograms during strongly tSC-expressing and not tSC-expressing theta cycles (calculated based on matched number of cycles), normalized them with the firing rate of the neuron, and took the average across all PV+ neurons coupled to a given tSC (Figure R2b). This analysis revealed peaks on the ISI difference histograms around each of the mean tSC frequencies and also in the fast gamma burst range in all cases, revealing that optogenetic pacing of beta and gamma bursts indeed have physiological counterparts in PV+ MS neurons.

We included these data in Extended Data Figures 7 and 12 and corresponding text in 'Most MS neurons are phase-coupled to beta-gamma band hippocampal tSCs' and 'Differential coupling of anatomically identified MS cells to CA1 tSCs' sections of Results.

a**b**
Figure R2 (Extended Data Figure 7 and 12). Intra-burst frequency of MS neurons is correlated with CA1 tSC presence. **a** Inter-spike interval histograms of three extracellularly recorded example MS neurons coupled to multiple tSCs (same examples as in Figure 3) during theta cycles strongly expressing a given tSC (colored bars) compared to theta cycles expressing the given tSC the least (gray bars). The colored vertical lines indicate the frequency of the concurrent tSC. The tSC4 panel of the example neuron #2 is faded to indicate that this neuron was not coupled to tSC4. **b** Difference between the inter-spike interval histograms during theta cycles with the most and least tSC content (colored and grey histograms in the examples), normalized by firing rate and averaged across identified PV+ tSC-coupled MS neurons. The solid line shows the difference on a linear scale (y-axis on the left), while the dashed line on a logarithmic scale to highlight the difference at slower frequencies (y-axis on the right; note that negative differences cannot be displayed on the logarithmic scale). The error shades show the standard error of the mean. The colored rectangles show the frequency ranges of the tSCs.

It would be important to show or at least discuss whether CA3 basket interneurons fire rebound spikes after individual spikes within bursts of MS neurons, like during the synchronous optogenetic stimulation of all MS projections, or whether they respond rather with the inhibition during bursts and discharge more profoundly only after a burst ends. In the former case, MS would be very closely integrated into the hippocampal gamma generation mechanisms, with the participation of known recurrent connections between the regions, and a possible differential involvement in different supra-theta oscillations. In the latter, MS would be probably important for a theta-phase dependent global resetting of hippocampal and EC gamma generators. What other features of bursts can be relevant for cross-frequency coupling apart from their intrinsic frequency (e.g. the duration of bursts)?

First, we investigated putative CA1 interneurons in the *in vivo* recordings with optogenetic MS PV stimulation, and indeed we did find neurons that fired rebound spikes after a brief initial suppression following individual pulses of the light stimulation burst, suggesting a closely integrated role for the MS PV neurons in the CA1 gamma generation mechanism (Figure R3; Figure 6c and Extended Data Figure 16a were updated in the manuscript accordingly).

Second, we performed a new ChR2-assisted circuit mapping and whole cell patch clamp experiment, which confirmed a direct connection from MS PV+ neurons to PV+ and somatostatin-expressing (SOM+) interneurons in the dorsal CA1 and CA3. When we applied theta-modulated burst stimulation (tmS) *in vitro*, most hippocampal interneurons were capable of following the tmS1-3 burst stimulation patterns with IPSCs/IPSPs, which could translate to modulated firing patterns at the respective frequencies (see Figure R11; see also our response to the 3rd point of Reviewer #3). tmS4 did not entrain interneurons' firing *in vitro*, contrary to some *in vivo* recorded interneurons. This was likely due to a less reliable, more stochastic drive of PV axons *in vivo*, resulting in an even distribution of evoked IPSPs through the tmS4 burst, likely modeling spontaneous activity of MS PV+ input to CA1 better. Cut-off vs. intact PV+ MS axons, constant somatic depolarization-induced firing *in vitro* vs. synaptic input-driven firing *in vivo* and potential sampling bias could also contribute to this difference.

Finally, the fine-grained analysis of MS intra-burst spiking and tSC presence was demonstrated in our response to the previous point, which also pointed to cycle-by-cycle MS-CA1 correlations within tSCs (in accordance with Figures 3 and 4), again arguing for the 'closely integrated' gamma generation model.

In addition, we considered examining simultaneous recordings of MS and hippocampal neurons *in vivo* to reveal potential spike-spike correlations between the MS PV+ and hippocampal (CA1 and/or CA3) putative

interneurons (note that our present recordings focused on CA1 laminar LFP using linear silicon probes). However, we would expect phase-locking to tSCs of both MS (Fig. 3-4) and hippocampal units; thus, revealing spike-spike correlations beyond what is expected from mutual phase-locking could require prohibitively large datasets. Therefore, we dropped this idea.

Figure R3 (Left: Figure 6c; Right: Extended Data Figure 16a). Putative CA1 interneurons follow MS PV+ neuron stimulation. Spike rasters and peri-stimulus time histograms of two putative CA1 interneurons aligned to the onset of each tmS (theta-modulated stimulation) burst sorted by the stimulation frequency (color-coded). Note that the single pulses of the stimulation burst (marked by blue shaded areas) are followed by a brief initial suppression and then an activation of spiking.

327-329 "Interestingly, when stimulating with theta rhythmic single pulses at 8 Hz, the evoked theta cycles also carried beta-gamma-band spectral content, but with smaller amplitude and without the characteristic phase-amplitude coupling of both spontaneous tSCs and tSCs evoked by theta-burst stimulation (Extended Data Fig. 12)". However, the amplitude of theta-modulated tSCs, according to the colour axis, in Ext. Data Fig. 12c (continuous laser pulses) seems higher than in spontaneous theta (Fig. 1c) but indeed lower than in Fig 6b (bursts of laser pulses), as far as one can judge by qualitative comparisons. LFP during pacing of theta by continuous pulses does feature the pattern of the gamma amplitude modulation similar to spontaneous theta oscillations (Bender et al., Nat. Commun., 2015, Fig 1c,e, compared in Fig 1d; Robinson et al., J Neurosci., 2016, Fig 4b; Quirk et al., Nat. Neurosci., 2021, Fig 1d). These recordings were obtained in freely behaving mice and, in part, by the stimulation of MS projections instead of somata, i.e. differently from the present study. The authors are encouraged to discuss possible similarities and differences between the preparations and provide precise quantitative comparisons of the cross-frequency coupling in a set of recordings, between spontaneous activity and various stimulation protocols.

We thank the Reviewer for this insightful comment. The cited sentence was indeed confusing; we changed it. Also, we introduced a more thorough discussion on this observation as suggested by the Reviewer:

'In accordance with previous studies¹⁰⁻¹², we observed that CA1 theta cycles induced by theta-rhythmic single-pulse stimulation also showed beta-gamma-band spectral content (Extended Data Fig. 12). These components were smaller in amplitude than tSCs evoked by tmS burst stimulation. We speculate that

theta-rhythmic intra-septal PV+ neuron stimulation could recruit part of the beta-gamma generating circuits of the MS; this is supported by a study where theta-rhythmic intra-septal stimulation of glutamatergic neurons induced theta and gamma oscillations in the CA1¹⁰, since local MS glutamatergic neurons provide the majority of excitatory inputs to MS GABAergic neurons¹³ and were suggested to relay the necessary excitation for MS GABAergic rhythm generation circuits¹⁴. However, intra-hippocampal gamma generation mechanisms could also contribute, as suggested by a study that applied intra-hippocampal theta-rhythmic stimulation of PV-expressing MS fibers¹¹. Nevertheless, this study did not exclude retrograde activation of cell bodies in the MS, thus not necessarily differentiating between septal and hippocampal mechanisms. Additionally, analyzing tSCs separately allowed us to reveal that beta/gamma components evoked by theta-rhythmic PV stimulation did not show the characteristic phase-amplitude coupling of spontaneous tSCs and tSCs evoked by theta-burst stimulation.'

Additionally, we calculated the average spectral content of theta cycles during different stimulus protocols and during spontaneous theta activity (n = 6, Figure R4a). First, we compared the mean power (area under the spectral content curve) and we found that theta cycles during 8 Hz stimulation indeed carried smaller beta-gamma spectral content (20-100 Hz) than those during any of the burst stimulations, also confirmed when comparing spectral peak amplitudes (Figure R4b). We included this result in Extended Data Figure 17.

When we compared spontaneous and optogenetically induced theta cycles, mean spectral power was higher in the induced case, as observed by the Reviewer. At the same time, peak amplitude of the spectral content curve was only significantly larger in case of tSC3 and tSC4. However, induced oscillations showed considerable variability in amplitude and we found it likely that those amplitudes depend on the number of recruited PV neurons; therefore, we think that this latter comparison is not conclusive.

Figure R4 (Extended Data Figure 17d-e). Power spectra of spontaneous and stimulation-induced theta cycles. **a** Left, average spectral signatures from control sessions of $n = 6$ animals computed based on spontaneous theta cycles expressing a given tSC or all theta cycles pooled together. Right, average spectral signature of all theta cycles during different stimulation protocols. The shaded areas represent the standard error of the mean. **b** Distributions of the area under the spectral signature curve (top) and the peak spectral signature amplitude (bottom) for the different spontaneous and stimulation-induced theta cycles. Boxes and whiskers show median, interquartile range and non-outlier range. Differences were statistically tested with the paired, two-sided Wilcoxon signed rank test. * $p < 0.05$.

Fast oscillations feature intrinsic accommodation of their frequency, likely related to the progressively diminishing excitatory drive during the envelopes. This property is likely present in the data here, since differences in gamma cycle amplitudes are mentioned. Do these LFP features during specific tSC correlate with the changes in (accommodating) intra-burst frequency of MS cells entrained to these tSCs?

We thank the Reviewer for this great idea. We examined average inter-trough intervals of the tSCs aligned to the tSC trough with the largest amplitude (Figure R5a) and tested for correlates in the firing patterns of tSC-coupled MS single units (Figure R5b). To do this, we bandpass-filtered the peri-event time histograms triggered on the largest tSC troughs (see Figure 3b) to extract the tSC-related components of the firing rate changes and searched for peaks and troughs (Figure R5b, blue curve). As we demonstrated in Figure 4, MS units usually coupled to CA1 tSCs with a temporal difference; therefore, we used the approach described in the ‘MS neurons temporally lead hippocampal beta-gamma band activity’ section of the manuscript to find the optimal lag that realized the strongest phase coupling, in order to estimate which troughs and peaks of the filtered PETH might correspond to the largest amplitude tSC cycle and calculated inter-

trough/peak intervals preceding and following this peak and trough. When we repeated this analysis for each recording session and tSC, we found, that CA1 tSCs were consistently slower around the largest trough (Figure R5c). We first focused on the clear peaks and troughs of tSC-coupled neurons and observed similar inter-trough/peak interval tendencies as of the tSCs. Namely, inter-trough/peak intervals were longer around the optimal lag (Figure R5b and d), with some variability in the position of the longest interval that might reflect small uncertainty in the detection of the corresponding CA1 tSC and MS firing rate peaks/troughs. When we calculated the population average inter-peak/trough interval profile of all MS neurons coupled to each tSC (Figure R5e), we again observed tendencies similar to those typical for CA1 tSCs (Figure R5c). These new results were added to the Extended Data Figure 11 and the 'MS neurons temporally lead hippocampal beta-gamma band activity' section (last paragraph) of the Results.

Figure R5 (Extended Data Figure 11). Frequency accommodation of CA1 tSCs is reflected in the firing of MS neurons. **a** Average tSC1 inter-trough intervals preceding (-2, -1) and following (+1, +2) the largest amplitude trough in an example session. Error bars show the standard deviation to highlight the large variability, while the shaded area represent the standard error of the mean to demonstrate that the trends revealed are statistically reliable due to the large number of cycles analyzed. **b** Left, peri-event time histogram (PETH) of an example neuron triggered on the largest troughs of CA1 tSC1 cycles from the same session as in panel 'a'. The blue line shows the bandpass filtered PETH (18 Hz-35 Hz), which was used to find tSC1-related peaks and troughs before and after the optimal time lag (red line) that realizes the strongest phase locking. Right, inter-trough and inter-peak intervals of the filtered firing rate signal of the example neuron. **c** Average tSC inter-trough intervals for each session (n = 32). **d** Inter-trough and inter-peak intervals of the peaks and troughs of the 10 most tSC-coupled neurons for each tSCs. **e** Average inter-peak/trough intervals of all MS neurons coupled to a given tSC. The shaded areas represent the standard error of the mean.

To investigate the dynamics of MS-hippocampal interactions during theta-coupled fast oscillations, the authors may consider optogenetic inhibition of hippocampal projections to the MS.

We thank the Reviewer for this suggestion. GABAergic neurons projecting from the hippocampus to the MS are in 93% somatostatin-expressing and most frequently located in the stratum oriens of the CA1 and CA3 areas¹⁵. We performed optogenetic suppression experiments using SwiChR in 4 of the freely moving SOM-IRES-Cre mice used in this study. Mice underwent bilateral injections of AAV5-Ef1a-DIO-SwiChRca-TS-EYFP in the dorsal CA1. To enable the inhibition of the hippocampo-septal projections, each shank of the septal silicon probes in these animals were equipped with an optical fiber. 4 additional mice were injected with AAV5-Ef1a-DIO-EYFP control virus and implanted with 2 optical fibers bilaterally.

These experiments revealed that the proportion of theta cycles strongly expressing tSCs significantly increased during the inhibition of these feedback projections compared to control periods without stimulation (paired, two-sided Wilcoxon signed rank test, $p = 0.0015$, Figure R6; no effect observed in control animals, $p = 0.7422$, Extended Data Fig. 19a). These results suggest that this inhibitory feedback loop may play a role in suppressing the septo-hippocampal generator of supra-theta oscillations. We included these results in a new Results section and in Figure 6.

Figure R6 (Figure 6f and 6g). Optogenetic inhibition of the hippocampo-septal projections increases tSC occurrence in the CA1. Left, schematic of the experiment using SwiChR injected into the CA1 and optic fibers implanted into the medial septum. Right, proportion of theta cycles expressing tSCs during inhibition and control periods (data points belonging to the same recording session are connected with gray lines). Boxes and whiskers show median, interquartile range and non-outlier range. **, $p < 0.01$; $p = 0.0015$, two-sided Wilcoxon signed rank test.

We further examined this question in our response to the 4th question of Reviewer #3, and we demonstrate that there are significantly more tSC-coupled neurons among those activated during photostimulation (possibly targeted by the inhibitory feedback pathway and thus disinhibited during photostimulation) compared to the suppressed ones. Our results also indicate that this difference is explained by a preferential recruitment of MS neurons that follow CA1 tSCs, suggesting a role for these neurons in the feedback loop of gamma generation (Figure R12).

The results in Fig 3 are nicely introduced by a clear description of the analysis shown in Fig 3a, yet the Results text related to a more extensive analysis in Fig. 3b and Extended Data Fig. 5 appears to be less clear. Line 217-218 state that "individual MS neurons showed phase-coupling to hippocampal tSCs" but Fig 3b has no phases, only time lags from tSC trough. Reyleigh's Z-value curves are shown for the first time only for further example cells in Extended Data Fig. 5. Further, line 220 "see the examples #3 and #4 in Fig. 3b", yet the #4 example cell does not seem to be in that figure. Lines 221-224 "Therefore, to test phase-locking of MS neurons to tSCs statistically, we performed Rayleigh's test for each neuron with temporal shifts added to the spikes in the -100 ms – 100 ms range. We found that 51% of all recorded MS neurons were phase-coupled to at least one tSC at any of the temporal shifts (Fig. 3c; $p < 0.05..$)" : how are the rasters of firing probability according to the gamma phase shown on Fig. 3c related to the Z-shift analysis mentioned in the previous sentence? It might be helpful to include references 49 and 130 in line 222, where the Z-shift method is briefly mentioned, and maybe to add more information to Methods as well.

We agree with the Reviewer that this part was not sufficiently clear. Also, there was a mistake in the text regarding the numbering of the example neurons. We edited the text according to the suggestions of the Reviewer and hope that it has become much clearer. We also added the Z-value curves for the examples in the main figure and marked their phase histograms in Fig.3C by arrows. We added more details to the Methods section on the Z-shift analysis.

Ext Fig 1d and e, considering that speed and theta frequency are correlated, one of these variables can be potentially spuriously correlated with ICS strength. What are the partial correlations for individual tSCs?

We thank the Reviewer for this very valid point. We calculated the partial correlations (Figure R7), and we found that partial correlations were only moderately weaker and still strongly significant ($p < 0.001$) when the effect of the third variable was removed. This also applied to the speed-theta frequency correlation with and without the effect of the tSC strengths. We included these data in the first Results section (last paragraph) and in Extended Data Figure 1f.

Figure R7 (Extended Data Figure 1f). Partial correlations between tSC strength, theta frequency and animal speed. Each matrix corresponds to correlations based on theta cycles strongly expressing a given tSC. The fields in the upper right triangles show correlation between the corresponding two variables without controlling for the third variable, while the bottom left corners represent partial correlations with controlling for the effect of the third variable. All correlations are significant ($p < 0.001$, Spearman correlation).

Lines 159-161 We hypothesized that a possible association between hippocampal tSCs and MS activity might also be reflected in differential theta-phase coupling properties of MS neurons." Would not testing this possibility actually involve the analysis of tSC theta phases in relation to the discharge phases of MS cells? The included in the manuscript analysis of theta phases of MS cells preferentially active during theta cycles with individual tSCs is informative but would be ultimately conclusive regarding the above hypothesis if theta LFP phases of tSC are also considered.

Following on the Reviewer's line of thoughts, we note that the MS participating in controlling tSCs would indeed predict that the MS theta phases change in correlation with tSC theta phase. We now demonstrate this by showing the preferred theta phase distribution of theta-coupled MS neurons relative to the preferred theta phase of the tSCs below. As we expected, most MS neurons had a preferred theta phase relatively close to the preferred phase of the tSCs (Figure R8a), especially for theta cycles with faster tSCs (Figure R8b, left). The majority of the MS population had a slightly earlier phase preference than the beta band tSC1 in cycles strongly expressing tSC1 (Figure Rb, right). However, we would like to note that this analysis will inevitably be influenced by temporal differences between MS and CA1 as revealed by the Z-shift analysis, but maybe even more so by the potentially uneven distribution of MS spikes across tSC cycles as well as MS spikes at other theta phases likely not directly related to tSCs.

Next, following up on the Reviewer's 'closely integrating vs. resetting' question, a closely integrated MS-CA1 generator suggests a correlation between tSC theta phase and MS spiking beyond the averages shown above, on a cycle-by-cycle basis. We addressed this by a shuffled control in which theta cycles are randomized, which largely removes the tSC-locking MS neurons (Extended Data Fig. 6). To examine this further, we calculated circular-circular correlation between the discharge phase of theta-coupled MS neurons and preferred theta-phase of the concurrent tSC, pooled across theta cycles strongly expressing any of the tSCs. Theta phase values relative to mean phase of the MS cells / tSCs were used, to specifically test whether deviations from the individual phase preferences occurred in a correlated manner. This approach revealed a significant circular-circular correlation ($p < 0.05$) in 23% of theta-coupled MS neurons. As expected, most of these correlations were positive (76%). However, we note that performing analyses across noisy signals (for the tSCs, the LFP processed by EEMD is only an approximation of the CA1 population dynamics; for MS, spike time variability and variability across cells are important factors) on a theta cycle basis maybe stretching the borders of the realm of possibilities, cautioned by the 24% negative correlations, part of which are likely spurious. Given these caveats, we did not include these analyses in the revised manuscript, but if the Reviewer and the Editor finds them informative, we are open to re-consider this.

Figure R8. Theta phase preference of MS neurons relative to the preferred phase of the tSCs. a Distribution of preferred theta phase of theta-coupled MS neurons relative to the preferred theta phase of the concurrent tSC. **b** Mean relative preferred theta phase of theta-coupled MS neurons in theta cycles expressing different tSCs. **c** Average relative theta phase histogram of theta-coupled MS neurons.

Lines 170-173 "...The majority of theta-coupled MS neurons fired at an earlier phase in tSC1-expressing cycles than in tSC4-expressing cycles (60%, 108/181, Fig. 2d; largest difference in preferred phase was found for the neurons most active during tSC1 cycles, Extended Data Fig.2a)." A more detailed presentation using mean phases and/or preferred phase histograms during each tSC would help here. Fig 2d seems to show that the (largest) difference between tSC1 and tSC4 phase is actually zero degrees, assuming that the red line is the mean of the distribution.

We thank the Reviewer for prompting us to perform a more in-depth analysis of theta-phase differences of MS neurons with respect to tSC occurrence. We calculated phase histograms of preferred theta phases during each tSC as suggested (Figure R9a). In addition, we calculated the mean absolute/signed theta phase differences of MS neurons for each pairs of tSCs, including the cycles without strong tSC expression (Figure R9b). These results demonstrate that the mean theta phase difference of MS neurons was indeed largest between tSC1 and tSC4 cycles (both in absolute and signed terms), and MS neurons were coupled to an earlier theta phase in tSC1 cycles compared to other theta cycle types. We added these data to Extended Data Figure 2 and the 'MS single neuron firing is correlated with hippocampal tSCs' section of the Results. The red line in Fig. 2d referred to zero degrees, but we agree that it was misleading and in the revised version it indicates the average.

Figure R9 (Extended Data Figure 2a and b). Theta-phase preference of MS neurons in theta cycles with different tSCs. a Rose diagram (circular histogram) of the preferred theta phase of the theta-coupled MS neurons during theta cycles expressing different tSCs. **b** Mean absolute (left) or signed (right) difference of preferred theta phase of MS neurons between theta cycles expressing different tSCs.

Color codes are confusing in Ext Data Fig 3: The colours of frequency bands (theta and delta) in filtered signals, present in both theta and non-theta states in the panel 3a,e, i, indicate not the frequency bands but the different states in 3b, d, f, h, j, l.

We thank the reviewer for pointing this out. We modified the colors.

Fig. 3a, the y axis of the spectrogram is clipped to frequencies lower than the characteristic one of the tSC5, which makes the latter not as informative as plots for other tSCs.

We agreed with the comment and modified the figure accordingly. This revealed increased hippocampal fast gamma activity following the spikes of the neuron, but it was not specific to tSC5 cycles.

The organization of Fig 2b-e needs optimization. The lower panel in Fig 2c is not readily associated with the upper one, where individual curves are difficult to distinguish, giving the impression that the point of 2c is not the difference in modulation strengths but something else, e.g. the similarity of phase patterns. The same happens with 2b, but there, the upper panel stands on its own and the lower one goes at first unnoticed as it is expected to come after 2d.

We rearranged the figure: the two graphs of panel b are now next to each other, and the former panel c has been split into two panels.

Line 1446, Top, average theta phase histogram (relative to the preferred phase): it would help to state that here, the phases of the discharge are considered, not to confuse it e.g. with the preferred phase of tSC.

We modified the sentence as follows. 'Average theta phase histogram (relative to the preferred phase) of theta-phase-coupled MS neurons' firing during theta cycles with different tSCs.'

Ext Fig 2 "tSC1 activated" sounds like they were experimentally activated rather than spontaneously more active during tSC1.

We agreed with the Reviewer and changed it to 'most active during theta cycles strongly expressing tSC1'.

Line 72, "Indeed, entorhinally projecting orchid neurons of the MS can couple to entorhinal gamma oscillations Ref 64". In that work, CA1 gamma oscillations originated from the entorhinal cortex.

Fixed.

Line 85 "...neurons best phase locked to the future of tSC signals" needs clarification.

We rephrased this part: 'most MS neurons best locked to phase values of the tSC signals occurring at a small temporal delay'.

Line 106 "intra-burst inter-spike intervals corresponding to beta/gamma band activity": sounds like intraburst intervals can be outside fast frequency bands, by definition they usually correspond to beta and gamma. Alternatively, e.g.inter-spike intervals matching those of beta and gamma bands.

We removed 'intra-burst' from the sentence.

Line 1413 wavelet spectrogram of which time series are referred to here? (of MS spike trains?)

MS spike trains convolved by an 8-ms Gaussian kernel. We added this information.

Line 123-124 "..an average of 30% of all theta cycles contained at least one strong tSC". It is a little surprising that only 30 % of cycles show tSC while certainly more cycles show the pattern of phase-modulated fast oscillations of LFP in the mouse.

Indeed, tSC expression strength across theta cycles shows a continuum. Following Lopes-dos-Santos et al. (2018), we defined 'strong tSC expression' by a thresholding algorithm detailed in the Methods section; however, many theta cycles expressed tSCs 'weakly' to a variable extent.

Lines 174-175 "MS neurons often show phase-locked rhythmic bursting activity during hippocampal theta oscillations 48,50,68": also King, Recce, O'Keefe , EJN, 98.

Thank you; the citation has been added.

Fig. 1f, how were the LFP amplitudes for different frequencies according to the theta phase computed and normalized?

Spectrograms of the supra-theta signal in Figs. 1f, 7a and Extended Data Fig. 14a were computed for each detected theta cycle with a set of complex Morlet wavelets with main frequencies ranging from the upper limit of the theta band to 200 Hz, in 1 Hz steps (we have added the relevant figure references in Methods, line 1060 and a note in the figure legend).

How were the data in Ext Data Fig 1 b and c, 2b exactly normalised with their grand average?

We calculated the grand average of these properties across all theta cycles (P_{all}) and used this value to normalize the average of the same property across theta cycles strongly expressing a given tSC (P_{tSC}) with the following formula:

$$P_{tSC, norm} = \frac{P_{tSC} - P_{all}}{P_{all}}$$

We expanded the corresponding Methods section (lines 1000-1004).

The weight of tSC5 is 179 Hz but the maximum of the corresponding curve in Fig 1g seems to be above 200 Hz.

Fig. 1g was an example session (same as panel 1f), whereas the numbers corresponded to the peak frequency of the average power spectra of tSCs across all sessions. These averages were presented in Extended Data Figure 1a. We agreed that it was confusing and switched Figure 1g and Extended Data Figure 1a.

Please clarify AAV injections for different experiments in methods. Lines 609-611: Why were AAVs injected into the dorsal hippocampus? There seems to be no experiment using this preparation.

We were conducting a parallel experiment in the animals focusing on hippocampo-septal feedback. As stated in the Methods, these sessions were not analyzed in the previous version of the manuscript. However, in response to Reviewer questions regarding the possible role of this feedback loop in gamma generation, we now discuss these experiments in the revised manuscript (see Figure R6 and R12). The SST-IRES-Cre mouse line and the above AAV constructs were used in a large number of studies without reporting adverse phenotypic changes, and results of these studies were interpreted as general for mice and not specific for the particular strain⁵⁻⁹. Therefore, we do not consider using these mice a significant limitation of our study, especially as the results fit well with the juxtacellular recordings.

Reviewer #3

The study by Kiraly B et al provides evidence that the MS contributes to CA1 activity dynamics, suggesting that the MS orchestrates hippocampal network activity through multiple temporal scales. Generally, this study was well designed, the data were clearly interpreted, and the manuscript was well organized and written. However, I have some concerns or comments as below:

We thank the Reviewer for appreciating our study and for their constructive comments.

1. The hippocampus is divided into two portions: dorsal hippocampus and ventral hippocampus, which have different functions. The dorsal portion is primarily for learning and memory while the ventral portion is for emotions. Is there any difference in the MS orchestrating neuron activities between the dorsal and ventral portions?

We thank the Reviewer for raising this interesting question. We focused on the dorsal hippocampus based on the broad literature of theta and gamma oscillations in the dorsal CA1 (dCA1), which include studies on different sources of gamma oscillation with different frequencies^{4,16,17}, laminar profile and CSD analyses of multiple oscillations¹⁸⁻²¹, and, importantly, a demonstration of the capability of the Empirical Mode Decomposition method to extract meaningful theta-nested spectral components with functional

significance¹, which are all important foundations of the present study. Nonetheless, once we established the role of the MS in controlling supra-theta oscillations in the dCA1, it is exciting to ask whether these mechanisms generalize to the ventral hippocampus, with possible relevance to the processing of emotions.

As suggested above, to approach this question, we had to start from an earlier step and examine whether the ventral hippocampal theta cycles are characterized by the same cross-frequency coupling mechanisms we have seen in the dorsal recordings. In order to do this, we used the same analysis pipeline to search for theta-nested spectral components in ventral hippocampal LFP recordings from mice in an elevated plus maze downloaded from a public repository (<https://datadryad.org/stash/dataset/doi:10.7272/Q6ZP44B9>)²². We found distinct tSCs following the same characteristic frequencies in the beta to mid-gamma range as their dCA1 counterparts (Figure R10). The theta-gamma phase amplitude coupling was largely similar to the dCA1, with tSC2 appearing closer to the theta trough (animals 860 and 858). One animal (4639) showed minor deviations from this pattern, likely due to layer specific differences that are also present in the dCA1. One consistent difference between the dorsal and ventral hippocampus was the frequency and phase preference of tSC5. tSC5 measured in the ventral CA1 was typically much slower (with a peak around 80-90 Hz) and preferred the same phase as the mid-gamma components. In contrast, tSC5 in the dCA1 had a peak above 100 Hz, coupled to theta troughs. Thus, while tSC5 in dCA1 is thought to reflect local processing^{21,23}, ventral hippocampal tSC5 might rather reflect conceptually different network mechanisms, possibly controlled by external inputs similarly to mid-gamma components.

There are multiple observations making it reasonable to expect that the MS has a similar role in orchestrating ventral hippocampal neuronal activities. First, there are broad MS PV projections to the ventral part, too (^{24,25}; see also our response to Point #3). Second, from the viewpoint of oscillations, the hippocampus might be better described by a gradient change (for example in theta power or coherence) than by two distinct portions. Since the phase relationships between different types of theta-coupled interneurons are maintained²⁶ across the septo-temporal axis, different MS cell types selectively targeting specific types of interneurons across the entire axis may orchestrate activity in a coordinated manner. Indeed, Teevra cells recorded and labeled juxtacellularly projected to different positions across the septo-temporal axis, and we found tSC-coupled cells from the more temporal side as well. While these are exciting follow-up questions, these experiments are generally of at least the same volume as the ones we demonstrated, and thus beyond the scope of a single study. Nevertheless, we added our analysis of the ventral hippocampal tSCs discussed above, to provide a significant first step towards these goals (last paragraph of the Results section; Extended Data Figure 21).

a

animal ID: 860

**b**
animal ID: 4639

animal ID: 858

Figure R10 (Extended Data Figure 21). Theta nested spectral components in the ventral hippocampal LFP of mice in an elevated plus maze. **a** Mean amplitude of supra-theta spectral components as a function of theta phase in three example mice, for all cycles (top left) and for cycles strongly expressing a given tSC. Two theta cycles are shown, indicated by white cosine curves. **b** The frequency content of each tSC in the same example sessions as in panel **a**. Peak frequencies are shown in brackets. Data downloaded from a public repository (<https://datadryad.org/stash/dataset/doi:10.7272/Q6ZP44B9>)²²

2. It remains unclear about the physiological relevance of MS orchestrating CA1 neurons, and what physiological or pathological conditions can alter or rewire the ability for MS to modulate dorsal and ventral CA1 neurons. Moreover, if chemogenetic or optogenetic assisted loss of function or gain of function of the septal-hippocampal projections can correct or reverse the altered CA1 activities in certain conditions, it would gain significant influence of this study in the research fields.

We agree with the Reviewer that these are very exciting future question that provide further relevance to our findings presented here. To start to tackle this, we need to first narrow down the broad question ‘what physiological or pathological conditions can alter or rewire the ability for MS to modulate dorsal and ventral CA1 neurons’. To this end, we recently edited a Frontiers article collection titled ‘The medial septum as a smart clock: New aspects of its function beyond pacemaking’, that explored this question in depth to provide clues with respect to the pathologies in which the MS might be involved. Specifically, authors reviewed and discussed the potential role of the MS in ‘oscillopathies’ such as Alzheimer’s disease, schizophrenia, anxiety and pain, and also suggesting appropriate MS stimulation strategies to alleviate the negative consequences of these conditions^{27,28}. Memory impairments related to the MS, and treatment approaches such as MS cholinergic stimulation and vagus nerve stimulation were also discussed²⁹. Griguoli and Pimpinella stressed the importance of the MS in social learning, suggesting that psychopathologies accompanied by the disruption of social life may be another condition in which MS-mediated oscillogenesis is impaired³⁰. We expect that exploring mouse models of these pathological conditions and test possible alterations of the MS orchestrating CA1 neurons will be exciting new directions in the future. We added a discussion (last paragraph) on this topic to highlight the importance of the questions raised by the Review.

3. Chr2-assisted circuit mapping and whole cell patch clamp recordings are needed to determine whether PV-expressing GABAergic neurons in the MS project to the neurons localized within the dorsal and ventral CA1, CA2, CA3, and/or DG respectively, which will identify the involved direct and indirect circuit pathways to increase the significance of this study.

We thank the Reviewer for this suggestion. We performed Chr2-assisted circuit mapping and whole cell patch clamp recordings in acute slices of the dorsal CA1 and CA3 and revealed direct MS PV GABAergic impact on PV+ and somatostatin-expressing (SOM+) neurons of these regions. We did not find a difference in IPSC amplitude between CA1 vs. CA3 neurons, soma-targeting vs. dendrite-targeting, or PV+ vs. SOM+ interneurons (Figure R11a-d). When we applied theta-modulated burst stimulation (tmS) *in vitro*, most hippocampal interneurons were capable of following the tmS1-3 burst stimulation patterns with IPSCs/IPSPs, which could translate to modulated firing patterns at the respective frequencies (see Figure R11). tmS4 did not entrain interneurons’ firing *in vitro*, contrary to some *in vivo* recorded interneurons. This was likely due to a less reliable, more stochastic drive of PV axons *in vivo*, resulting in an even distribution of evoked IPSPs through the tmS4 burst, likely reflecting spontaneous activity of MS PV+ input

to CA1 better. Cut-off vs. intact PV+ MS axons, constant somatic depolarization-induced firing *in vitro* vs. synaptic input-driven firing *in vivo* and potential sampling bias could also contribute to this difference.

These new data were included in the 'Optogenetic stimulation of PV-expressing MS neurons evokes tSC-like CA1 activities' section of Results and in Extended Data Figure 13. We also point to the analysis of anatomically identified MS neurons with known projection targets in the 'Differential coupling of anatomically identified MS cells to CA1 tSCs' section of Results.

Figure R11 (Extended Data Figure 13). CA1 and CA3 PV+ and SOM+ interneurons are targeted by PV+ MS fibers. **a** Top, schematic of the acute slice electrophysiology experimental design. Cre-dependent AAV vector containing ChR2-eYFP was injected into the MS of PV-Cre animals (male, n = 6). After 2 weeks of expression time, animals were sacrificed, and coronal acute hippocampal slices were prepared to characterize the PV+ MS inputs to hippocampal interneurons using a series of tmS patterns. Bottom, example of a hippocampal coronal slice with interneurons recorded in the CA3 and CA1 regions (red, biocytin) and PV+ axons arising from the MS expressing ChR2-eyfp (green). Scale bar: 500 μ m. **b** Example image of a perisomatic PV+ fast-spiking interneuron recorded in the CA1 region, receiving PV+ MS inhibitory inputs. Scale bar: 100 μ m. **c** Spiking pattern (top) and evoked inhibitory currents in response to optogenetic stimulation (2 ms pulse width, 2 pulses @22Hz) of MS fibers (bottom) of the example neuron presented in panel **b** (n = 10 trials overlaid in grey, average in red). Scale bars: top - 200 ms, 20 mV; bottom - 20 ms, 100 pA. **d** Comparison of IPSC amplitudes evoked optogenetically by pulse pairs show moderate but significant short-term depression (n = 29 recorded cells; 2 pulses delivered @22 Hz; P1 and P2 medians, 165.41 and 119.94 pA, respectively). No significant differences were found between IPSC amplitudes evoked in CA1 versus CA3 (n = 15 and n = 14, respectively), soma-targeting versus dendrite-targeting (n = 7 and n = 13) or PV+ versus SOM+ (n = 12 and n = 7) interneurons. Example image of a PV+ and a SOM+ neuron receiving PV+ MS input are shown on the right (scale bar, 10 μ m). **e** Evoked inhibitory currents in response to optogenetic stimulation at different tmS patterns (n = 10 trials overlaid in grey, average in red; scale bar, 100 pA). **f** Effect of tmS stimulation on the spiking of the recorded neuron in panel **e** (n = 50 trials in grey, a single trial is shown in red; scale bar, 20 mV). **g** Raster plot of action potentials of the example neuron for the same tmS patterns (scale bar, 200 ms). **h** Normalized spike histograms of n = 17 cells upon tmS stimulation corresponding to the marked time window in panel **g**. Each line represents normalized spike histogram of a recorded cell, with darker colors indicating higher values. **i** Average normalized spike histograms for the n = 17 cells. Arrowheads highlight peaks in histograms indicating spike modulation by tmS1-3, absent case of tmS4.

4. Does CA1 modulate MS neurons, particularly the PV neurons in the MS?

The septum receives inputs from hippocampal somatostatin-expressing (SOM+) GABAergic neurons, forming a reciprocally connected inhibitory circuit³¹⁻³⁵. To fully explore the role of this circuit in the generation of the supra-theta oscillations, we used SwiChR (n = 4 mice) and ChR2 (n = 2 mice) to inhibit or excite hippocampo-septal projections in SOM-IRES-Cre mice specifically expressing the optogenetic actuator in CA1 SOM+ neurons.

First, we found that significantly more theta cycles expressed strong tSCs during the inhibition of hippocampo-septal projections compared to control periods without photostimulation. This suggests that the inhibitory feedback pathway may play an important role in suppressing septo-hippocampal supra-theta controlling mechanisms. These results are introduced in our response to Reviewer #2 and displayed in Figure R6 (included in the revised version in Figure 6f and g).

Next, we went one step further and examined the impact of optogenetic suppression of SOM+ hippocampo-septal projections in the MS on tSC-coupled and non-coupled MS neurons. More tSC-coupled MS neurons showed a >10% firing rate increase during stimulation (tSC-coupled, 20%; non-coupled, 13%). These neurons were likely directly targeted by the hippocampo-septal pathway and thus disinhibited upon inhibition of the hippocampo-septal GABAergic fibers. We also found MS neurons that showed firing rate

suppression, possibly via the disinhibited neurons. We found that less tSC-coupled MS neurons were suppressed than non-coupled neurons (tSC-coupled, 32%; non-coupled, 39%; Figure R12a).

Next, we tested the effect of 20 Hz optogenetic stimulation of SOM+ fibers in the Fimbria on tSC-coupled and non-coupled MS neurons. Many MS neurons showed firing rate suppression, which might be due to direct targeting of these cells by the hippocampo-septal inhibitory pathway. Some MS neurons showed firing rate increase, possibly disinhibited through the suppressed group. This population was more numerous among the not tSC-coupled MS neurons compared to the tSC-coupled ones (tSC-coupled, 6%; non-coupled, 18%; Figure R12b).

We found a significantly higher proportion of tSC-coupled neurons among the MS neurons activated upon SwiChR-mediated inhibition of hippocampo-septal GABAergic fibers than among the suppressed ones (65% vs. 48%, chi-squared test, $p = 0.0322$). This difference was due to a larger percentage of 'follower' neurons (ones with positive lags in the Z-shift analysis, demonstrating activity changes following/lagging CA1 activity changes) in the activated group (Figure R12c). In accordance, during optogenetic stimulation of the hippocampo-septal pathway, we found the lowest proportion of tSC-coupled neurons among the activated group, which was significantly less compared to the non-reactive neurons (31% vs. 63%, chi-squared test, $p = 0.0428$, Figure R12c). These results may indicate that there might be more tSC-coupled neurons among MS neurons directly targeted by the hippocampo-septal feedback. We included these results as a new Results section titled 'GABAergic septo-hippocampal feedback suppresses tSC-coupling in the MS'.

Figure R12 (Extended Data Figure 19). The impact of optogenetic manipulations of the hippocampo-septal pathway on MS neurons with different tSC-coupling. a Proportion of MS neurons that showed more than 10% firing rate increase or decrease upon SwiChR-mediated inhibition of the hippocampo-septal projections, shown separately for tSC-coupled and non-coupled MS neuron populations. **b** Same as panel **a** during optogenetic activation of the hippocampo-septal projections. **c** Proportion of tSC-coupled ‘predictor’ neurons (maximal tSC phase-locking strength realized by negative time lags), tSC-coupled ‘follower’ neurons (maximal tSC phase-locking strength realized by positive time lags) and not tSC-coupled neurons among the activated, not reactive and suppressed groups of MS neurons.

References

1. Lopes-dos-Santos, V. *et al.* Parsing Hippocampal Theta Oscillations by Nested Spectral Components during Spatial Exploration and Memory-Guided Behavior. *Neuron* **100**, 940-952.e7 (2018).
2. Hasselmo, M. E., Wyble, B. P. & Wallenstein, G. V. Encoding and retrieval of episodic memories: role of cholinergic and GABAergic modulation in the hippocampus. *Hippocampus* **6**, 693-708 (1996).
3. Colgin, L. L. & Moser, E. I. Gamma Oscillations in the Hippocampus. *Physiology* **25**, 319-329 (2010).
4. Colgin, L. L. *et al.* Frequency of gamma oscillations routes flow of information in the hippocampus. *Nature* **462**, 353-357 (2009).
5. Adler, A., Zhao, R., Shin, M. E., Yasuda, R. & Gan, W.-B. Somatostatin-Expressing Interneurons Enable and Maintain Learning-Dependent Sequential Activation of Pyramidal Neurons. *Neuron* **102**, 202-216.e7 (2019).
6. Cummings, K. A. & Clem, R. L. Prefrontal somatostatin interneurons encode fear memory. *Nat. Neurosci.* **23**, 61-74 (2020).
7. Kim, T. *et al.* Cortically projecting basal forebrain parvalbumin neurons regulate cortical gamma band oscillations. *Proc. Natl. Acad. Sci.* **112**, 3535-3540 (2015).
8. Viney, T. J. *et al.* Shared rhythmic subcortical GABAergic input to the entorhinal cortex and presubiculum. *Elife* **7**, 1-35 (2018).
9. Kvitsiani, D. *et al.* Distinct behavioural and network correlates of two interneuron types in prefrontal cortex. *Nature* **498**, 363-6 (2013).
10. Robinson, J. *et al.* Optogenetic activation of septal glutamatergic neurons drive hippocampal theta rhythms. *J. Neurosci.* (2016) doi:10.1523/JNEUROSCI.2141-15.2016.
11. Bender, F. *et al.* Theta oscillations regulate the speed of locomotion via a hippocampus to lateral septum pathway. *Nat. Commun.* (2015) doi:10.1038/ncomms9521.
12. Quirk, C. R. *et al.* Precisely timed theta oscillations are selectively required during the encoding phase of memory. *Nat. Neurosci.* (2021) doi:10.1038/s41593-021-00919-0.
13. Hajszan, T., Alreja, M. & Leranth, C. Intrinsic vesicular glutamate transporter 2-immunoreactive input to septohippocampal parvalbumin-containing neurons: Novel glutamatergic local circuit cells. *Hippocampus* (2004) doi:10.1002/hipo.10195.
14. Kocsis, B. *et al.* Huygens synchronization of medial septal pacemaker neurons generates hippocampal theta oscillation. *bioRxiv* 1-55 (2021) doi:10.1101/2021.01.22.427736.
15. Jinno, S. & Kosaka, T. Immunocytochemical characterization of hippocamposeptal projecting GABAergic nonprincipal neurons in the mouse brain: A retrograde labeling study. *Brain Res.* (2002) doi:10.1016/S0006-8993(02)02804-4.
16. Belluscio, M. A., Mizuseki, K., Schmidt, R., Kempter, R. & Buzsáki, G. Cross-frequency phase-phase

coupling between θ and γ oscillations in the hippocampus. *J. Neurosci.* **32**, 423–35 (2012).

17. Csicsvari, J., Jamieson, B., Wise, K. D. & Buzsáki, G. Mechanisms of Gamma Oscillations in the Hippocampus of the Behaving Rat. *Neuron* **37**, 311–322 (2003).
18. Lasztóczy, B. & Klausberger, T. Layer-Specific GABAergic Control of Distinct Gamma Oscillations in the CA1 Hippocampus. *Neuron* **81**, 1126–1139 (2014).
19. Lasztóczy, B. & Klausberger, T. Hippocampal Place Cells Couple to Three Different Gamma Oscillations during Place Field Traversal. *Neuron* (2016) doi:10.1016/j.neuron.2016.05.036.
20. Fernández-Ruiz, A. *et al.* Entorhinal-CA3 Dual-Input Control of Spike Timing in the Hippocampus by Theta-Gamma Coupling. *Neuron* **93**, 1213–1226.e5 (2017).
21. Schomburg, E. W. *et al.* Theta Phase Segregation of Input-Specific Gamma Patterns in Entorhinal-Hippocampal Networks. *Neuron* **84**, 470–485 (2014).
22. Cunniff, M. M., Markenscoff-Papadimitriou, E., Ostrowski, J., Rubenstein, J. L. R. & Sohal, V. S. Altered hippocampal-prefrontal communication during anxiety-related avoidance in mice deficient for the autism-associated gene *pogz*. *Elife* (2020) doi:10.7554/eLife.54835.
23. Klausberger, T. & Somogyi, P. Neuronal diversity and temporal dynamics: the unity of hippocampal circuit operations. *Science* **321**, 53–7 (2008).
24. Freund, T. F. GABAergic septohippocampal neurons contain parvalbumin. *Brain Res.* **478**, 375–381 (1989).
25. Joshi, A., Salib, M., Viney, T. J., Dupret, D. & Somogyi, P. Behavior-Dependent Activity and Synaptic Organization of Septo-hippocampal GABAergic Neurons Selectively Targeting the Hippocampal CA3 Area. *Neuron* **96**, 1342–1357.e5 (2017).
26. Forro, T., Valenti, O., Lasztóczy, B. & Klausberger, T. Temporal organization of GABAergic interneurons in the intermediate CA1 hippocampus during network oscillations. *Cereb. Cortex* (2015) doi:10.1093/cercor/bht316.
27. Ariffin, M. Z. *et al.* Forebrain medial septum sustains experimental neuropathic pain. *Sci. Rep.* (2018) doi:10.1038/s41598-018-30177-3.
28. Takeuchi, Y. *et al.* The Medial Septum as a Potential Target for Treating Brain Disorders Associated With Oscillopathies. *Frontiers in Neural Circuits* at <https://doi.org/10.3389/fncir.2021.701080> (2021).
29. Broncel, A., Bocian, R., Kłos-Wojtczak, P. & Konopacki, J. Medial septal cholinergic mediation of hippocampal theta rhythm induced by vagal nerve stimulation. *PLoS One* (2018) doi:10.1371/journal.pone.0206532.
30. Griguoli, Marilena, and D. P. Medial septum: relevance for social memory. *Front. Neural Circuits* (2022).
31. Damborsky, J. C. & Yakel, J. L. Regulation of hippocamposeptal input within the medial septum/diagonal band of Broca. *Neuropharmacology* **191**, 108589 (2021).
32. Manseau, F., Goutagny, R., Danik, M. & Williams, S. The hippocamposeptal pathway generates rhythmic firing of GABAergic neurons in the medial septum and diagonal bands: an investigation

using a complete septohippocampal preparation in vitro. *J. Neurosci.* **28**, 4096–107 (2008).

33. Toth, K., Borhegyi, Z. & Freund, T. F. Postsynaptic targets of GABAergic hippocampal neurons in the medial septum-diagonal band of Broca complex. *J. Neurosci.* **13**, 3712–3724 (1993).
34. Melonakos, E. D., White, J. A. & Fernandez, F. R. Gain Modulation of Cholinergic Neurons in the Medial Septum-Diagonal Band of Broca Through Hyperpolarization. *Hippocampus* **26**, 1525–1541 (2016).
35. Mattis, J. *et al.* Frequency-Dependent, Cell Type-Divergent Signaling in the Hippocamposeptal Projection. *J. Neurosci.* **34**, 11769–11780 (2014).

REVIEWER COMMENTS

Reviewer #1 (Remarks to the Author):

The authors have satisfactorily responded to mine, and in my opinion, also to the comments of other reviewers. The revised version of the manuscript delivers an even stronger message related to the largely overlooked role of medial septum networks in controlling supra-theta oscillations. I congratulate the authors on their insightful work.

Reviewer #2 (Remarks to the Author):

The authors nicely revised the manuscript, adding many new interesting analytical results, further investigating the circuit dynamics using optogenetics, and clarifying the interpretations. Apart from a few minor comments, my main remaining concern is a still insufficient analysis and the apparent misinterpretation of control stimulation results.

1. The new analysis of beta-gamma components during theta burst and theta single pulse (at 8 Hz) stimulation is informative about oscillations amplitude but not about phase-amplitude coupling (the point made in the third original comment). It is still not clear to which results the reformulated statement (lines 601-604) refers to: "Additionally, analyzing tSCs separately allowed us to reveal that beta/gamma components evoked by theta rhythmic PV stimulation did not show the characteristic phase-amplitude coupling of spontaneous tSCs and tSCs evoked by theta-burst stimulation.". I am guessing it refers to the interpretation of example recordings for each condition in Fig 1f, 6b and Extended Fig. 17c. If the latter is based on the recording shown Extended Fig. 17a, this str. radiatum signal could be recorded near the "theta-null" zone, leading to partial theta phase distortions and a potentially spuriously dominant amplitude of slow supra-theta frequencies. This interferes with the evaluation of true beta-gamma components. Still, the coupling, which is denied in lines 601-604, seems evident in the Extended Fig. 17c (for stSC2 and 3). An LFP of spontaneous recording from the same session, and the respective average colour-coded amplitude of supra-theta spectral components vs. phase, could aid interpretation of stimulation effects. Furthermore, as mentioned in the original comment, precise quantitative comparisons of the cross-frequency coupling and its variance across animals should be provided for theta single pulse and theta burst conditions. Presently, lines 601-604 are at odds both with the literature cited in the original comment and with the Extended Data Fig. 17b (middle panel) which does show a cross-frequency coupling during theta single pulse stimulation (8 Hz). Alternatively, if the theta single pulse (at 8Hz) protocol is not directly relevant to the examination of the sufficiency of theta-burst inputs for the generation of stSCs, the authors may consider leaving the theta single pulse protocol out.

2. Related to the fourth original comment. The analysis shown in Extended Data Figure 11 is interesting but not entirely clear. Optimal lags of MS discharge are estimated elsewhere in the manuscript below 10 ms, but Extended Data Figure 11 indicates they are around 100 ms from the tSC-trough, i.e., well beyond the duration of a gamma envelope.

3. The saturation of colours in Extended Data Fig. 17b should be adjusted according to the range in the colour bar.

4. Is the duration of laser pulses provided? It seems not to be included in the Methods for acute optogenetic stimulation experiments.

5. In Extended Data Fig. 2a, it is interesting that preferred phases of MS cells are broadly distributed. How does this relate to the theta rhythmicity of MS cells and the phase coordination, or lack thereof, within their population? What would Extended Data Fig. 2a histograms look like, including only MS cells with a high theta rhythmicity, are they locked to similar phases?

Reviewer #3 (Remarks to the Author):

The authors substantially addressed my concerns, and I do not have additional questions.

Response to Reviewers: The medial septum controls hippocampal supra-theta oscillations

Structure:

Reviewer comments: black, italic

Our replies: blue

We thank the Reviewers for the overall positive evaluation of our revised study. We addressed the remaining points of the Reviewers:

1. We agreed with Reviewer #2 that the single-pulse protocol may not be directly relevant to the examination of the theta-nested gamma burst inputs, and therefore omitted this part.
2. We provided new analysis on the theta-phase preference of MS neurons in theta cycles with different tSCs.
3. We fixed the remaining issues and added the missing information noted by Reviewer #2.

All additions to the text are red-lined in the revised manuscript. Our point-by-point responses are laid out in details below.

Reviewer #1

The authors have satisfactorily responded to mine, and in my opinion, also to the comments of other reviewers. The revised version of the manuscript delivers an even stronger message related to the largely overlooked role of medial septum networks in controlling supra-theta oscillations. I congratulate the authors on their insightful work.

We thank the Reviewer for appreciating our study.

Reviewer #2

The authors nicely revised the manuscript, adding many new interesting analytical results, further investigating the circuit dynamics using optogenetics, and clarifying the interpretations. Apart from a few minor comments, my main remaining concern is a still insufficient analysis and the apparent misinterpretation of control stimulation results.

We greatly appreciate the Reviewer's positive evaluation of our work and the important comments, which have very significantly contributed to the manuscript. We discuss the remaining concerns point-by-point below.

1. *The new analysis of beta-gamma components during theta burst and theta single pulse (at 8 Hz) stimulation is informative about oscillations amplitude but not about phase-amplitude coupling (the point made in the third original comment). It is still not clear to which results the reformulated statement (lines 601-604) refers to: "Additionally, analyzing tSCs separately allowed us to reveal that beta/gamma components evoked by theta rhythmic PV stimulation did not show the characteristic phase-amplitude coupling of spontaneous tSCs and tSCs evoked by theta-burst stimulation.". I am guessing it refers to*

the interpretation of example recordings for each condition in Fig 1f, 6b and Extended Fig. 17c. If the latter is based on the recording shown Extended Fig. 17a, this str. radiatum signal could be recorded near the "theta-null" zone, leading to partial theta phase distortions and a potentially spuriously dominant amplitude of slow supra-theta frequencies. This interferes with the evaluation of true beta-gamma components. Still, the coupling, which is denied in lines 601-604, seems evident in the Extended Fig. 17c (for stSC2 and 3). An LFP of spontaneous recording from the same session, and the respective average colour-coded amplitude of supra-theta spectral components vs. phase, could aid interpretation of stimulation effects. Furthermore, as mentioned in the original comment, precise quantitative comparisons of the cross-frequency coupling and its variance across animals should be provided for theta single pulse and theta burst conditions. Presently, lines 601-604 are at odds both with the literature cited in the original comment and with the Extended Data Fig. 17b (middle panel) which does show a cross-frequency coupling during theta single pulse stimulation (8 Hz). Alternatively, if the theta single pulse (at 8Hz) protocol is not directly relevant to the examination of the sufficiency of theta-burst inputs for the generation of stSCs, the authors may consider leaving the theta single pulse protocol out.

We thank the Reviewer for this important comment. We agree that the quoted sentence was misleading, for which we apologize. We did not mean to suggest that there was no phase-amplitude coupling during 8 Hz stimulation, but rather to point out that it does not follow the characteristics (i.e., theta phase preference) of spontaneous theta-nested gamma bursts and those evoked by theta-burst stimulation. However, we agree with the Reviewer that the single-pulse protocol may not be directly relevant to the examination of the theta-nested gamma burst inputs, and therefore we have chosen to leave it out.

2. Related to the fourth original comment. The analysis shown in Extended Data Figure 11 is interesting but not entirely clear. Optimal lags of MS discharge are estimated elsewhere in the manuscript below 10 ms, but Extended Data Figure 11 indicates they are around 100 ms from the tSC-trough, i.e., well beyond the duration of a gamma envelope.

We thank the Reviewer for catching this error. The time values in panel b were not correct; we have now fixed this in the revised figure (also attached below). The correct value of the middle tick is 0 ms, which corresponds to the largest tSC trough and confirms that the optimal lag (indicated by the red line) is indeed below 10 ms.

Reviewer Figure 1 (Extended Data Figure 11b). Peri-event time histogram (PETH) of an example neuron triggered on the largest troughs of CA1 tSC1 cycles. The blue line shows the bandpass filtered PETH (18 Hz-35 Hz), which was used to find tSC1-related peaks and troughs before and after the optimal time lag (red line) that realizes the strongest phase locking.

3. The saturation of colours in Extended Data Fig. 17b should be adjusted according to the range in the colour bar.

Extended Data Fig. 17b was removed from the revised manuscript in accordance with our answer to the first point of the Reviewer.

4. Is the duration of laser pulses provided? It seems not to be included in the Methods for acute optogenetic stimulation experiments.

The pulse duration (2 ms) was only indicated in the figure legend and indeed missing from the corresponding Methods section. We thank the Reviewer for noticing this. We added this information to the revised manuscript.

5. In Extended Data Fig. 2a, it is interesting that preferred phases of MS cells are broadly distributed. How does this relate to the theta rhythmicity of MS cells and the phase coordination, or lack thereof, within their population? What would Extended Data Fig. 2a histograms look like, including only MS cells with a high theta rhythmicity, are they locked to similar phases?

We thank the Reviewer for raising this interesting question. We first plotted the histograms for theta rhythmic cells only and found that the preferred phase of the rhythmic cells was still broadly distributed. Therefore, we took a step further and examined two theta rhythmic subtypes separately: the constitutive bursting and the theta-associated bursting cells (see Extended Data Figure 3). This analysis revealed that constitutive bursting neurons were more coupled to the trough and the ascending phase of the theta, while theta-associated bursting neurons were less frequently coupled to the ascending phase and instead exhibited a broad distribution around the peak, the descending phase, and the trough. These characteristics were largely preserved between theta cycles expressing different tSCs.

Reviewer Figure 2. Theta-phase preference of MS neurons in theta cycles with different tSCs. Rose diagram (circular histogram) of the preferred theta phase of theta-coupled, theta rhythmic, constitutive bursting and theta-associated bursting MS neurons during theta cycles expressing different tSCs.

Reviewer #3

The authors substantially addressed my concerns, and I do not have additional questions.

We thank the Reviewer for the constructive suggestions and the positive evaluation of our work.

REVIEWERS' COMMENTS

Reviewer #2 (Remarks to the Author):

The authors have properly addressed my concerns and have substantially improved the manuscript. I have no further comments.

Response to Reviewers: The medial septum controls hippocampal supra-theta oscillations

Structure:

Reviewer comments: black, italic

Our replies: blue

Reviewer #2

The authors have properly addressed my concerns and have substantially improved the manuscript. I have no further comments.

We thank the Reviewer for the important suggestions and for appreciating our study.